# A type 1 immunity-restricted promoter of the IL−33 receptor gene directs antiviral T-cell responses

**Tobias M. Brunner** [1,2,9] ✉, **Sebastian Serve** [1,2,3,9], **Anna-Friederike Marx**[4], **Jelizaveta Fadejeva**[1,2], **Philippe Saikali** [1,2], **Maria Dzamukova** [1,2], **Nayar Durán-Hernández** [1,2], **Christoph Kommer**[5,6], **Frederik Heinrich** [7], **Pawel Durek**[7], **Gitta A. Heinz** [7], **Thomas Höfer** [5,6], **Mir-Farzin Mashreghi** [7], **Ralf Kühn** [8], **Daniel D. Pinschewer** [4] & **Max Löhning** [1,2] ✉

The pleiotropic alarmin interleukin-33 (IL-33) drives type 1, type 2 and regulatory T-cell responses via its receptor ST2. Subset-specific differences in ST2 expression intensity and dynamics suggest that transcriptional regulation is key in orchestrating the context-dependent activity of IL-33−ST2 signaling in T-cell immunity. Here, we identify a previously unrecognized alternative promoter in mice and humans that is located far upstream of the curated ST2-coding gene and drives ST2 expression in type 1 immunity. Mice lacking this promoter exhibit a selective loss of ST2 expression in type 1- but not type 2-biased T cells, resulting in impaired expansion of cytotoxic T cells (CTLs) and T-helper 1 cells upon viral infection. T-cell-intrinsic IL-33 signaling via type 1 promoter-driven ST2 is critical to generate a clonally diverse population of antiviral short-lived effector CTLs. Thus, lineage-specific alternative promoter usage directs alarmin responsiveness in T-cell subsets and offers opportunities for immune cell-specific targeting of the IL-33−ST2 axis in infections and inflammatory diseases.

Endogenous danger-associated molecules released upon cellular damage, so-called alarmins, act as central orchestrators of inflammation[1]. Amongst alarmins, IL-33 stands out as a potent cytokine that triggers pro- and anti-inflammatory responses by engaging its receptor ST2 on immune cells[2,3]. ST2, also known as T1 (refs. 4,5), was first detected on T-helper 2 (Th2) cells and mast cells[6–8], and its activation elicited production of type 2 cytokines, suggesting an important role in type 2 immunity[9–11]. In accordance, disruption of IL-33−ST2 signaling ameliorated type 2 airway inflammation in mice and impaired immunity against nematode infections[6,12–14]. More recently, studies established ST2 as a marker for type 2 innate lymphoid cells (ILC2s) and demonstrated a critical role of IL-33 for their development and function[15,16].

[1]Experimental Immunology and Osteoarthritis Research, Department of Rheumatology and Clinical Immunology, Charité – Universitätsmedizin Berlin, corporate member of Freie Universität Berlin and Humboldt-Universität zu Berlin, Berlin, Germany. [2]Pitzer Laboratory of Osteoarthritis Research, German Rheumatism Research Center (DRFZ), a Leibniz Institute, Berlin, Germany. [3]Berlin Institute of Health at Charité – Universitätsmedizin Berlin, BIH Biomedical Innovation Academy, Berlin, Germany. [4]Division of Experimental Virology, Department of Biomedicine, University of Basel, Basel, Switzerland. [5]Division of Theoretical Systems Biology, German Cancer Research Center (DKFZ), Heidelberg, Germany. [6]BioQuant Center, University of Heidelberg, Heidelberg, Germany. [7]Therapeutic Gene Regulation, German Rheumatism Research Center (DRFZ), a Leibniz Institute, Berlin, Germany. [8]Max Delbrück Center for Molecular Medicine in the Helmholtz Association (MDC), Berlin, Germany. [9]These authors contributed equally to this work: Tobias M. Brunner, Sebastian Serve. ✉e-mail: tobias.brunner@drfz.de; max.loehning@charite.de

By activating ILC2s and regulatory T (T_reg) cells, IL-33 controls tissue homeostasis, promotes wound healing and mitigates pathology in acute or chronic inflammation[3,17–22].

IL-33 has also emerged as a key driver of type 1 immune responses. It is released by fibroblastic reticular cells in lymphoid organs and promotes clonal expansion and activation of antiviral CTLs and T-helper 1 (Th1) cells to confer protection against replicating viruses[23–27]. Moreover, IL-33-mediated amplification of type 1 immune cells was shown to exacerbate tissue damage during graft-versus-host disease (GVHD)[28,29] and contribute to immune dysregulation in systemic inflammatory diseases[30].

Considering its multifaceted mode of action, IL-33 is now recognized to amplify pro- or anti-inflammatory T-cell subsets in a context-specific manner[31,32]. To accomplish this versatility, it was suggested that transcription of the ST2-coding gene interleukin-1 receptor-like 1 (*Il1rl1*) requires cell-type specific regulation, such that certain T-cell subsets become sensitive to IL-33 signals dependent on the inflammatory environment[31,32]. ST2 is absent from naive T cells but expressed constitutively at high levels by type 2-biased immune cells in which its transcription is controlled by the master-regulator transcription factor of type 2 immunity GATA-3 (refs. 6,21,33). In contrast, antiviral CTLs and Th1 cells express low levels of ST2 transiently upon infection, and this expression depends on STAT4 and T-bet, key transcription factors of type 1 immunity[24,27]. This dynamic expression pattern renders it difficult to study ST2 on type 1 immune cells. Consequently, the molecular mechanism allowing for T-cell lineage-specific ST2 expression patterns has remained enigmatic.

## Results

### Identification of a type 1 immunity-restricted *Il1rl1* promoter

Previous studies have shown that the protein-coding exons of the *Il1rl1* gene are preceded by two non-coding exons (exon 1a and exon 1b) located in a distal and proximal promoter region, respectively[4,34] (Fig. 1a). The proximal promoter drives ST2 expression in fibroblasts, whereas the distal promoter mediates ST2 expression in Th2 cells and mast cells[33,35]. To assess which promoter is used by type 1-polarized T cells, we generated CTLs, Th1 or Th2 cells in vitro, which all express substantial levels of ST2 (Fig. 1b and Extended Data Fig. 1a–c). Of note, at the per-cell level, type 1 T cells express less ST2 than Th2 cells (Fig. 1b). Thus, to stain ST2, we utilized a multi-step amplification protocol, yielding a more sensitive detection compared to stainings with frequently used ST2 antibodies (Extended Data Fig. 1d,e). By analyzing leader exons of 5′ untranslated regions (UTRs) of ST2-coding transcripts, we found that none of the described promoters could possibly account for the expression of *Il1rl1* by type 1-polarized T cells (Fig. 1c). Thus, to map the origin of *Il1rl1* transcripts in these cells, we next subjected ST2⁺ CTLs, Th1 and Th2 cells to RNA-sequencing (RNA-seq) analysis. Thereby, we discovered a transcriptional start site (TSS) located ~40 kb upstream of the annotated *Il1rl1* gene, which was selectively used in CTLs and Th1 cells (type 1 promoter) (Fig. 1d). This TSS gave rise to two *Il1rl1* transcript isoforms with distinct leader sequences but unaltered protein-coding sequences. We refer to the leader exons in these transcripts as exons A, B and D. Further, alternative splicing of exon B to exon C resulted in a transcript that did not contain ST2-coding exons. As expected, the 'distal' promoter (exon 1a) presented the primary origin of *Il1rl1* transcripts in Th2 cells (type 2 promoter). To assess the usage of the type 1 promoter in vivo, we next transferred naive lymphocytic choriomeningitis virus (LCMV)-specific T-cell receptor (TCR)-transgenic CD4⁺ T cells (Smarta) or LCMV-specific TCR-transgenic CD8⁺ T cells (P14) into wild-type (WT) mice and infected the recipients with LCMV. At day 7 postinfection (d7 p.i.), we reisolated transferred cells and quantified *Il1rl1* promoter usage. In contrast to Th2 cells, CTLs and Th1 cells had largely incorporated exons A and B but not exon 1a into 5′ UTRs of ST2-coding transcripts (Fig. 1e,f).

ST2 expression by CTLs and Th1 cells, but not by Th2 cells or T_reg cells, relies on IL-12 and the transcription factors T-bet and STAT4 (Extended Data Fig. 1f–j)[24,27]. Hence, we analyzed T-bet- and STAT4 binding as well as activation-induced changes in chromatin accessibility at the *Il1rl1* locus in type 1-polarized T cells using publicly available chromatin immunoprecipitation sequencing (ChIP-seq)[36–38] and assay for transposase-accessible chromatin sequencing (ATAC-seq)[39,40] data. Although GATA-3 binds predominantly upstream of the type 2 promoter, T-bet and STAT4 binding was detected in the vicinity of exons A and B, at sites that were inaccessible in naive T cells but accessible in LCMV-primed Th1 cells and CTLs (Fig. 1g).

Alternative promoters are abundant[41]. We were unaware, however, of other genes with comparably selective alternative promoter usage in type 1 and type 2 immune cells. We thus used RNA-seq data to conduct a genome-wide search to identify additional genes with highly type 1 or type 2 immunity-specific promoters[42]. *Il1rl1* was reliably detected when comparing Th2 cells to CTLs and Th1 cells but no other gene with lineage-specific TSS usage was found (Fig. 1h–j). This suggested that such highly restrictive type 1 and type 2 T-cell lineage-specific utilization of alternative first exons is rare, with the *Il1rl1* gene potentially representing a unique case.

### *Il1rl1* promoter usage is conserved between mice and humans

Next, we assessed DNA conservation at the type 1 promoter[43–45] and compared it to T-bet- and STAT4-ChIP-seq as well as ATAC-seq data (Fig. 2a). Thereby, we identified a 275-nt-spanning, well-conserved sequence located ~5 kb upstream of exon A (CNS-5), which is bound by T-bet and STAT4 in Th1 cells and is marked by a sharp ATAC-seq peak in CTLs (Fig. 2a). In contrast, the sequence surrounding prominent peaks ~1.5 kb downstream of exon A appears less conserved. Mapping of T-bet and STAT4 binding motifs within CNS-5 indicated that both transcription factors putatively bind in close proximity at sequences almost identical between mice and humans[46] (Fig. 2b).

To assess whether T-cell lineage-specific promoter usage at the *Il1rl1* locus is conserved between mice and humans, we first examined cap analysis of gene expression (CAGE) and transcriptomic data from the FANTOM5 resource[41,47], which provided evidence for a putative TSS upstream of the *IL1RL1* gene. In human Th1 cells this site is preceded by T-bet binding sites (Fig. 2c)[48]. Of note, the exon structure closely resembles the exons A and B identified in mice (cf. Fig. 1d). To determine whether this TSS is utilized in primary human CTLs and Th1 cells, we isolated in vivo-differentiated T cells from peripheral blood to quantify *IL1RL1* promoter usage (Extended Data Fig. 2a–f). Congruently with mouse T cells, *IL1RL1* transcripts of human CTLs and Th1 cells contained exons A and B within their 5′ UTRs, whereas Th2 cells had incorporated exon 1a (Fig. 2d). In summary, we have identified a previously unrecognized type 1 immunity-restricted *Il1rl1* promoter, which is instructed by the lineage-associated transcription factors T-bet and STAT4 and orchestrates ST2 expression in CTLs and Th1 cells of humans and mice.

### *Il1rl1* promoters allow lineage-specific targeting of ST2

Modulation of the IL-33–ST2 axis could represent a promising approach in treating inflammatory diseases[49]. For instance, blockade of IL-33 using therapeutic antibodies has shown encouraging efficacy in clinical trials of asthma and chronic obstructive pulmonary disease[50]. However, due to hard-to-predict effects on the balance between IL-33-mediated inflammation and tissue repair, fine-tuned targeting approaches may offer critical advantages. We thus asked whether lineage-specific promoters can be leveraged to target ST2 expression in a T-cell subset-specific manner. Hence, we retrovirally transduced T cells to express small-hairpin RNAs (shRNAs) targeting distinct *Il1rl1* 5′ UTRs (Fig. 3a). Selective downregulation of ST2 was achieved in either CTLs and Th1 cells or Th2 cells using shRNAs binding exons A and B or exon 1a, respectively (Fig. 3b,c).

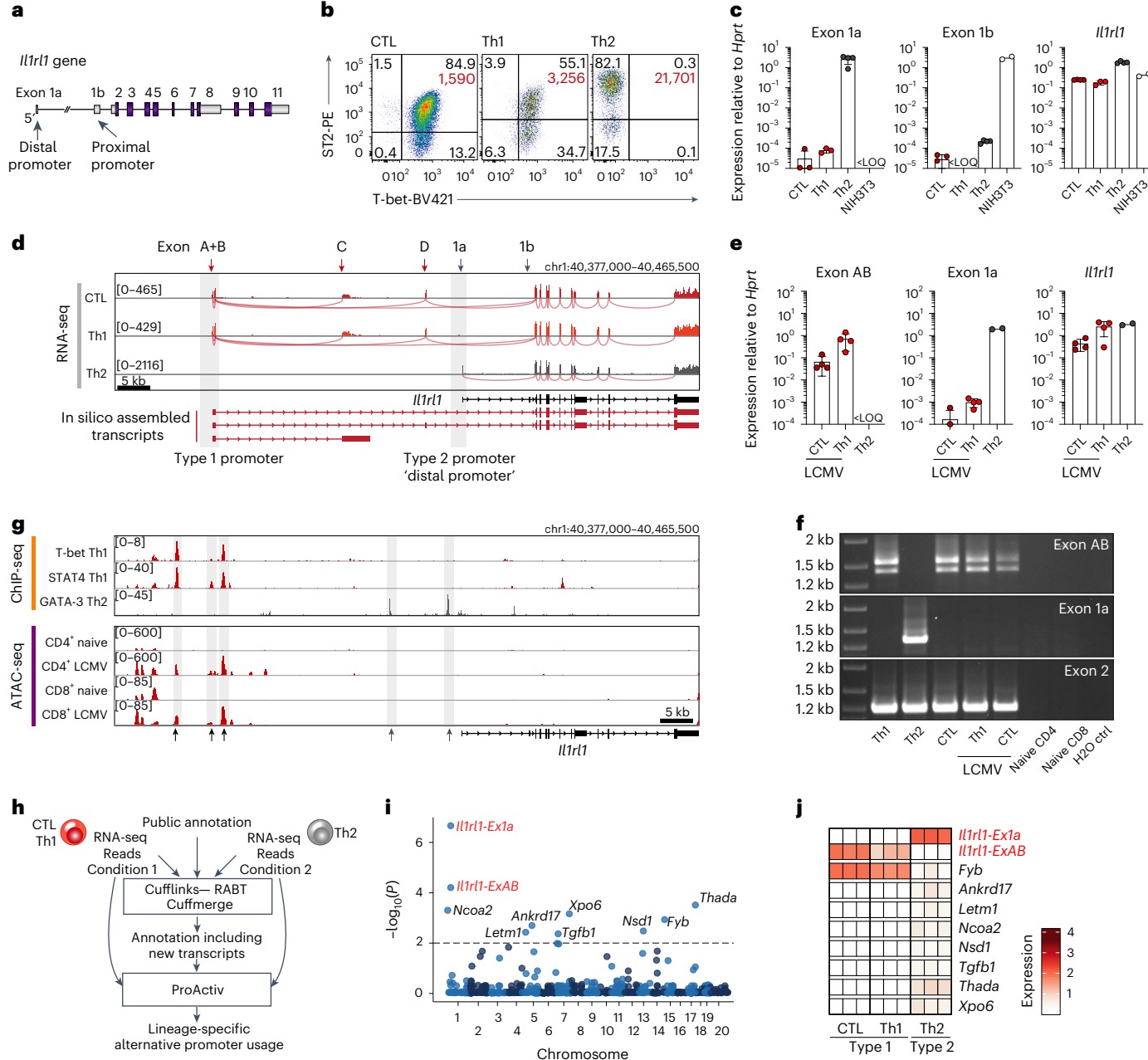

**Fig. 1 | A previously unrecognized alternative promoter drives IL-33 receptor expression in antiviral T cells. a**, Scheme depicting the curated *Il1rl1* gene. **b**, ST2 surface expression by in vitro differentiated T-cell subsets. Percentages in black, mean fluorescence intensity of ST2⁺ T cells in red. **c**, *Il1rl1* first exon usage by differentiated T cells (CTL: n = 4 with one sample less than the limit of quantification (LOQ) in exon 1a and 1b reactions, Th1: n = 3, Th2: n = 4, NIH3T3: n = 2). **d**, RNA-seq coverage and splice junction tracks of ST2⁺ CTLs and Th1 and Th2 cells (n = 3 per subset) at the *Il1rl1* locus. chr1:40,377,000–40,465,500; GRCm38.p6/mm10 is shown. **e,f**, WT mice received LCMV-specific P14 or Smarta T cells and were infected with LCMV-WE. P14 CTLs and Smarta Th1 cells were isolated on d7 p.i., and *Il1rl1* first exon usage was analyzed by qPCR (**e**; CTL: n = 4 with two samples <LOQ in exon 1a reaction, Th1: n = 4, Th2 control (ctrl) (in vitro):

n = 2) and RT-PCR (**f**). **g**, ChIP-seq tracks indicating T-bet[36], STAT4 (ref. 38) and GATA-3 (ref. 37) binding and ATAC-seq tracks showing chromatin accessibility in naive or activated LCMV-specific T cells[39,40]. **h**, Computational pipeline to identify alternative TSSs between type 1- (CTL, Th1) and type 2- (Th2) polarized T cells. **i**, Manhattan plot showing identified hits (parameters minAbs = 0.25 and promoter fold change (FC) = 2, dashed line: P = 0.01). **j**, Heatmap of identified TSSs with P < 0.01 and their respective ProActiv-normalized expression across all replicates. In **i** and **j**, red texts are used to highlight the important transcriptional start sites detected. Data in panels **c**, **e** and **f** are representative of two independent experiments. Data are presented as mean ± standard deviation, with each dot representing T cells isolated from individual mice. P was determined using two-tailed t-tests with Benjamini–Hochberg (BH) correction (**i** and **j**).

Furthermore, to study the type 1 immunity-restricted promoter in vivo, we generated knockout mice with a deletion of exons A and B (*Il1rl1-ExAB⁻/⁻*). A second mouse strain lacking exon C (*Il1rl1-ExC⁻/⁻*) was generated to control for potential effects of exon C-containing transcripts (Fig. 3d and Extended Data Fig. 3a–c). In steady state, both strains harbored normal numbers of T cells at an activation state

comparable to WT mice (Extended Data Fig. 3d–i). Likewise, introduced deletions did not affect the composition of splenic innate immune cells (Extended Data Fig. 3j–l). Importantly, deletion of the type 1 promoter severely impaired ST2 expression by in vitro activated CTLs and Th1 cells, whereas *Il1rl1-ExAB⁻/⁻* Th2 cells differentiated from the same pool of naive CD4⁺ T cells exhibited normal ST2 expression (Fig. 3e,f).

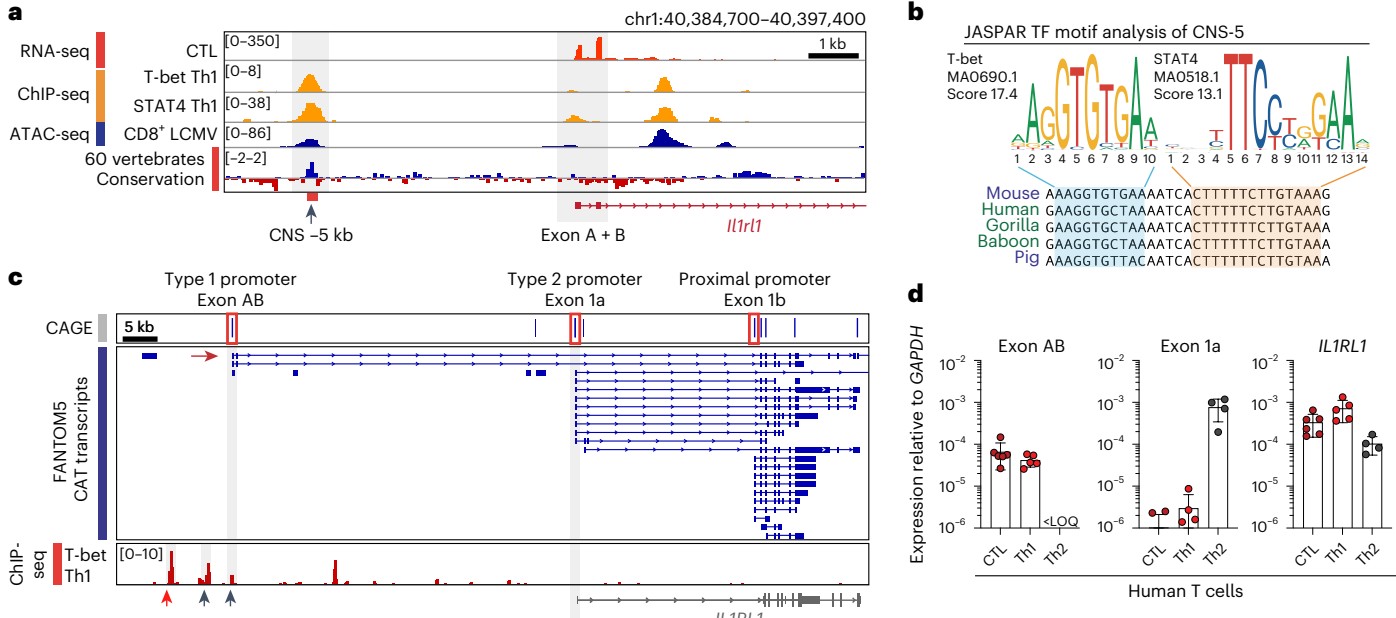

**Fig. 2 | Alternative promoter usage at the *Il1rl1* locus is conserved between mice and humans. a**, Integrated Genome Viewer (IGV) browser display of the *Il1rl1* type 1 promoter region (chr1:40,384,700–40,397,400; GRCm38.p6/mm10) showing a CTL RNA-seq track, T-bet and STAT4 binding in Th1 cells[36,38], chromatin accessibility in activated CTLs[39,40] and a PhyloP track indicating evolutionary conservation across 60 vertebrate species[43,45]. CNS-5, as identified using Vista[44], is highlighted (chr1:40,386,282–40,386,541; GRCm38.p6/mm10). **b**, T-bet and STAT4 binding motifs and their respective prediction score within CNS-5 as determined using JASPAR motif analysis[46] (chr1:40,386,429–40,386,459;

GRCm38.p6/mm10). **c**, IGV browser display of cap analysis of gene expression sequencing (CAGE-seq) data and corresponding CAGE-associated transcripts (CAT) as provided and published by the FANTOM5 consortium[41,47]. ChIP-data track showing T-bet binding in the vicinity of the type 1 promoter (arrows) in human Th1 cells[48] (chr2:102,862,400–102,970,100; GRCh37.13/hg19). The red arrow indicates the conserved region corresponding to mouse CNS-5. **d**, *IL1RL1* first exon expression in human T-cell subsets (CTL: *n* = 6, Th1: *n* = 5, Th2: *n* = 4). Data are presented as mean ± standard deviation, with each dot representing T cells isolated from individual donors. TF, transcription factor.

As expected, ST2 on *Il1rl1-ExC*[−/−] T cells was not reduced (Fig. 3e,f). Due to a lack of ST2 expression, *Il1rl1-ExAB*[−/−] CTLs and Th1 cells, but not *Il1rl1-ExAB*[−/−] Th2 cells, were unresponsive to IL-33 (Fig. 3g). Further, usage of the type 1 promoter is not limited to T cells, as natural killer (NK) T cells and NK cells of *Il1rl1-ExAB*[−/−] mice failed to express ST2 upon activation ex vivo (Extended Data Fig. 4a–f). To verify that ST2 expression is preserved on type 2-biased immune cells of *Il1rl1-ExAB*[−/−] mice in vivo, we analyzed peritoneal mast cells and lung ILC2s, which use the GATA-3-regulated type 2 *Il1rl1* promoter (Extended Data Fig. 5a). These cells indeed displayed normal ST2 expression in *Il1rl1-ExAB*[−/−] mice (Extended Data Fig. 5b–g). Lastly, bone marrow eosinophils and neutrophils of *Il1rl1-ExAB*[−/−] mice expressed ST2 at levels slightly reduced, but largely comparable to WT mice (Extended Data Fig. 5h–l).

To investigate the requirement of the type 1 *Il1rl1* promoter for ST2 expression by T cells responding to viral challenge, we infected WT, *Il1rl1*[−/−], *Il1rl1-ExAB*[−/−] and *Il1rl1-ExC*[−/−] mice with LCMV (Fig. 3h). ST2 surface expression was almost absent from CTLs in *Il1rl1-ExAB*[−/−] mice and was significantly reduced in Th1 cells at day 7 p.i., whereas T_reg cells displayed unaltered ST2 surface levels (Fig. 3i–n). Conversely, *Il1rl1-ExC*[−/−] CTLs and Th1 cells exhibited slightly enhanced ST2 expression (Fig. 3i–l), suggesting that exon C may act as a transcriptional decoy (cf. Fig. 1d). Altogether, we found that targeting of individual *Il1rl1* promoters allowed for a selective T-cell lineage-specific manipulation of ST2 expression.

**The type 1 *Il1rl1* promoter drives antiviral T-cell responses**
Next, we studied whether CD8⁺ T-cell responses to LCMV required the type 1 *Il1rl1* promoter. At the peak of the response (d7 p.i.), *Il1rl1-ExAB*[−/−] mice harbored substantially reduced numbers of CTLs in spleens and livers, resembling in its extent the impairment observed in *Il1rl1*[−/−] mice (Fig. 4a–c and Extended Data Fig. 6a–c). Diminished CTL counts were

largely accounted for by a reduction in CD44⁺CD62L⁻ effector T cells expressing the proliferation marker Ki67 (Fig. 4d–f). Accordingly, CTLs specific for the immunodominant GP₃₃₋₄₁ and NP₃₉₆₋₄₀₄ epitopes of LCMV were substantially reduced (Fig. 4g and Extended Data Fig. 6d–f). Ultimately, *Il1rl1-ExAB*[−/−] mice displayed significantly lower numbers of CTLs expressing effector cytokines or cytolytic molecules, and systemic IFN-γ levels were reduced by >70% (Fig. 4h,i and Extended Data Fig. 6g–i). In contrast and as expected, CTL responses of *Il1rl1-ExC*[−/−] mice were comparable to those of WT mice (Fig. 4b,c,e–i and Extended Data Fig. 6b–i).

To determine whether observed effects were due to a T-cell-intrinsic impairment in ST2 expression, mixed bone marrow chimeras were generated by reconstituting irradiated WT mice with bone marrow from *Il1rl1*[−/−], *Il1rl1-ExAB*[−/−] or *Il1rl1-ExC*[−/−] mice, each of them mixed 1:1 with WT bone marrow. Following LCMV infection, WT:*Il1rl1-ExC*[−/−] chimeras mounted CTL responses that derived at approximately equal parts from both bone marrow compartments. In contrast, WT bone marrow-derived CTLs outnumbered the CTLs derived from *Il1rl1*[−/−] or *Il1rl1-ExAB*[−/−] bone marrow in the respective chimeras (Fig. 4j–m and Extended Data Fig. 6j,k). Lastly, to study the impact of type 1 promoter-driven ST2 expression on T-cell responses in the absence of any potentially confounding irradiation effects, congenically marked naive *Il1rl1-ExAB*[−/−] P14 CTLs and WT P14 cells were cotransferred into recipients, which were infected with LCMV and analyzed at d10 p.i. Analogously to the data from mixed bone marrow chimeras, *Il1rl1-ExAB*[−/−] and *Il1rl1*[−/−] P14 T cells expanded much less than their respective cotransferred WT P14 T-cell populations (Fig. 4n–r). Similarly, albeit less pronounced, *Il1rl1-ExAB*[−/−] Smarta T cells expanded less than cotransferred WT Smarta cells (Fig. 4s–w and Extended Data Fig. 6n–r). Interestingly, analysis of Smarta cells revealed that *ExAB*-deficient, but not WT, Smarta cells used the proximal promoter

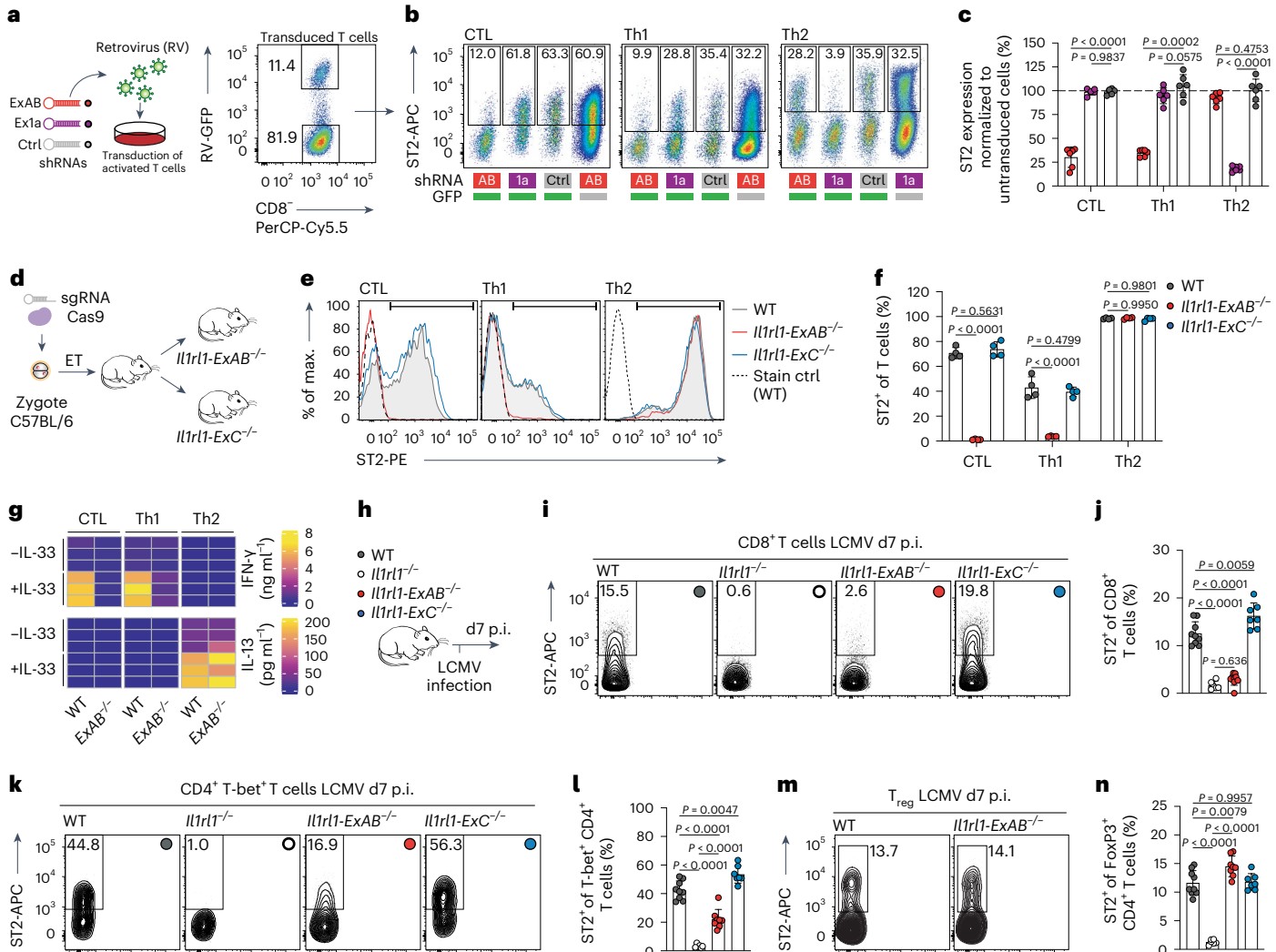

**Fig. 3 | Usage of distinct promoters allows T-cell subset-specific targeting of ST2 expression. a**, Experimental outline and representative FACS plot showing GFP expression by transduced T cells. **b,c**, Representative FACS plots (**b**) and quantification (**c**) of ST2 surface expression by transduced T cells analyzed after 5 days of culture (*n* = 6 cultures pooled from two (CTL) or three (Th1, Th2) independent experiments). **d**, Scheme depicting generation of *Il1rl1-ExAB*⁻/⁻ and *Il1rl1-ExC*⁻/⁻ mice using CRISPR/Cas9 in murine zygotes. **e,f**, Representative histograms (**e**) and quantification (**f**) of ST2 surface expression by WT, *Il1rl1-ExAB*⁻/⁻ or *Il1rl1-ExC*⁻/⁻ T cells (*n* = 4). **g**, IFN-γ and IL-13 secretion by WT or *Il1rl1-ExAB*⁻/⁻ T cells after IL-33 stimulation (*n* = 3). **h–n**, WT, *Il1rl1*⁻/⁻, *Il1rl1-ExAB*⁻/⁻ and

*Il1rl1-ExC*⁻/⁻ mice were infected with LCMV-WE (**h**), and ST2 expression by splenic CTLs (**i** and **j**), Th1 cells (**k** and **l**) and T_reg cells (**m** and **n**) was quantified on d7 p.i. (WT: *n* = 9, *Il1rl1*⁻/⁻: *n* = 6, *Il1rl1-ExAB*⁻/⁻: *n* = 8, *Il1rl1-ExC*⁻/⁻: *n* = 7). In **j**, **l** and **n**, *x* axis ticks represent the four analyzed genotypes. Group allocation of symbols is depicted in **h**. Data represent one (**g**), two (**e** and **f**) or three (**i–n**) independent experiments and are presented as mean ± standard deviation with each dot representing one mouse (**j**, **l** and **n**) or one experiment performed with T cells from individual mice (**c** and **f**). *P* was determined using one-way (**j**, **l** and **n**) or two-way (**c** and **f**) ANOVA with Tukey's post hoc test.

(exon 1b) (Extended Data Fig. 6s–v). In summary, optimal expansion of antiviral T cells critically depends on T-cell-intrinsic activity of the type 1 *Il1rl1* promoter.

### The type 1 *Il1rl1* promoter drives short-lived effector formation

At the peak of the acute response, the antiviral CTL population is heterogeneous and comprises functionally distinct subsets[51]. To delineate the impact of type 1 promoter-driven ST2 expression on CTL differentiation, we sorted activated CD44⁺ CTLs from LCMV-infected WT or *Il1rl1-ExAB*⁻/⁻ mice for combined single-cell gene expression and TCR repertoire analysis. T cells were clustered into six separate populations using nearest neighbor modularity optimization and annotated based on signature gene expression (Fig. 5a,b, Extended Data Fig. 7a,b and Supplementary Table 1). The two dominant clusters showed expression of genes associated with short-lived effector cells (SLECs; *Klrg1*,

*Gzma* and *Id2*) or memory precursor effector cells (MPECs; *Il7r*, *Sell* and *Ccr7*) (Fig. 5c). The third cluster was enriched in CTLs expressing *Pdcd1* (encoding PD-1) and *Lag3*, markers of exhausted CTLs[52], whereas cells of the fourth cluster exhibited higher expression of *Tcf7* (encoding TCF-1) and *Id3*, thus likely presenting stem-like precursors of effector CTLs[53,54]. Lastly, the two remaining clusters were enriched in CTLs expressing higher levels of mitochondrial genes or cell cycle-related markers (*Mki67*, *Top2a*). A cluster-wise comparison between genotypes revealed that type 1 *Il1rl1* promoter disruption affected gene expression in all CTL subsets (Extended Data Fig. 8) but resulted in a particularly pronounced curtailment of SLECs (Fig. 5d). This translated into a 90–95% reduction of SLEC numbers in *Il1rl1-ExAB*⁻/⁻ mice, mirroring the phenotype of *Il1rl1*⁻/⁻ mice (Fig. 5e,f). Further, fewer *Il1rl1-ExAB*⁻/⁻ CTLs exhibited surface expression of the SLEC-associated molecule CXCR3 (Fig. 5g)[55]. Importantly, despite a relative increase in the frequency of MPECs, MPEC counts were slightly decreased (Fig. 5h).

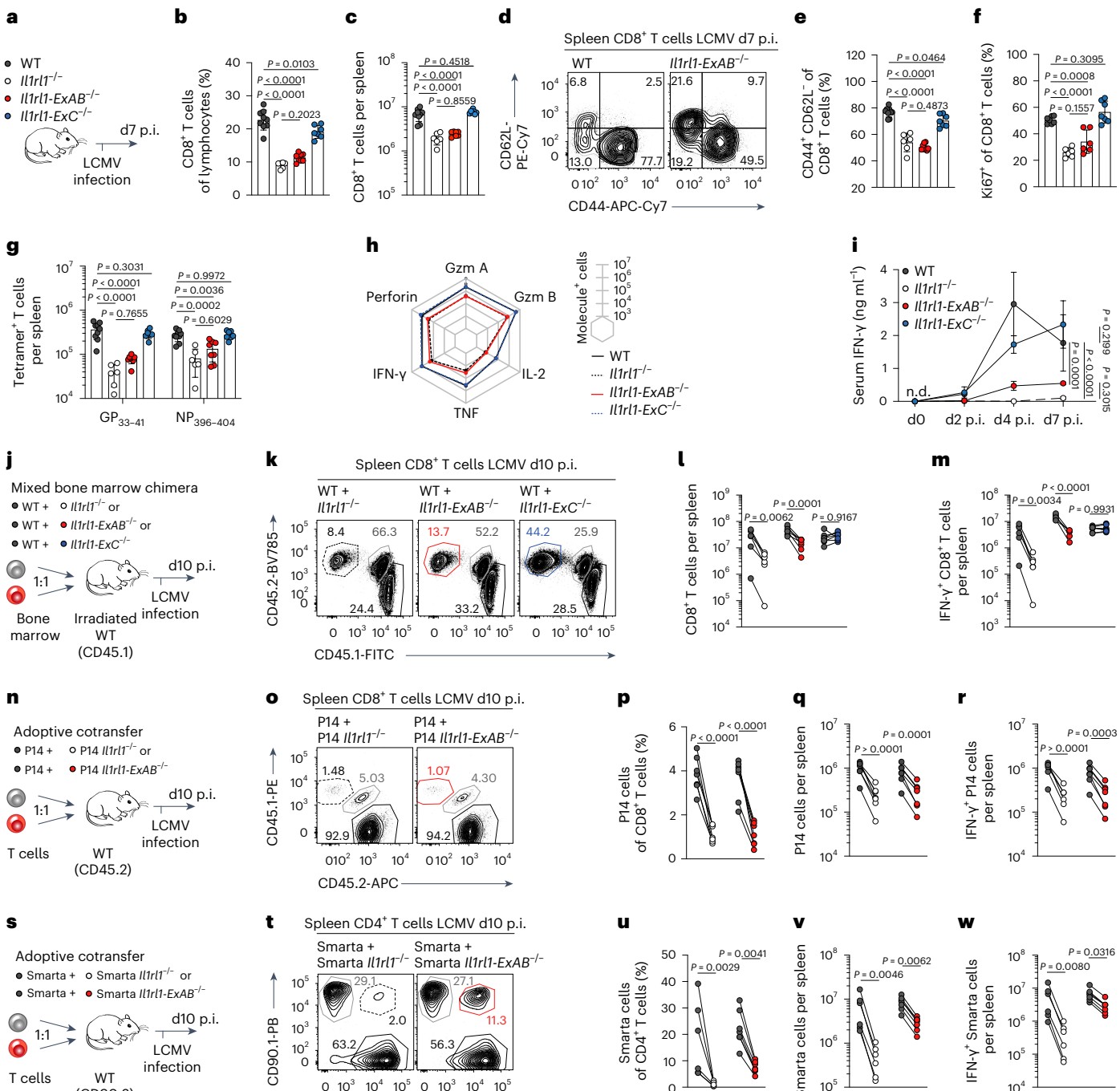

**Fig. 4 | T-cell-intrinsic activity of the type 1 *Il1rl1* promoter is vital for efficient expansion of antiviral T cells. a–h**, WT, *Il1rl1^-/-^*, *Il1rl1-ExAB^-/-^* and *Il1rl1-ExC^-/-^* mice were infected with LCMV-WE and analyzed on d7 p.i. (WT: *n* = 9, *Il1rl1^-/-^*: *n* = 6, *Il1rl1-ExAB^-/-^*: *n* = 8, *Il1rl1-ExC^-/-^*: *n* = 7). **a**, experimental outline. **b** and **c**, frequencies (**b**) and counts (**c**) of CTLs. **d–f**, representative staining (**d**) and frequencies of CD44⁺ CD62L⁻ (**e**) and Ki67⁺ (**f**) CTLs. **g**, counts of LCMV-specific CTLs. **h**, counts of effector molecule⁺ CTLs after restimulation with GP₃₃₋₄₁. **i**, IFN-γ serum levels after infection with LCMV-WE (WT and *Il1rl1^-/-^*: *n* = 4, *Il1rl1-ExAB^-/-^*: *n* = 5, *Il1rl1-ExC^-/-^*: *n* = 3). **j–m**, Irradiated WT recipients (CD45.1⁺) were reconstituted with WT (CD45.1⁺ CD45.2⁺) and *Il1rl1^-/-^*, *Il1rl1-ExAB^-/-^* or *Il1rl1-ExC^-/-^* (CD45.2⁺) bone marrow, infected with LCMV-WE and analyzed on d10 p.i. (WT+*Il1rl1^-/-^*: *n* = 6, WT+*Il1rl1-ExAB^-/-^* and WT+*Il1rl1-ExC^-/-^*: *n* = 7). **j**, experimental outline. **k**, representative FACS plots showing CTL populations. **l** and **m**, cell counts of splenic CTLs (**l**) and IFN-γ⁺ CTLs after restimulation with GP₃₃₋₄₁ (**m**). **n–r**, P14 T cells (CD45.1⁺ CD45.2⁺) were cotransferred with *Il1rl1^-/-^* or *Il1rl1-ExAB^-/-^* P14 T cells (CD45.1⁺) into WT mice (CD45.2⁺). Recipients were infected with LCMV-Cl13 and analyzed at d10 p.i.

(*n* = 7). **n**, experimental outline. **o**, representative FACS plots showing CTL populations. **p** and **q**, frequencies (**p**) and counts (**q**) of splenic P14 cells. **r**, counts of IFN-γ⁺ P14 cells after restimulation with GP₃₃₋₄₁. **s–w**, Smarta cells (CD90.1⁺) were cotransferred with *Il1rl1^-/-^* or *Il1rl1-ExAB^-/-^* Smarta cells (CD90.1⁺ CD90.2⁺) into WT mice (CD90.2⁺). Recipients were infected with LCMV-Cl13 and analyzed at d10 p.i. (*Il1rl1^-/-^* Smarta: *n* = 6, *Il1rl1-ExAB^-/-^* Smarta: *n* = 7). **s**, experimental outline. **t**, representative FACS plots showing CD4⁺ T-cell populations. **u** and **v**, frequencies (**u**) and counts (**v**) of splenic Smarta cells. **w**, counts of IFN-γ⁺ Smarta cells after restimulation with GP₆₄₋₇₉. Data represent one (**i–m**), two (**n–w**) or three (**a–h**) independent experiments and are presented as mean ± standard deviation, with each dot representing one mouse or the arithmetic mean of all replicates (**h** and **i**). *P* was determined using one-way ANOVA (**b**, **c**, **e** and **f**), two-way (**g** and **i**) ANOVA with Tukey's post hoc test or two-way repeated measures ANOVA with Šidák's post hoc test (**l**, **m**, **p–r** and **u–w**). In **l** and **m**, *x* axis ticks represent groups receiving different combinations of donor T cells as depicted in **j**, as well as **p–r** and **u–w** as depicted in **n** and **s**, respectively. n.d., not detectable.

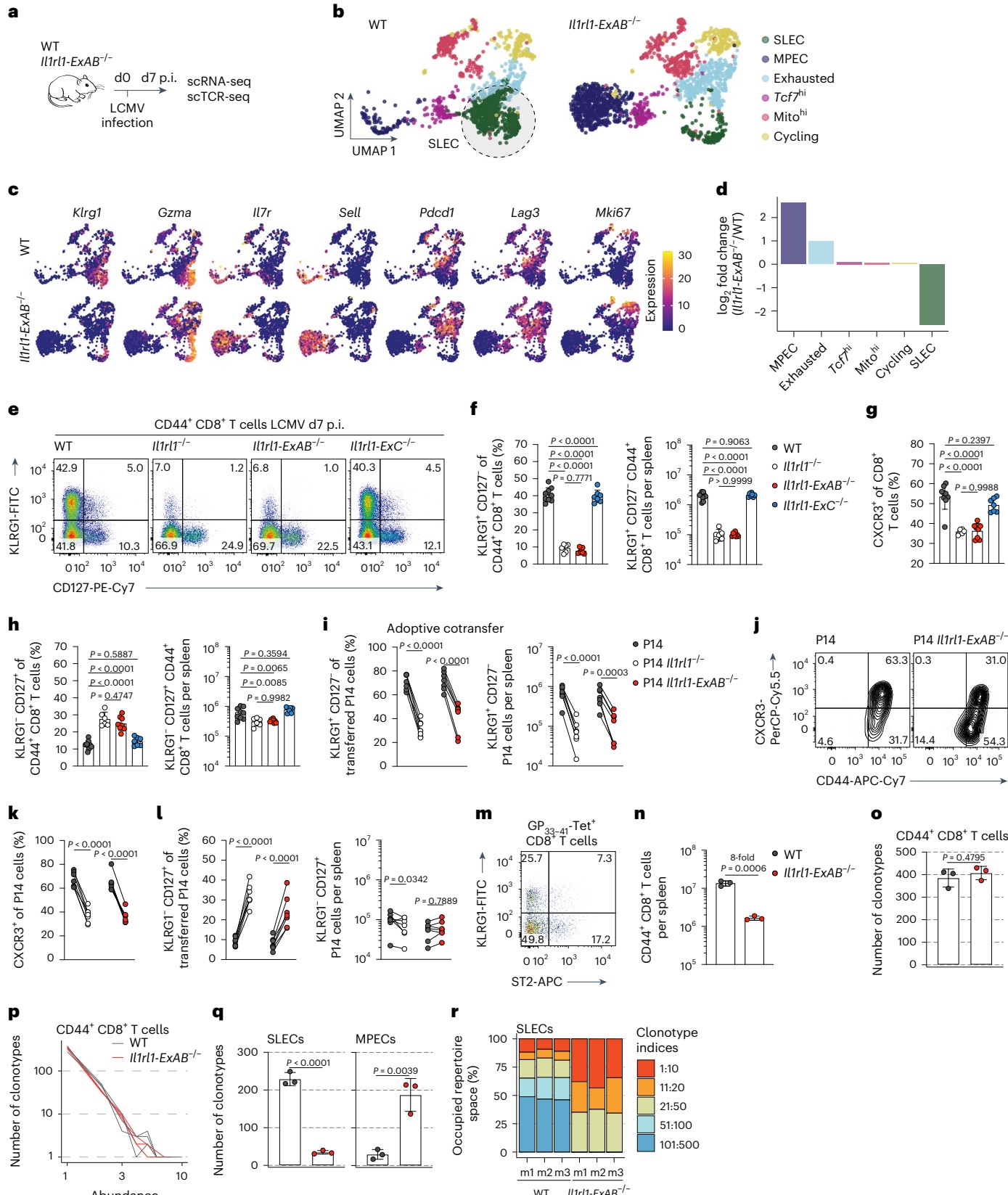

Correspondingly, *Il1rl1-ExAB*−/− CTLs in mixed bone marrow chimeras featured a pronounced defect in SLEC generation and moderately lower MPEC numbers, similar to *Il1rl1*−/− CTLs (Extended Data Fig. 6l,m). In contrast, *Il1rl1-ExC*−/− bone marrow-derived CTLs were as proficient as WT cells in populating the SLEC and MPEC compartments. Lastly, the proportion of *Il1rl1-ExAB*−/− P14 T cells differentiating into SLECs and/ or CXCR3+ cells was reduced as compared to adoptively cotransferred WT P14 cells, whereas MPEC counts were largely unaffected (Fig. 5i–l). In line with these results, analysis of *Il1rl1-ExAB*−/− and *Il1rl1*−/− P14 cells at d30 p.i. revealed a modest decrease in numbers of memory CTLs

**Fig. 5 | Type 1 *Il1rl1* promoter engagement facilitates effector differentiation of CTLs to generate a clonally diverse SLEC population. a–d, n–r** WT and *Il1rl1-ExAB*−/− mice were infected with LCMV-WE, and splenic CD44+ CTLs were analyzed by multiplexed scRNA-seq and scTCR-seq analysis on d7 p.i. (*n* = 3 mice pooled per group). **a**, experimental outline. **b**, Uniform Manifold Approximation and Projection (UMAP) plots colored by cluster type. **c**, UMAP plots showing normalized expression of selected genes in both genotypes. **d**, change in cluster composition of *Il1rl1-ExAB*−/− CTLs relative to WT CTLs. **e**, Representative FACS plots showing KLRG1 and CD127 expression by CD44+ CD8+ T cells in LCMV-WE infected WT, *Il1rl1*−/−, *Il1rl1-ExAB*−/− and *Il1rl1-ExC*−/− mice at d7 p.i. **f**, Frequencies and counts of KLRG1+ CD127− SLECs. **g**, Frequencies of CXCR3+ CTLs. **h**, Frequencies and counts of KLRG1− CD127+ MPECs (**e–h**, WT: *n* = 9, *Il1rl1*−/−: *n* = 6, *Il1rl1-ExAB*−/−: *n* = 8,

*Il1rl1-ExC*−/−: *n* = 7). **i–l**, Frequencies and counts of KLRG1+ CD127− SLECs (**i**), CXCR3+ P14 cells (**j** and **k**) and KLRG1− CD127+ MPECs (**l**) in adoptive cotransfer experiments on d10 p.i. (*n* = 7). **m**, ST2 expression by GP33-41-specific SLECs and MPECs in WT mice at d7 p.i. (*n* = 9). **n**, counts of CD44+ CTLs per spleen. **o**, TCR clonotypes in sequenced CD44+ CTLs. **p**, graph displaying number of TCR clonotypes and their abundance among all analyzed CD44+ CTLs. **q**, TCR clonotypes in SLEC and MPEC clusters. **r**, TCR repertoire occupation in individual WT and *Il1rl1-ExAB*−/− mice. Data represent one (**b–d** and **m–r**) or two (**e–l**) independent experiments and are presented as mean ± standard deviation, with each dot or line representing one mouse (**f–i**, **k**, **l** and **n–q**). *P* was determined using two-tailed *t*-tests (**n** and **q**), one-way ANOVA with Tukey's post hoc test (**f–h**) or two-way repeated measures ANOVA with Šidák's post hoc test (**i**, **k** and **l**).

compared to WT P14 cells (Extended Data Fig. 9a–d). However, both *Il1rl1-ExAB*−/− and *Il1rl1*−/− P14 cells were able to give rise to both effector memory (T_em) as well as central memory cells (T_cm) (Extended Data Fig. 9e–g) and formed tissue-resident memory cells (Extended Data Fig. 9h–k). Thus, a lack of ST2 signaling leads to a generalized impairment in CTL expansion. This was accentuated in the SLEC compartment during the acute antiviral response and extended in part to the population of circulating memory CTLs, whereas formation of tissue-resident memory cells appeared less ST2 dependent.

Next, we asked whether ST2 expression by the type 1 promoter drives the selective proliferation of SLEC-differentiated T-cell clones or whether it enforces the differentiation of precursors into SLECs. ST2 is found on both KLRG1+ and KLRG1− CTLs of WT mice (Fig. 5m), suggesting IL-33 signaling can occur prior to SLEC differentiation. Further, we integrated single-cell gene expression data with a TCR repertoire analysis. Despite an eight-fold difference in CD44+ CTL counts per spleen (Fig. 5n), equivalent numbers of clonotypes were identified in *Il1rl1-ExAB*−/− and WT mice when equal numbers of CD44+ CTLs were compared (Fig. 5o). This finding suggested that during the acute phase of infection, IL-33 expands activated T cells in a clonotype-unselective manner, which does not substantially alter the TCR diversity amongst the most abundant clonotypes. Of note, the majority of clonotypes identified were represented less than three times per mouse and no clonotype was found more often than seven times (Fig. 5p). In line with previous reports, this indicated that TCR diversity within the CTL population was high during the acute phase of infection[56]. By consequence, the severe reduction in SLEC numbers in *Il1rl1-ExAB*−/− mice resulted in reduced SLEC clonotype numbers (Fig. 5q,r). Taken together, without type 1 promoter-driven ST2 expression, most CTL clones achieve basal activation, but fail to develop into fully differentiated SLECs. Thus, type 1 promoter-driven ST2 expression is vital to establish a numerically relevant and clonally diverse population of short-lived antiviral effector CTLs.

## RNA profiling indicates a TCR-cooperative role of IL-33

To gain mechanistic insight on how IL-33 signaling modulates T-cell activation and differentiation, we performed a comprehensive analysis of early IL-33 target genes. To this end, naive T cells were differentiated into CTLs, Th1 or Th2 cells, followed by a resting period without antigenic stimulation. Because ST2 signaling is subject to negative feedback mechanisms and oxidation of IL-33 rapidly reduces its activity[57,58], gene expression was analyzed before (0 h) and 2 h after treatment with or without IL-33 (Fig. 6a). Short-term stimulation with IL-33 had a profound effect on the transcriptome of all subsets, and it strongly induced or, in fewer cases, reduced expression of target genes rather than preventing a loss or gain of transcription (Fig. 6b). Whereas many differentially regulated genes were shared between subsets, others were regulated in a lineage-specific manner (Fig. 6c). Gene Ontology-enrichment analysis revealed a broad role of IL-33-responsive genes in T-cell activation, proliferation and differentiation (Fig. 6d). Importantly, IL-33 stimulation of CTLs amplified expression of *Tbx21*, *Zeb2* and *Prdm1*, encoding transcription factors critical for SLEC differentiation[59–61]

(Fig. 6b,e). Across the three T-cell subsets we observed a prominent upregulation of genes frequently used as indicators of recent TCR activation (*Nr4a1*, *Cd69* and *Batf*) (Fig. 6b,e)[62–64]. Coherently, gene set enrichment analysis showed a significant overlap between IL-33- and TCR-downstream signaling in CTLs (Fig. 6f and Supplementary Table 2). This finding suggested that IL-33 may support TCR stimulation to promote potent antiviral T-cell responses.

To further investigate the interplay between ST2 and TCR signaling strength, we made use of a genetically engineered LCM virus that differs from the WT counterpart only by a GP-A39C mutation, rendering its GP33 epitope a weak P14 TCR agonist[65]. P14 and *Il1rl1*−/− P14 cells were cotransferred into WT recipients, which were subsequently infected with LCMV expressing either the high- or the low-affinity GP33 variant (Fig. 6g). We found that ST2-sufficient and ST2-deficient P14 cells expanded less when primed with the low-affinity ligand (Fig. 6h). Further, P14 cells depended on ST2 for optimal expansion and effector differentiation, irrespective of the TCR stimulation strength (Fig. 6i). Interestingly, the response of WT P14 cells responding to low-affinity virus was comparable to or exceeded the one of high-affinity ligand-primed *Il1rl1*−/− P14 cells in terms of total and effector cell progeny, respectively (Fig. 6h,i). This finding suggested that ST2 signals can help reaching effector T-cell responses of critical size even when confronted with low-affinity ligands. Lastly, in comparison to WT P14 cells, *Il1rl1*−/− P14 cells yielded slightly fewer MPECs, irrespective of the TCR stimulation strength (Fig. 6j). Of note, the impairment in T-cell expansion and effector differentiation between mice infected with high- or low-affinity GP33-expressing LCMV were unlikely due to any potential differences in inflammation or IL-33 release, as the endogenous NP396-specific T-cell responses to the two LCMV variants were indistinguishable (Fig. 6k).

In summary, these data demonstrate that IL-33–ST2 signaling provides a strong costimulatory signal that can act cooperatively with TCR signaling to promote the expansion and effector cell differentiation of antiviral CTLs.

## Discussion

IL-33 has long been recognized as a type 2 immunity-related cytokine[2,3,6,7,12,31,32]. Over the past decade its important role in promoting type 1 immunity has become widely accepted yet remains mechanistically less well understood, particularly due to a lack of understanding how ST2 expression is regulated in these cells[31,32]. Here, we studied the transcriptional regulation of ST2 expression in antiviral T cells and discovered a dedicated type 1 immunity-restricted promoter located ~40 kb upstream of the curated *Il1rl1* gene in mice and humans. This type 1 promoter drives ST2 expression by CTLs and Th1 cells in vitro and in viral infections in vivo. As opposed to the previously described type 2 promoter, which is regulated by GATA-3 (ref. 33) and is utilized by type 2 immune cells and T_reg cells[21], the identified promoter is controlled by the type 1 immunity-associated transcription factors T-bet and STAT4 and is subject to epigenetic remodeling during type 1 T-cell differentiation. Thus, we provide evidence for a dedicated regulatory genetic element to control ST2 expression selectively in type 1-polarized T cells, as well as NKT and NK cells.

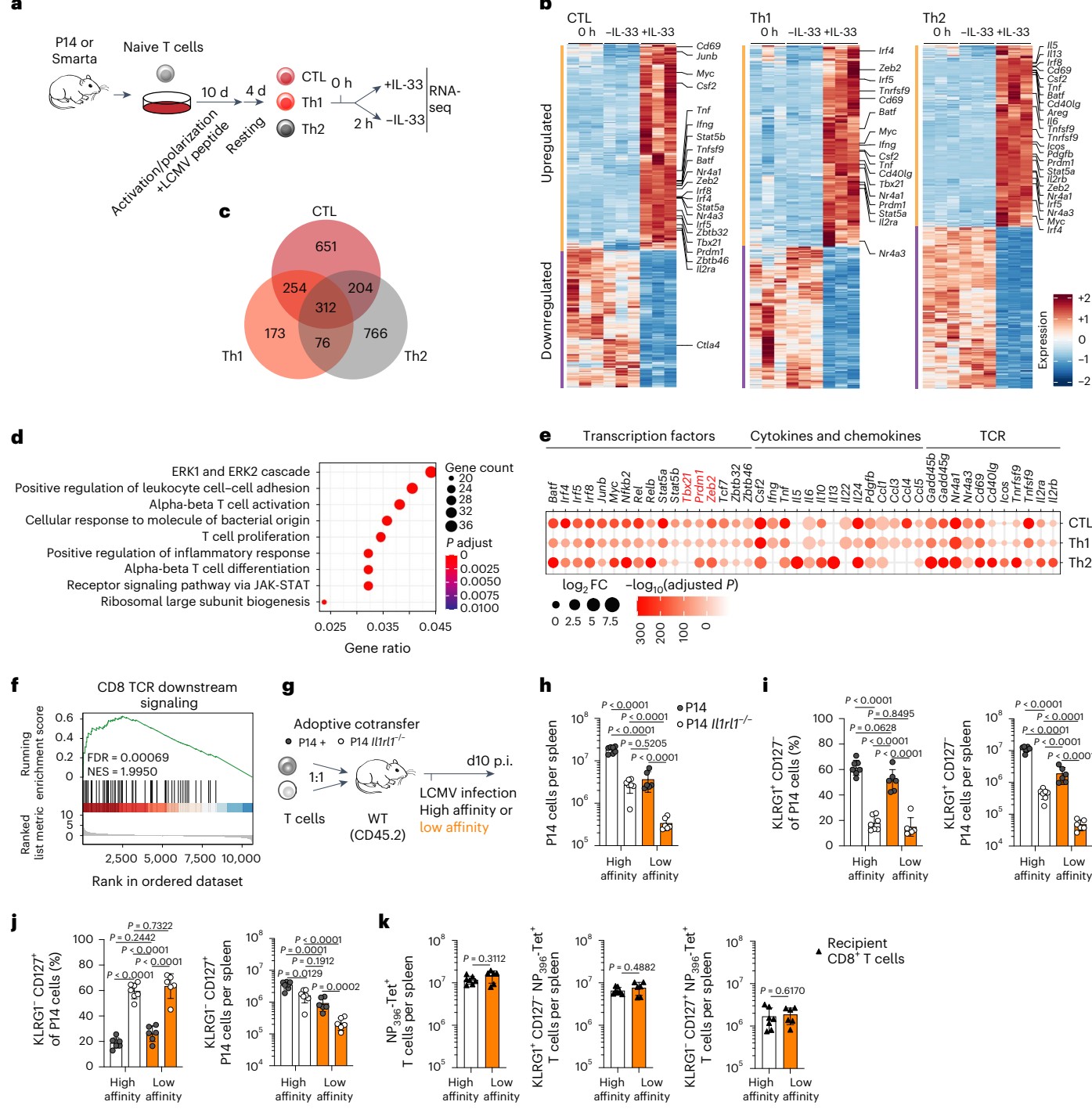

**Fig. 6 | Transcriptional profiling indicates broad costimulatory and TCR-cooperative functions of IL-33–ST2 signaling in T cells. a–f,** Differentiated CTLs, Th1 cells and Th2 cells were stimulated with IL-33 or left untreated for 2 h and subjected to RNA-seq analysis (n = 3 independent cultures per subset). **a**, experimental outline. **b**, heatmaps depicting differentially expressed genes in each T-cell subset (log2 fold change > 1.0; P adjusted < 0.01). **c–f,** comparison of IL-33-stimulated and untreated T cells (2 h timepoint). **c**, Venn diagram illustrating the overlap in differentially regulated genes between CTLs, Th1 cells and Th2 cells. **d**, gene ontology biological process overrepresentation analysis of IL-33-induced genes shared among all subsets. **e**, expression of selected transcription factors, cytokines and chemokines, and TCR-regulated genes in each T-cell subset. **f**, gene set enrichment analysis of TCR-downstream genes in IL-33-stimulated versus unstimulated CTLs. **g–k,** P14 T cells (CD45.1⁺ CD45.2⁺)

were adoptively cotransferred with *Il1rl1*⁻/⁻ P14 T cells (CD45.1⁺) into WT mice (CD45.2⁺). Recipients were infected with high- or low-affinity GP33-expressing LCMV-Cl13 and analyzed at d10 p.i. (high-affinity group: n = 7, low-affinity group: n = 6). **g**, experimental outline. **h**, counts of recovered P14 and *Il1rl1*⁻/⁻ P14 T cells. **i**, frequencies and counts of KLRG1⁺ CD127⁻ P14 cells. **j**, frequencies and counts of KLRG1⁻ CD127⁺ P14 cells. **k**, counts of endogenous NP396-404-specific CTLs and KLRG1⁺ CD127⁻ or KLRG1⁻ CD127⁺ NP396-404-specific CTLs. Data represent one (**a–f**) or two (**g–k**) independent experiments. Data are presented as mean ± standard deviation, with each dot representing one mouse (**h–k**). P value was determined using one-way ANOVA with Tukey's post hoc test (**h–j**), two-tailed t-tests (**k**), two-sided Wald test with BH correction (**b** and **e**), one-sided hypergeometric test with BH correction (**d**) and two-sided permutation test with BH correction (**f**).

Although the *Il1rl1* gene has been studied intensively[8,35,66], the type 1 promoter has remained unrecognized, likely because it is only transiently active, often resulting in a low abundance of ST2-coding transcripts[24,27]. The latter renders it difficult to obtain adequate read coverage for a clear definition of exon structures by commonly used RNA-seq techniques[67]—a challenge we approached by analyzing T cells that express a high amount of *Il1rl1* transcripts. Subsequently, we have validated the crucial role of this regulatory element in vivo, and by analyzing human T cells have extended the concept to our species.

Above all, our finding was surprising, as to the best of our knowledge no other gene has been identified to date, for which type 1 and type 2 immune cells exhibit a similarly distinct lineage-specific promoter usage. Consistently, our own attempts at identifying genes with an analogous promoter usage were unsuccessful. We acknowledge that technical limitations might have prevented us from identifying such genes. Still, our results suggest that this is not a common feature but represents a fairly unique mechanism to spatiotemporally orchestrate ST2 expression.

IL-33 is an exceptionally potent alarmin, which can act as a pro- or anti-inflammatory cytokine, depending on the local composition of immune cells and their responsiveness to IL-33 (ref. 32). Likely due to its potential to cause severe inflammation, IL-33 responsiveness requires stringent regulation. Transcription from the type 1 immunity-restricted promoter enables transient ST2 expression by CTLs and Th1 cells in response to inflammatory stimuli[23,24,27], which may serve to prevent continuous activation of cells with a high tissue-destructive potential. In contrast, constitutive type 2 promoter-driven ST2 expression on T$_{reg}$ cells and ILC2s allows for rapid anti-inflammatory responses to tissue damage[19–21,32].

Importantly, this dual mode of action constitutes a major hurdle for the therapeutic modulation of IL-33–ST2 signaling[32,68]. We here demonstrate that the usage of distinct promoters offers opportunities for a T-cell subset-specific targeting of ST2 expression. *Il1rl1-ExAB*$^{-/-}$ mice exhibit a type 1 immunity-restricted impairment of ST2 expression and display curtailed CTL and Th1 responses against LCMV, whereas ST2 expression by T$_{reg}$ cells and type 2 immune cells was fully preserved. Of note, in CTLs ST2 expression was almost exclusively dependent on the type 1 promoter, whereas some *Il1rl1-ExAB*$^{-/-}$ Th1 cells could compensate for the defect by engaging the proximal *Il1rl1* promoter. Nevertheless, the type 1 promoter was critical for optimal expansion of antiviral Th1 cells. T-cell subset-specific targeting approaches could be of interest to modulate IL-33 responses in inflammatory diseases. For instance, IL-33 administration was shown to drive T$_{reg}$ expansion in the context of GVHD, promoting tolerance induction and disease amelioration[69–71]. However, IL-33 also augments type 1 alloimmunity by acting as a costimulatory molecule for donor CTLs and Th1 cells[28,29]. A targeted disruption of ST2 selectively on type 1 immune cells might minimize the pathological response during GVHD, whereas the protective effects of IL-33 should remain preserved.

Our study provides insight into the role of type 1 promoter-driven ST2 expression and IL-33 signaling in CTL differentiation. scRNA-seq analysis of antiviral CTLs revealed that *Il1rl1-ExAB*$^{-/-}$ mice display a pronounced reduction in SLECs. Moreover, clonotype diversity in the SLEC population was high in WT mice and diminished proportionally to cell counts in *Il1rl1-ExAB*$^{-/-}$ mice. This suggests that IL-33 can foster the transition of an activated CTL into a cell with potent effector functions rather than selectively expanding a pool of predifferentiated SLECs. Of note, although the effect was strongly magnified in the SLEC compartment, *Il1rl1*$^{-/-}$ as well as *Il1rl1-ExAB*$^{-/-}$ CTLs showed a generalized reduction in expansion that in most instances also negatively affected MPECs. Consequently, this impairment in primary expansion likely accounts for the lower numbers of circulating memory CTLs 1 month after infection. The formation of tissue-resident memory cells appeared less affected. This finding might suggest a particular importance of IL-33 signals for the generation of antiviral effector CTLs but only to a lower extent for tissue-resident memory CTLs. However, further work is needed to thoroughly test this hypothesis.

Terminal differentiation has been associated with STAT4 signaling and with high levels of T-bet, Blimp-1 and Zeb2 (refs. 59–61). Likewise, the activity of the type 1 *Il1rl1* promoter is positively regulated by STAT4 and T-bet. Interestingly, RNA-seq of IL-33 target genes in CTLs demonstrated an induction of *Tbx21* (T-bet), *Prdm1* (Blimp-1) and *Zeb2* expression, thus inferring a positive feedback loop that further reinforces ST2 expression and effector differentiation via T-bet. Besides these cell-intrinsic factors, TCR signaling strength is linked to acquisition of effector properties[72]. Our data show that IL-33 stimulation of CTLs strongly induces the transcription of several TCR-dependent genes. Moreover, IL-33 signals can restore the otherwise suboptimal expansion and effector differentiation of CTLs in response to a low-affinity antigenic peptide. Recent studies demonstrated a loss of IL-33 in lymphoid organs early after LCMV infection, suggesting substantial release during T-cell priming[26]. Together, this implies that ST2 signaling might act in conjunction with TCR signaling to achieve above-threshold activation required for fully functional effector differentiation.

In summary, we here uncover lineage-specific promoter usage as molecular mechanism governing disparate expression patterns of ST2 in distinct T-cell subsets. Using newly generated knockout mice, we demonstrate that the type 1 immunity-restricted *Il1rl1* promoter is essential for fully functional antiviral T-cell responses and critical for the formation of a clonally diverse population of effector CTLs. These findings open new avenues for the modulation and exploitation of IL-33 signaling in type 1 immunity-mediated inflammatory diseases and T-cell-based cancer immunotherapy, respectively.

## Online content

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

## Methods

### Mice

C57BL/6 J mice (WT), LCMV-TCR[tg] P14 (ref. [73]) and Smarta[74] mice expressing the congenic markers CD45.1 or CD90.1, respectively, *Il1rl1*[−/−] (ref. [12]), *Il1rl1-ExAB*[−/−], *Il1rl1-ExC*[−/−], *Stat4*[−/−] (ref. [75]), *Tbx21*[−/−] (ref. [76]), *Il1rl1-ExAB*[−/−] Smarta, *Stat4*[−/−] Smarta, *Tbx21*[−/−] Smarta, *Il1rl1*[−/−] P14, *Il1rl1-ExAB*[−/−] P14 and *Tcrbd*[−/−] (ref. [77]) mice were bred under specific-pathogen-free conditions in approved animal-care facilities at the Research Institute for Experimental Medicine of the Charité – Universitätsmedizin Berlin or at the Laboratory Animal Facility of the ETH Zürich (ETH Phenomics Center). Mice were housed in individually ventilated cages with a 12 h light/dark cycle at an ambient temperature of 21 °C and 45% to 65% relative humidity. Mice had ad libitum access to drinking water and chow. Both, male and female mice between 8 and 26 weeks of age were used for experiments. For LCMV infections, experimental groups were age and sex matched. Mice used for scRNA-seq analyses were cohoused for 4 weeks before infection. Animal experiments were performed in accordance with the German or Swiss law for animal protection and were approved by the respective governmental authority (Landesamt für Gesundheit und Soziales Berlin and the Cantonal Veterinary Office of the Canton of Basel; T0058/08, G0111/17, G0206/17, G0245/19).

### Generation of *Il1rl1-ExAB*[−/−] and *Il1rl1-ExC*[−/−] mice

*Il1rl1-ExAB*[−/−] and *Il1rl1-ExC*[−/−] mice were generated in the Transgenics Core Facility of the Max Delbrück Centrum Berlin using established protocols[78]. In brief, gRNA sequences with minimal predicted off-target effects were identified using the web-based tool CRISPOR[79]. Zygotes were collected from C57BL/6 J mice (Charles River), microinjected with synthetic gRNAs (Integrated DNA Technologies) and recombinant Cas9 protein (Integrated DNA Technologies) and subsequently transferred into pseudo-pregnant C57BL/6 J mice. Resulting F0 offspring mice were screened for successful deletion by PCR amplification of WT or knockout alleles. gRNA and PCR primer sequences are listed in Supplementary Table 4.

### Lymphocyte isolation

To isolate lymphocytes, spleens were mechanically disrupted and filtered through 70-µm strainers. Erythrocytes were lysed by 35 min of incubation in erythrocyte lysis buffer (10 mM $KHCO_3$, 155 mM $NH_4Cl$, 0.1 mM EDTA, pH 7.5). Livers were collected in PBS/BSA, meshed and centrifuged at 30 *g* for 2 min to remove debris. Supernatants were subjected to Histopaque density centrifugation (1.083 g ml[−1], Sigma-Aldrich) and lymphocytes were collected at the gradient interphase. To stain ILC2s, lungs were cut into small pieces and digested with Collagenase D (0.1 U ml[−1]) in RPMI1640 (supplemented with 10% fetal calf serum (FCS) and 15 mM HEPES) for 1 h at 37 °C. Afterwards, lymphocytes were isolated by Histopaque density centrifugation (1.083 g ml[−1], Sigma-Aldrich). To isolate peritoneal cavity cells, 5 ml cold PBS was injected into the peritoneal cavity of euthanized mice. After a brief massage of the peritoneum, cell-containing liquid was collected and subjected to Histopaque density centrifugation (1.083 g ml[−1], Sigma-Aldrich). For analysis of tissue-resident memory T cells, lungs, kidneys and salivary glands were cut into pieces and digested in RPMI1640 + GlutaMax I (Thermo Scientific) medium containing FCS (5% v/v, Thermo Scientific), $MgCl_2$ (2 µM, Carl Roth), $CaCl_2$ (2 µM, Carl Roth) and collagenase type I (100 U ml[−1], Gibco) at 37 °C for 45 min. Subsequently, tissue was further disrupted using a GentleMACS Dissociator (setting m_Spleen_01.01). Cells were filtered through 70-µm strainers, subjected to erythrocyte lysis and analyzed.

### Flow cytometry

Surface stainings of purified lymphocytes were performed using different combinations of antibodies diluted in PBS. A list of antibodies and dilutions used in this study is provided in Supplementary Table 3.

Unspecific staining was minimized by blocking with rat immunoglobulin G (Jackson ImmunoResearch) and anti-mouse CD16/32 (2.4G2, DRFZ inhouse production) prior to staining. Dead cells were labeled using Zombie Aqua or Zombie NIR fixable live/dead staining reagents (BioLegend) or by adding propidium iodide (PI) prior to acquisition. For detection of ST2 on murine T cells, lymphocytes were first stained with digoxigenin-conjugated antibody against ST2 (DJ8), followed by a secondary staining with PE- or APC-conjugated anti-digoxigenin Fab fragments (Roche). Further, stainings were enhanced by two rounds of PE- or APC-FASER amplification (Miltenyi Biotec). To identify LCMV-specific T cells, lymphocytes were stained with LCMV GP$_{33\text{-}41}$ or NP$_{396\text{-}404}$ peptide-loaded MHC class I (H2-Db) tetramers (PE or APC conjugated, respectively) for 30 min at 37 °C. For detection of transcription factors or Ki67 expression, surface-stained cells were fixed and stained using the FoxP3 staining buffer set (Thermo Scientific). Briefly, cells were fixed with 1x fixation/permeabilization reagent for 30 min at 4 °C and washed with permeabilization buffer. Subsequently, cells were stained with antibodies diluted in permeabilization buffer for 30 min at 4 °C.

For flow-cytometric detection of cytokines, lymphocytes were restimulated with phorbol myristate acetate (5 ng ml[−1], Sigma-Aldrich) and ionomycin (5 µg ml[−1], Sigma-Aldrich), recombinant LCMV GP$_{33\text{-}41}$ (1 µg ml[−1], Charité Berlin) or LCMV GP$_{64\text{-}79}$ (1 µg ml[−1], Charité Berlin) for 4 h at 37 °C. After 35 min, brefeldin A (5 µg ml[−1], Sigma-Aldrich) was added. Restimulated cells were labeled with surface antibodies and fixable live/dead staining reagents, followed by fixation in 2% paraformaldehyde for 10 min at room temperature. Intracellular cytokines were stained with antibodies diluted in PBS containing 0.05% saponin (Sigma-Aldrich) for 30 min at 4 °C and washed before acquisition. Cells were acquired using Canto II or LSRFortessa flow-cytometers (BD) with Diva software (BD). Sorting was performed on Aria and Aria II devices (BD). Cell numbers were determined using MACSQuant (Miltenyi Biotec) or ImmunoSpot (CTL) analyzers. Analyses were performed using FlowJo (v.10.7.1).

### Viruses and LCMV infection

LCMV-WE and LCMV-Cl13 strains were propagated on L929 or BHK-21 cells, respectively. Viral titers in stock solutions were determined by immunofocus assay on MC57G cells as described before[80]. In brief, MC57G cells were plated with virus stock dilutions and overlaid with 2% methylcellulose. After 48 h at 37 °C, the confluent monolayer of cells was fixed with 4% formaldehyde, permeabilized with Triton X-100 (1%, v/v) and stained with antibodies against LCMV nucleoprotein. After a secondary staining step with peroxidase-conjugated anti-rat immunoglobulin G antibody, foci were developed by 20-min incubation with OPD substrate (Sigma-Aldrich). Mice were infected intravenously (i.v.) with either 200 plaque-forming units (PFU) of LCMV-WE (mixed bone marrow chimera experiments), 200 PFU LCMV-Cl13 (adoptive transfer experiments, LCMV-Cl13 WT or C6 variant where indicated) or $2 \times 10^6$ PFU of LCMV-WE in minimal essential medium (Thermo Scientific).

### Adoptive T-cell transfers

For adoptive transfer experiments, TCR-transgenic T cells expressing CD45.1 or CD90.1 were enriched in a negative selection approach. Splenocytes of donor mice were stained with biotinylated antibodies against CD11b, CD11c, CD19, CD25, Gr-1, NK1.1, CXCR3 and CD8a (for isolation of Smarta T cells) or CD4 (for isolation of P14 T cells) followed by incubation with anti-biotin microbeads (Miltenyi Biotec). Subsequently, labeled cells were depleted by magnetic activated cell sorting (MACS) using LS columns (Miltenyi Biotec). $5 \times 10^4$ T cells (single transfer experiments), $1 \times 10^3$ P14 T cells or $1 \times 10^4$ Smarta T cells (cotransfer experiments) were transferred i.v. into C57BL/6 J mice. For analysis of memory T cells at d30 p.i., $2.5 \times 10^4$ P14 cells were transferred. Recipients were infected 1–2 days after transfer and analyzed at indicated timepoints.

## In vivo labeling of T cells

To distinguish between tissue-resident and intravascular T cells, mice were injected i.v. with 3 µg PE-conjugated CD90.2 antibody (30-H12, BioLegend) and sacrificed 3 min after injection.

## Mixed bone marrow chimeras

To generate mixed bone marrow chimeras, CD45.1$^{+/+}$ WT recipients were lethally irradiated (two doses of 5.5 Gy given in a 6-h interval). One day later, recipients were reconstituted with a 1:1 mixture of CD45.1$^{+/-}$ WT and CD45.2$^{+/+}$ knockout bone marrow cells and splenocytes. After 8 weeks of hematopoietic reconstitution, CTL frequencies of respective donor populations were determined in blood, and mice were infected with LCMV-WE (200 PFU i.v.). Data were analyzed on d10 p.i. and normalized to CTL frequencies before infection.

## Legendplex cytometric bead assay

To assess cytokine production by T cells in response to IL-33, 5 × 10$^5$ T cells were stimulated in 48-well plates with IL-33 (R&D, 10 ng ml$^{-1}$) for 24 h at 37 °C. Afterwards, individual wells were harvested and centrifuged for 5 min at 350 g. To obtain serum, blood of individual mice was collected using yellow microtainers (BD). Serum and cell-free supernatant were frozen at −80 °C until analysis. Cytokine content was measured using LEGENDplex bead-based immunoassays (BioLegend) according to manufacturer's instructions and acquired at a Canto II flow-cytometer (BD). Cytokine concentration was extrapolated from standard titrations.

## Mouse T-cell cultures

Naive T cells from spleens of indicated mice were preenriched by staining with biotinylated antibodies against CD8a or CD4, followed by incubation with anti-biotin microbeads (Miltenyi Biotec) and subsequent separation by MACS using LS columns (Miltenyi Biotec). Following enrichment, naive (CD62L$^+$ CD44$^-$ CD25$^-$ CXCR3$^-$) CD8$^+$ or CD4$^+$ T cells were flow-cytometrically sorted and differentiated in the presence of irradiated Tcrbd$^{-/-}$ splenocytes and antibodies against CD3ε and CD28 (2.5 µg ml$^{-1}$ each). When naive T cells were isolated from LCMV-TCR$^{tg}$ mice, cognate LCMV GP$_{33-41}$ (P14 mice) or GP$_{64-79}$ peptide (Smarta mice, both 1 µg ml$^{-1}$) were added instead. T cells were cultivated in RPMI1640 + GlutaMax I (Thermo Scientific) medium supplemented with FCS (10% v/v, Thermo Scientific), penicillin (100 U ml$^{-1}$, Thermo Scientific), streptomycin (100 µg ml$^{-1}$, Thermo Scientific), gentamycin (10 µg ml$^{-1}$, Thermo Scientific) and β-mercaptoethanol (50 ng ml$^{-1}$, Sigma-Aldrich). For CTL and Th1 differentiation, IL-12 (5 ng ml$^{-1}$), IL-2 (5 ng ml$^{-1}$, all Miltenyi Biotec) and anti-IL-4 (11B11, 10 µg ml$^{-1}$, DRFZ inhouse production) were added. For Th2 differentiation, IL-4 (5 ng ml$^{-1}$), IL-2 (5 ng ml$^{-1}$, all Miltenyi Biotec), anti-IL-12 (C18.2, 10 µg ml$^{-1}$) and anti-IFN-γ (XMG1.2, 10 µg ml$^{-1}$, all DRFZ inhouse production) were added. T cells were split after 2–3 days of culture in a 1:3 ratio with fresh medium containing IL-2 (5 ng ml$^{-1}$), harvested at day 5 of culture using Histopaque density centrifugation (1.083 g ml$^{-1}$, Sigma-Aldrich) and cultivated for additional 5 days in identical culture conditions.

## Mouse NKT cell and NK cell cultures

Murine NKT cells were preenriched by incubating thymocytes with anti-CD8 and anti-CD62L microbeads (Miltenyi Biotec) followed by subsequent MACS separation using LS columns (Miltenyi Biotec). Enriched cells were stained with PE-conjugated, α-galactosylceramide (α-GalCer)-loaded CD1d tetramers (MBL) and antibodies against TCRβ, CD19 and CD8. CD1d-Tet$^+$ TCRβ$^+$ CD19$^-$ CD8$^-$ NKT cells were flow-cytometrically sorted and activated in 96-well plates precoated with antibodies against CD3ε and CD28 (2.5 µg ml$^{-1}$ each). NKT cells were cultivated in CTL/Th1 culture medium as described above. After 2 days of stimulation, NKT cells were transferred to uncoated wells and split in a 1:3 ratio with fresh medium containing IL-2 (5 ng ml$^{-1}$). Cells were analyzed at day 6 of culture.

To isolate murine NK cells, splenocytes were stained with biotinylated antibodies against CD8, CD4 and B220, followed by incubation with anti-biotin microbeads (Miltenyi Biotec) and subsequent separation by MACS using LS columns (Miltenyi Biotec). CD8-, CD4- and B220-depleted splenocytes were then stained with antibodies against NKp46, TCRβ and streptavidin PE. NKp46$^+$ TCRβ$^-$ CD8$^-$ CD4$^-$ NK cells were flow-cytometrically sorted and activated in RPMI1640 + GlutaMax I (Thermo Scientific) medium supplemented with FCS (10% v/v, Thermo Scientific), penicillin (100 U ml$^{-1}$, Thermo Scientific), streptomycin (100 µg ml$^{-1}$, Thermo Scientific), gentamycin (10 µg ml$^{-1}$, Thermo Scientific), β-mercaptoethanol (50 ng ml$^{-1}$, Sigma-Aldrich), IL-15 (10 ng ml$^{-1}$), IL-12 (10 ng ml$^{-1}$) and IL-33 (10 ng ml$^{-1}$). NK cells were analyzed after 2 days of culture.

## Retroviral transduction of T cells

For cloning of shRNA expression vectors, sense- and antisense-shRNA sequences were ordered as phosphorylated oligos with a 5' SalI restriction overhang (Eurofins Genomics) and annealed by subjecting equimolar amounts of oligos diluted in oligo annealing buffer (100 mM Tris-HCl, 1 M NaCl and 10 mM EDTA, pH 7.5) to a decreasing temperature gradient (95 °C to 25 °C with 1 °C min$^{-1}$). Oligo sequences are provided in Supplementary Table 4. PQCXIX-GFP target vector[81] was digested by SalI and HpaI restriction enzymes (Thermo Scientific) and dephosphorylated with FastAP alkaline phosphatase (Thermo Scientific). Annealed oligos were ligated using T4 Ligase according to standard protocols (NEB). Heat-inactivated ligation reactions were directly used for heat-shock transformation into Oneshot TOP10 chemically competent Escherichia coli (Thermo Scientific). Single transformed bacterial clones were selected on LB-agar plates (MP Biomedicals) containing ampicillin (100 µg ml$^{-1}$, Sigma-Aldrich), and plasmid DNA was prepared using QIAprep Spin Plasmid Maxi or Midi kits (Qiagen). Correct plasmid sequences were verified by Sanger-sequencing (Eurofins Genomics). Virus particles were generated by co-transfection of HEK293T cells with shRNA-containing vectors and packaging plasmids pCGP and pECO[82] using Transporter 5 transfection reagent (Polysciences). For retroviral transduction, mouse T cells were activated in the presence of irradiated APCs and antibodies against CD3ε and CD28 described above. 36–48 h after plating, culture medium was temporarily replaced with virus-containing supernatant, polybrene (8 µg ml$^{-1}$, Sigma-Aldrich) was added and plates were centrifuged for 90 min at 450 g at room temperature. T cells were incubated at 37 °C for 6–8 h. Afterwards, viral supernatant was replaced with conditioned cell culture medium and cells were split in a 1:3 ratio with fresh IL-2-containing medium. Transduced T cells were analyzed between day 5 and day 7 of culture.

## Human T-cell cultures

Human peripheral blood was obtained from the German Red Cross (DRK Berlin; ethics approval EA1/149/12) with consent from donors. For isolation of T cells, blood was first subjected to Ficoll-Paque PLUS density centrifugation (1.077 g ml$^{-1}$, Cytiva). Interphases were collected, stained with anti-CD4 microbeads (Miltenyi Biotec) and separated using LS columns (Miltenyi Biotec). CD4$^+$ T-cell-depleted fractions were used for a MACS enrichment of CD8$^+$ T cells using anti-CD8a microbeads (Miltenyi Biotec). CD4-enriched fractions were stained with antibodies against human CD4, CXCR3 and CRTH2 for 15 min at 4 °C followed by a secondary staining with streptavidin PE. CD8-enriched fractions were stained with antibodies against human CD8, CD56, CD62L and CD45RA. In vivo-differentiated Th1 cells were sorted as CD4$^+$ CXCR3$^+$ CRTH2$^-$, and Th2 cells were sorted as CD4$^+$ CXCR3$^-$ CRTH2$^+$. CD8$^+$ effector/effector memory T cells were sorted as CD8$^+$ CD56$^-$ CD45RA$^-$ CD62L$^-$. For activation of human T cells, suspension culture plates were coated with antibodies against human CD3ε and CD28, and sorted T cells were plated in RPMI1640 + GlutaMax I (Thermo Scientific) medium supplemented with FCS (10% v/v, Thermo Scientific), penicillin (100 U ml$^{-1}$, Thermo Scientific), streptomycin (100 µg ml$^{-1}$, Thermo Scientific),

gentamycin (10 μg ml⁻¹, Thermo Scientific) and β-mercaptoethanol (50 ng ml⁻¹, Sigma-Aldrich). To CTL and Th1 cultures, IL-12 (10 ng ml⁻¹, R&D Systems), IL-2 (10 ng ml⁻¹, R&D Systems) and anti-IL-4 (7A3-3, 10 μg ml⁻¹, Miltenyi Biotec) were added, whereas Th2 cells were cultured in the presence of IL-4 (10 ng ml⁻¹, R&D Systems), IL-2 (10 ng ml⁻¹, R&D Systems), anti-IL-12 (C8.6, 10 μg ml⁻¹, Miltenyi Biotec) and anti-IFN-γ (45-15, 10 μg ml⁻¹, Miltenyi Biotec). Cells were withdrawn from coated plates after 24 h of activation, split after 3 days in a 1:3 ratio with fresh medium containing IL-2 (10 ng ml⁻¹, R&D Systems) and analyzed on day 5 of culture. All analyses were carried out in compliance with the relevant ethical regulations.

## RNA isolation and qRT-PCR

To isolate RNA for qRT-PCR analysis or bulk RNA-seq, $10^5$–$10^6$ T cells were harvested and lysed in RA-1 buffer (Macherey & Nagel). Total RNA was purified using the Nucleospin RNA XS Micro kit (Macherey & Nagel) according to manufacturer's instructions, without addition of carrier RNA. For qRT-PCR analysis, RNA was transcribed into cDNA utilizing Taqman Reverse Transcription Reagents (Applied Biosystems). cDNA was then subjected to qRT-PCR analysis using PowerUp SYBR Green or Taqman Fast Advanced Mastermix reagents (Applied Biosystems). Primer sequences and Taqman probes are listed in Supplementary Table 4. Amplifications were performed in triplicates by using a QuantStudio 7 device (Applied Biosystems) and expression levels were quantified with the ΔΔCt-method by normalizing target gene expression to levels of *Hprt* (mouse) or *GAPDH* (human).

## Single-cell RNA library preparation, sequencing and analysis

Single-cell suspensions of CD90⁺ CD8⁺ CD44⁺ T cells were obtained by flow-cytometrical sorting of CD19-depleted splenocytes from LCMV-infected WT and *Il1rl1-ExAB*⁻/⁻ mice at day 7 p.i. with LCMV-WE (2 × 10⁶ PFU). Sorted T cells of individual mice were barcoded using TotalSeq-C anti-mouse Hashtags (anti-mouse Hashtag 1, 2 and 3, all BioLegend). T cells of each genotype were then pooled and applied to the 10x Genomics workflow for cell capturing. For the preparation of scRNA gene expression (GEX), TCR and CiteSeq libraries, the Chromium Next GEM Single Cell 5′ Library & Gel Bead Kit v1.1 as well as the Chromium Single Cell 5′ Feature Barcode Library Kit were used in conjunction with Chromium Controller (10x Genomics). After cDNA amplification the CiteSeq libraries were prepared separately using the Single Index Kit N Set A (10x Genomics). TCR target enrichment was performed using the Chromium Single Cell V(D)J Enrichment Kit for mouse T cells (10x Genomics). Final GEX and TCR libraries were obtained after fragmentation, adapter ligation and final Index PCR using the Single Index Kit T Set A. The Qubit dsDNA HS assay kit (Life Technologies) and a Qubit 2.0 Fluorometer were used for library quantification. Fragment sizes were determined using a Fragment Analyzer device with the NGS Fragment Kit (1–6,000 bp) (Agilent). Sequencing was performed on a NextSeq2000 device (Illumina) using P2 Reagents v3 (200 cycles) with the recommended sequencing conditions for 5′ GEX and barcode libraries (read 1: 26 nt, read 2: 98 nt, index1: 8 nt, index 2: n.a.) and on a NextSeq500 device (Illumina) using a Mid Output v2 Kit (300 cycles) for TCR libraries (read 1: 150 nt, read 2: 150 nt, index 1: 8 nt, index 2: n.a., 20% PhiX spike-in). Raw data were processed using cellranger-3.1.0 with refdata-gex-mm10-2020-A and refdata-cellranger-vdj_GRCm38_alts_ensembl-mouse-2.2.0 as reference. Mkfastq, count and vdj were used in default parameter settings with 3,000 expected cells for demultiplexing, detection of intact cells, quantification of gene expression, antibody capture as well as assembly and quantification of T-cell receptor sequences.

The cellranger output was further analyzed in R using the Seurat package (version 4.0.0)[83]. Hashtag sequences of three individual *Il1rl1-ExAB*⁻/⁻ and WT mice were imported and combined. Centered log ratio transformation was used for normalization. Seurat's default method was used for scaling. Features with correlation coefficients >0.85 to *Gm42418*, *Malat1*, *AY036118* and *Lars2* were removed from the count matrix. Hashtag demultiplexing (representing the three biological replicates per genotype) was performed based on Seurat's HTODemux with the parameter 'positive-quantile' at 0.99. Doublets and untagged cells were filtered out. Cells with expression values for *Cd8a* or *Cd8b1* and *Cd3g*, *Cd3d* or *Cd3e*, with >200 and <4,500 features, and <10% UMI for mitochondrial genes were kept for further analysis.

After ranking by residual variance, 3000 variable genes were determined. The genes encoding TCR variable regions (*Trav*, *Trbv*, *Trdv* and *Trgv*) were removed. 30 principal components were computed and stored. UMAP and t-distributed stochastic neighbor embedding were run using the first 15 principal components. Transcriptionally similar clusters were identified using shared nearest neighbor modularity optimization, with a resolution of 0.35. For visualization, cells of the *Il1rl1-ExAB*⁻/⁻ condition were down-sampled to match the number of cells in the WT condition. Signature genes were identified using the FindAllMarkers function in default parameter settings (only.pos = TRUE, min.pct = 0.25, logfc.threshold = 0.25). Heatmaps and dotplots for the single-cell data were plotted with Seurat's DoHeatmap and DotPlot function, respectively, using default settings. Identified clusters were annotated based on the expression of key markers for CD8⁺ T-cell subsets and cell functions. Two *Klrg1* expressing clusters were merged to the SLEC cluster. Further, two clusters were merged to form the Mito^hi cluster based on their expression of mitochondrial genes. After combining the stated clusters, signature genes were identified again using the same method as described above. Cluster size was defined as the number of cells in one cluster. Relative cluster sizes were calculated by analyzing the number of cells in one cluster per genotype divided by the total number of cells. For feature plots, the expression of single features was plotted on the UMAP by using Seurat's default function FeaturePlot with the option "keep.scale='all'". For differential expression analysis between clusters the FindMarkers function was used.

For the analysis of TCRs, immune profiles were integrated using identical cellular barcodes. For cells with more than one contig for the heavy or light TCR chain the most abundant, productive contig was chosen. Cell numbers were equalized by subsampling the larger condition. Cells without TCR annotation were excluded from the analysis. The R package immunarch (version 0.6.6)[84] was used for clonality analysis after downsampling to the WT condition.

## RNA-seq

Smarta and P14 T cells used for bulk RNA-seq experiments were differentiated as described above. To enrich for ST2-expressing CTLs and Th1 and Th2 cells, ST2⁺ T cells were flow-cytometrically sorted at day 10 of culture. For global analysis of IL-33-responsive genes, T cells were harvested at day 10 of culture and rested for 3 days in the presence of IL-2 (5 ng ml⁻¹) and IL-7 (5 ng ml⁻¹, both Miltenyi Biotec), without irradiated splenocytes or cognate peptide. At day 13, IL-12 (5 ng ml⁻¹) was added to CTLs and Th1 cells to trigger ST2 expression. At day 14 of culture, T cells were subjected to Histopaque density centrifugation (1.083 g ml⁻¹, Sigma-Aldrich) and stimulated in conditioned medium with IL-33 (10 ng ml⁻¹, R&D Systems) for 2 h. When used for RNA-seq, quality of the isolated RNA was assessed with a Fragment Analyzer System (Agilent). All processed samples showed high RNA integrity (RQN > 8). cDNA libraries were prepared using the Smart-seq v4 mRNA Ultra Low Input RNA Kit (Clontech) with up to 10 ng RNA (IL-33-stimulated T cells) or TrueSeq stranded total RNA library kit (Illumina) with up to 1 μg of RNA (ST2-enriched T cells) according to manufacturer's instructions. Paired-end sequencing (2 × 75 bp) of cDNA libraries was performed on an Illumina NextSeq500 device using the NextSeq 500/550 High output Kit v2. Obtained reads were mapped to the mm10 genome (annotation release GRCm38.p6) using Tophat2 (ref. 85) and Bowtie2 (ref. 86) with very sensitive settings. Read counts were determined with featureCounts[87]. DESeq2 (ref. 88) was used in RStudio for differential gene expression analysis. The DESeq2 count matrix was pre-filtered

for genes with ≥100 summarized read counts across the analyzed samples. A gene was considered as differentially expressed when log2 fold change > 1.0 and P adjusted < 0.01. AnnotationDbi[89], Enhanced-Volcano[90], ComplexHeatmap[91], pheatmap[92] and ggplot2 (ref. [93]) were used in RStudio for data visualization.

For PCA and sample distance calculation, a blind variance stabilizing transformation was performed on the unnormalized counts across all samples. Sample distance plots are based on pairwise calculation of the Pearson correlation.

### Gene set enrichment and overrepresentation analysis
For gene set enrichment analysis, the R package clusterProfiler[94] was used. A gene list ranked by log2 fold change containing all expressed genes served as input and was tested for enrichment of biological process gene sets from the gene ontology resource[95,96].

For overrepresentation analysis, differentially expressed genes were split into up- and downregulated genes. Overrepresentation analysis was performed against biological process gene sets from the gene ontology resource using a one-sided hypergeometric test with BH correction. All expressed genes in the respective conditions were used as a background gene list. The results were simplified to reduce overlaps between ontology terms by using the clusterProfiler::simplifyGO function with a cutoff of 0.7.

### Analysis of alternative transcription start sites
Raw RNA-seq reads of indicated T-cell subsets were aligned using hisat2 (version 2.2.1)[97] and assembled using Cufflinks (version 2.2.1)[98] in RABT mode with EnsEMBL annotation release 67. Assemblies were then merged into a new reference annotation with the public reference using the cuffmerge function. The resulting annotation was used for an analysis with the R package ProActiv[42]. The getAlternativePromoters function was used with standard parameters except for minAbs = 5. Only results with false discovery rate < 0.01 were considered.

### Processing of published RNA-seq, ChIP-seq and ATAC-seq data
Fastq files of published RNA-seq datasets were obtained from the NCBI Sequence read archive and aligned using hisat2 with default settings[97]. ChIP- and ATAC-seq data sets were downloaded from the NCBI GEO Database and if required crossmapped to the mm10 genome using CrossMap[99]. All NGS data tracks were visualized in the IGV browser[100].

### Quantification and statistical analysis
Statistical analysis on untransformed or log2-transformed values was performed using GraphPad Prism (v10.0.3). Normal distribution was tested using Shapiro–Wilk and Kolmogorov–Smirnov tests. Unpaired or paired two-tailed Student's t-tests were used when two groups were compared with respect to one parameter. More than two groups were analyzed by one-way ANOVA with Tukey's post hoc test for multiple comparison. For comparisons of more than one parameter between two or more groups, two-way ANOVA with Tukey's post hoc tests (unpaired samples) or two-way repeated measures ANOVA with Šidák's post hoc tests (paired samples) were performed as indicated in the figure legends.

### Reporting summary
Further information on research design is available in the Nature Portfolio Reporting Summary linked to this article.

### Data availability
Generated single-cell and bulk RNA-seq data are provided via the Gene Expression Omnibus (GEO) under accession codes GSE204695 and GSE204693. Published data are accessible via the following GEO accession codes: ChIP-seq: GSM550303 (STAT4), GSM998272 (T-bet), GSM523226 (GATA-3), GSM776557 (human T-bet); ATAC-seq: GSE120532 (CD4+ T cells) and GSE111902 (CD8+ T cells). PhyloP conservation

tracks are provided by the UCSC Sequence and Annotation database https://hgdownload.soe.ucsc.edu/goldenPath/mm10/phyloP60way/ (refs. 43,101). Mast cell and ILC2 RNA-seq data are available at the NCBI Sequence read archive via Run ID SRR7549295 (ILC2s) and SRR6155875 (mast cells). FANTOM5 CAGE-seq data and CAGE-associated transcript data are available from the FANTOM5 collection (https://fantom.gsc.riken.jp/5/)[47,102,103]. Genome releases GRCm38.p6/mm10 and GRCh37.13/hg19 are accessible at Ensembl (http://www.ensembl.org/index.html). Mouse data can be inspected in the UCSC genome browser: https://genome.ucsc.edu/s/agloehning/BrunnerServeetal2023. Source data are provided with this paper.

### Code availability
Code related to the data analysis has been deposited to GitLab (https://agloehninggitlab.gitlab.io/BrunnerServe-Type1-ST2/).

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

## Acknowledgements

We thank J. Kirsch, T. Kaiser (Flow Cytometry Core Facility, DRFZ, Berlin), G. Guerra, K. Lehmann, V. Holecska, I. Panse, C.L. Tran (DRFZ, Berlin), C. Scholl, A. Leschke (Transgenics Core facility, MDC, Berlin), T. Abreu Mota, M. Ji-Lu (University of Basel), A. Greco (DKFZ, Heidelberg) and A.N. Hegazy (Charité, Berlin) for technical assistance and advice. This work was supported by the German Research Foundation (DFG grants LO 1542/5-1 and LO 1542/4-1 to M.L.), the Swiss National Science Foundation (Sinergia grant CRSII3_160772/1 to M.L. and D.D.P., project grant 310030_185318/1 to D.D.P.), the Willy Robert Pitzer Foundation (Pitzer Laboratory of Osteoarthritis Research, 21-033 to M.L.), the Dr. Rolf M. Schwiete Foundation (Osteoarthritis Research Program, 2021-035 to M.L.) and the state of Berlin and the European Regional Development Fund (ERDF 2014–2020 and EFRE 1.8/11 to M.F.M.). S.S. is a member of the Berlin Institute of Health at Charité – Universitätsmedizin Berlin. T.M.B., M.D. and N.D.H. were fellows of the International Max Planck Research School for Infectious Diseases and Immunology.

## Author contributions

T.M.B., S.S and M.L. conceptualized the study. T.M.B., S.S., A.F.M., D.D.P. and M.L. interpreted the results. T.M.B., S.S. and M.L. wrote the manuscript. T.M.B., S.S., A.F.M., J.F., P.S., M.D. and N.D.H. conducted experiments. T.M.B., S.S., F.H., P.D., G.A.H. and M.F.M. performed and analyzed RNA-seq and scRNA-seq experiments. T.M.B. and R.K. designed and generated *Il1rl1-ExAB*[−/−] and *Il1rl1-ExC*[−/−] mice. C.K., T.H. and D.D.P. provided expertise and advice. All authors reviewed and edited the manuscript.

## Funding

## Competing interests

The authors declare the following competing interests: D.D.P. is a founder, shareholder and advisor to Hookipa Pharma, commercializing arenavirus vectors. The remaining authors declare no competing interests.

## Additional information

**Extended data** is available for this paper at https://doi.org/10.1038/s41590-023-01697-6.

**Correspondence and requests for materials** should be addressed to Tobias M. Brunner or Max Löhning.

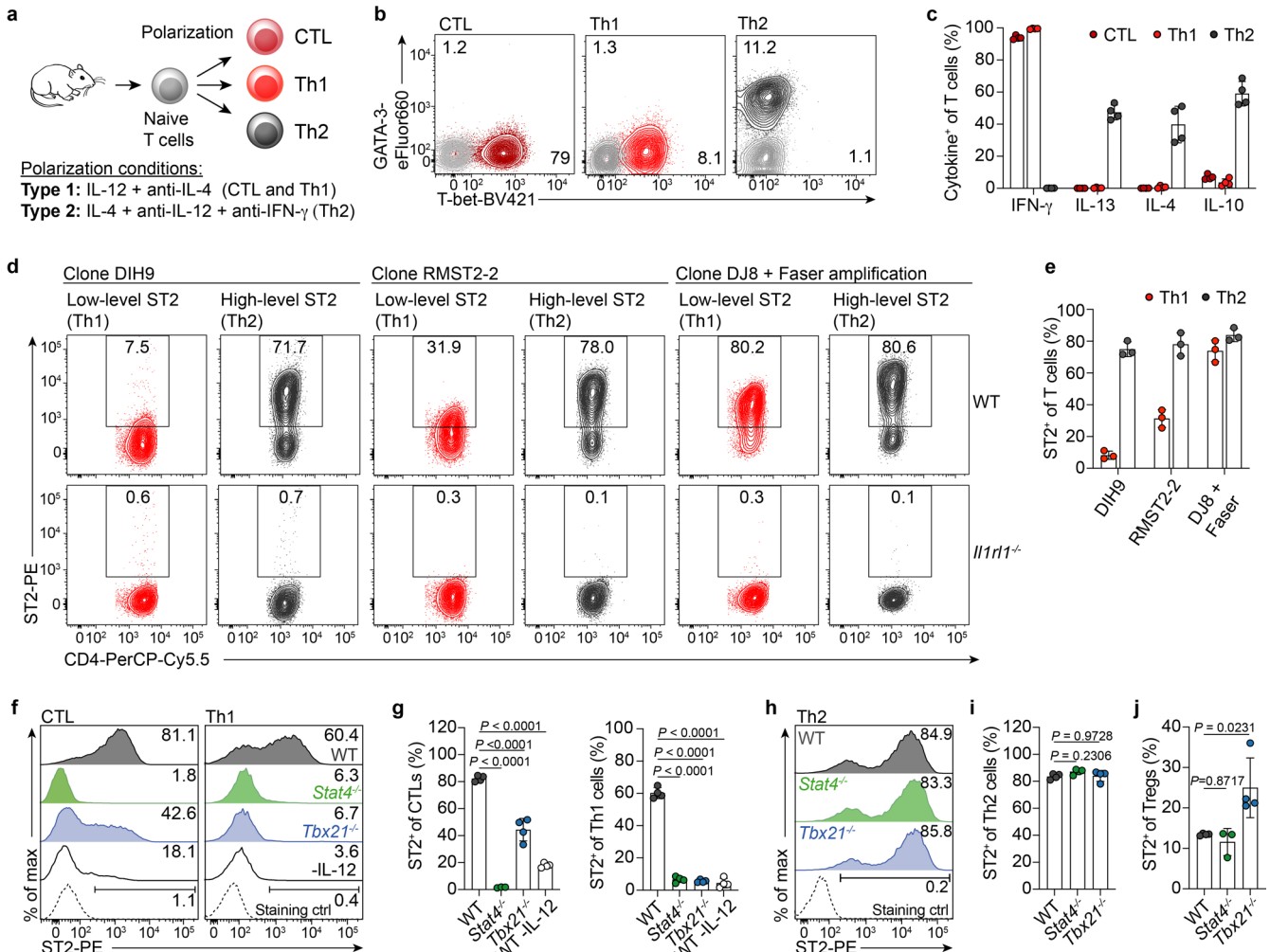

**Extended Data Fig. 1 | ST2 expression by CTLs and Th1 cells is regulated by T-bet and STAT4. a**, Experimental scheme of in vitro T-cell differentiation. **b**, Representative T-bet and GATA-3 stainings of differentiated T cells with expression intensity depicted as geometric mean index (GMI), normalized to isotype control stainings (gray) (*n* = 4). **c**, Cytokine expression of polarized T cells upon restimulation (*n* = 4). **d,e**, In vitro differentiated Th1 and Th2 cells were stained with PE-conjugated ST2 antibodies (clones DIH9 and RMST2-2) or with a digoxigenin-conjugated ST2 antibody (clone DJ8) followed by secondary anti-digoxigenin-PE staining and two rounds of Faser amplification. Representative FACS plots (**d**) and quantification (**e**) of ST2 stainings (*n* = 3).

**f-i**, Representative histograms (**f,h**) and quantification (**g,i**) of ST2 surface expression by in vitro differentiated WT, STAT4- or T-bet-deficient T cells, or T cells activated in the absence of IL-12 (*n* = 4). Stainings with secondary reagents without primary ST2 antibody served as staining controls (ctrl) (dotted line, bottom). **j**, ST2 expression by splenic T$_{reg}$ cells in WT, *Stat4$^{-/-}$* or *Tbx21$^{-/-}$* mice (WT, *Stat4$^{-/-}$*: *n* = 3; *Tbx21$^{-/-}$*: *n* = 4). Data represent two independent experiments and are presented as mean ± SD with each dot representing one mouse (**j**) or one culture performed with T cells from individual mice (**c,e,g,i**). *P* was determined using one-way ANOVA with Tukey's post hoc test (**g,i,j**).

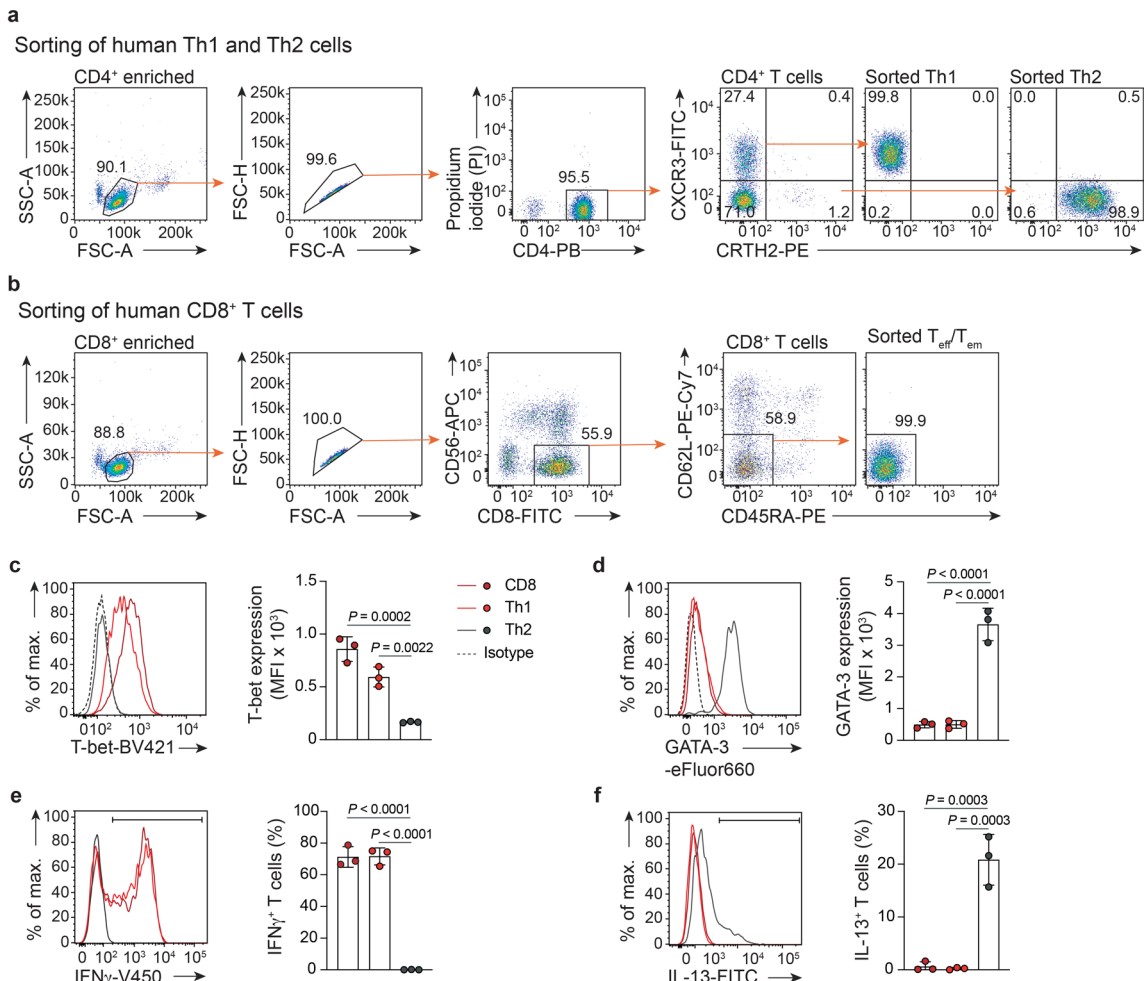

**Extended Data Fig. 2 | Flow-cytometric sorting and analysis of human T cells. a**, Gating strategy for the sorting of human CXCR3⁺ CRTH2⁻ Th1 cells and CXCR3⁻ CRTH2⁺ Th2 cells. **b**, Gating strategy for the sorting of human CD8⁺ T cells. **c,d**, Representative histograms and quantification of T-bet (**c**) and GATA-3 (**d**) expression by activated T cells at day 5 of culture (*n* = 3). **e,f**, Representative histograms and quantification of IFN-γ (**e**) and IL-13 (**f**) expression by T cells stimulated with PMA/ionomycin at day 5 of culture (*n* = 3). Data represent two independent experiments and are presented as mean ± SD with each dot representing one culture performed with T cells from individual donors (**c-f**). *P* was determined using one-way ANOVA with Tukey's post hoc test (**c-f**).

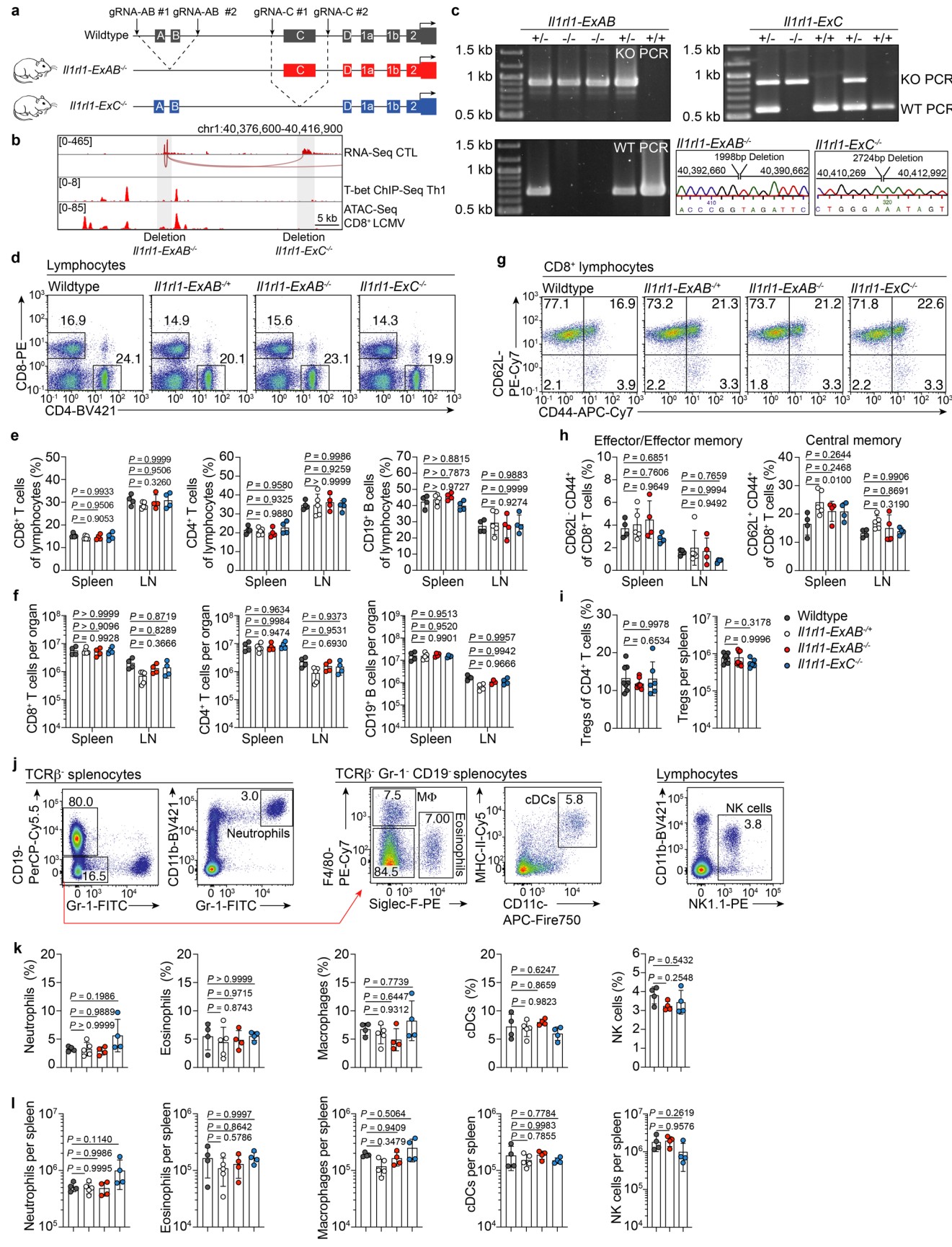

**Extended Data Fig. 3 | See next page for caption.**

**Extended Data Fig. 3 | Generation of *Il1rl1-ExAB*$^{-/-}$ and *Il1rl1-ExC*$^{-/-}$ mice.**
**a**, Schematic depiction of the gene-targeting approach for the generation of *Il1rl1-ExAB*$^{-/-}$ and *Il1rl1-ExC*$^{-/-}$ mice. **b**, RNA-seq coverage and splice junction tracks of ST2$^+$ CTLs showing the areas of deletions (gray), T-bet binding sites and ATAC-seq peaks. **c**, Representative genotyping PCRs to identify heterozygous and homozygous mutant mice. Chromatograms depicting the sequence of joined DNA segments as analyzed by Sanger-sequencing. **d-l**, Analysis of adaptive and innate immune cells in spleens and lymph nodes (LN) of *Il1rl1-ExAB*$^{-/+}$, *Il1rl1-ExAB*$^{-/-}$ and *Il1rl1-ExC*$^{-/-}$ mice. **d**, Representative staining of CD4 and CD8 on splenic T cells. **e,f**, Frequencies (**e**) and absolute cell counts (**f**) of CD8$^+$ T cells, CD4$^+$ T cells and B cells. **g**, Representative staining of CD62L and CD44 on splenic CD8$^+$ T cells. **h**, Frequency of effector and central memory CD8$^+$ T cells. **i**, Frequencies and absolute cell counts of splenic T$_{reg}$ cells. **j**, Gating strategy for the analysis of innate immune cells. **k,l**, Frequencies (**k**) and absolute cell counts (**l**) of splenic neutrophils, eosinophils, macrophages (MΦ), conventional dendritic cells (cDCs) and natural killer (NK) cells. Data represent two independent experiments and are presented as mean ± SD with each dot representing one mouse (WT, *Il1rl1-ExAB*$^{-/-}$, and *Il1rl1-ExC*$^{-/-}$: n = 4, *Il1rl1-ExAB*$^{-/+}$: n = 5; T$_{reg}$ cell analysis: WT, *Il1rl1-ExAB*$^{-/-}$: n = 8 and *Il1rl1-ExC*$^{-/-}$: n = 6; NK cell analysis: WT, *Il1rl1-ExAB*$^{-/-}$ and *Il1rl1-ExC*$^{-/-}$: n = 4). P was determined using one-way ANOVA (**i,k,l**) or two-way AONVA (**e,f,h**) with Tukey's post hoc test.

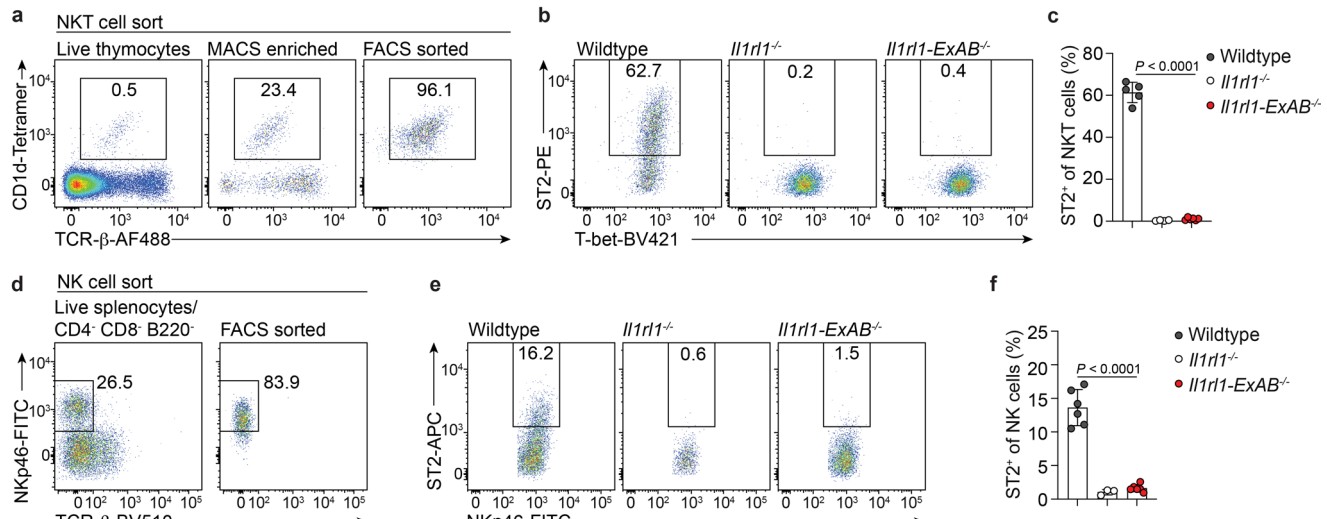

**Extended Data Fig. 4 | Type 1 *Il1rl1* promoter deficiency abrogates ST2 expression by in vitro activated NKT cells and NK cells. a-c**, Thymic CD1d (α-GalCer-loaded)-Tetramer⁺ NKT cells were flow-cytometrically sorted from WT, *Il1rl1⁻/⁻* or *Il1rl1-ExAB⁻/⁻* mice and stimulated in vitro with antibodies against CD3ε and CD28 in CTL/Th1 culture medium for 6 days. **a**, Representative FACS plots showing the purity of NKT cells after MACS pre-enrichment and after FACS sorting. **b,c**, Representative FACS plots (**b**) and quantification (**c**) of ST2 expression by NKT cells (WT: *n* = 5, *Il1rl1⁻/⁻*: *n* = 4, *Il1rl1-ExAB⁻/⁻*: *n* = 6). **d-f**, Splenic NKp46⁺ NK cells were flow-cytometrically sorted from CD4-, CD8- and B220-depleted splenocytes and stimulated with IL-12 + IL-33 for 48 h. **d**, Representative FACS plots showing the purity of NK cells after FACS sorting. **e,f**, Representative FACS plots (**e**) and quantification (**f**) of ST2 expression by NK cells (WT: *n* = 6, *Il1rl1⁻/⁻*: *n* = 3, *Il1rl1-ExAB⁻/⁻*: *n* = 6). Data are pooled from two independent experiments and are presented as mean ± SD with each dot representing one mouse (**c,f**). *P* was determined using one-way ANOVA with Tukey's post hoc test (**c,f**).

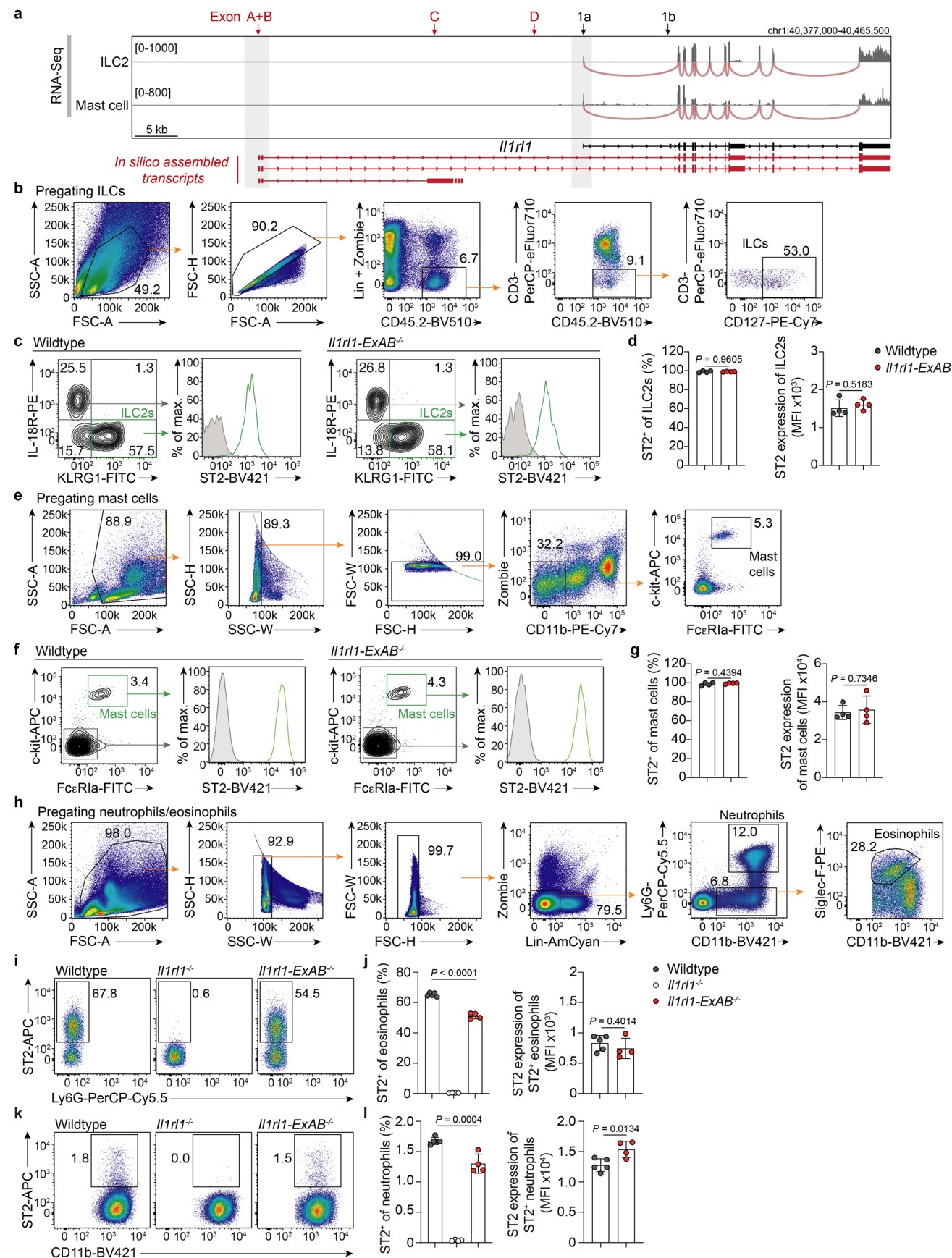

**Extended Data Fig. 5 | See next page for caption.**

**Extended Data Fig. 5 | ST2 expression by ILC2s, mast cells, eosinophils, and neutrophils is largely unaffected by the type 1 *Il1rl1* promoter deletion.**
**a**, RNA-seq coverage tracks and detected splice junctions at the *Il1rl1* locus of ILC2s[104] and mast cells[105]. chr1:40,377,000-40,465,500; GRCm38.p6/mm10 is shown. **b**, Gating strategy for the analysis of Lin⁻CD45⁺ CD3⁻ CD127⁺ ILCs.
**c**, Representative histograms showing ST2 expression by KLRG1⁺ IL-18R⁻ ILC2s isolated from lungs of WT or *Il1rl1-ExAB*⁻/⁻ mice (*n* = 4). **d**, Frequencies of ST2⁺ ILC2s and ST2 expression intensities (MFI). **e**, Gating strategy for the analysis of c-kit⁺ FcεRIa⁺ peritoneal mast cells. **f**, Representative histograms showing ST2 expression by peritoneal mast cells isolated from WT or *Il1rl1-ExAB*⁻/⁻ mice.
**g**, Frequencies of ST2⁺ mast cells and ST2 expression intensities (MFI) (*n* = 4).

**h**, Gating strategy for the analysis of bone marrow eosinophils and neutrophils. **i**, Representative FACS plots showing ST2 expression by eosinophils from WT, *Il1rl1*⁻/⁻ and *Il1rl1-ExAB*⁻/⁻ mice. **j**, Frequencies of ST2⁺ eosinophils and ST2 expression intensity of ST2⁺ eosinophils. **k**, Representative FACS plots showing ST2 expression by neutrophils from WT, *Il1rl1*⁻/⁻ and *Il1rl1-ExAB*⁻/⁻ mice. **l**, Frequencies of ST2⁺ neutrophils and ST2 expression intensity of ST2⁺ neutrophils (WT: n = 5, *Il1rl1*⁻/⁻: n = 4, *Il1rl1-ExAB*⁻/⁻: n = 4). Results are presented as mean ± SD with each dot representing one mouse. Data are representative of two experiments. *P* was determined using two-tailed *t*-tests (**d,g,j,l**) or one-way ANOVA with Tukey's post hoc test (**j,l**, left panels).

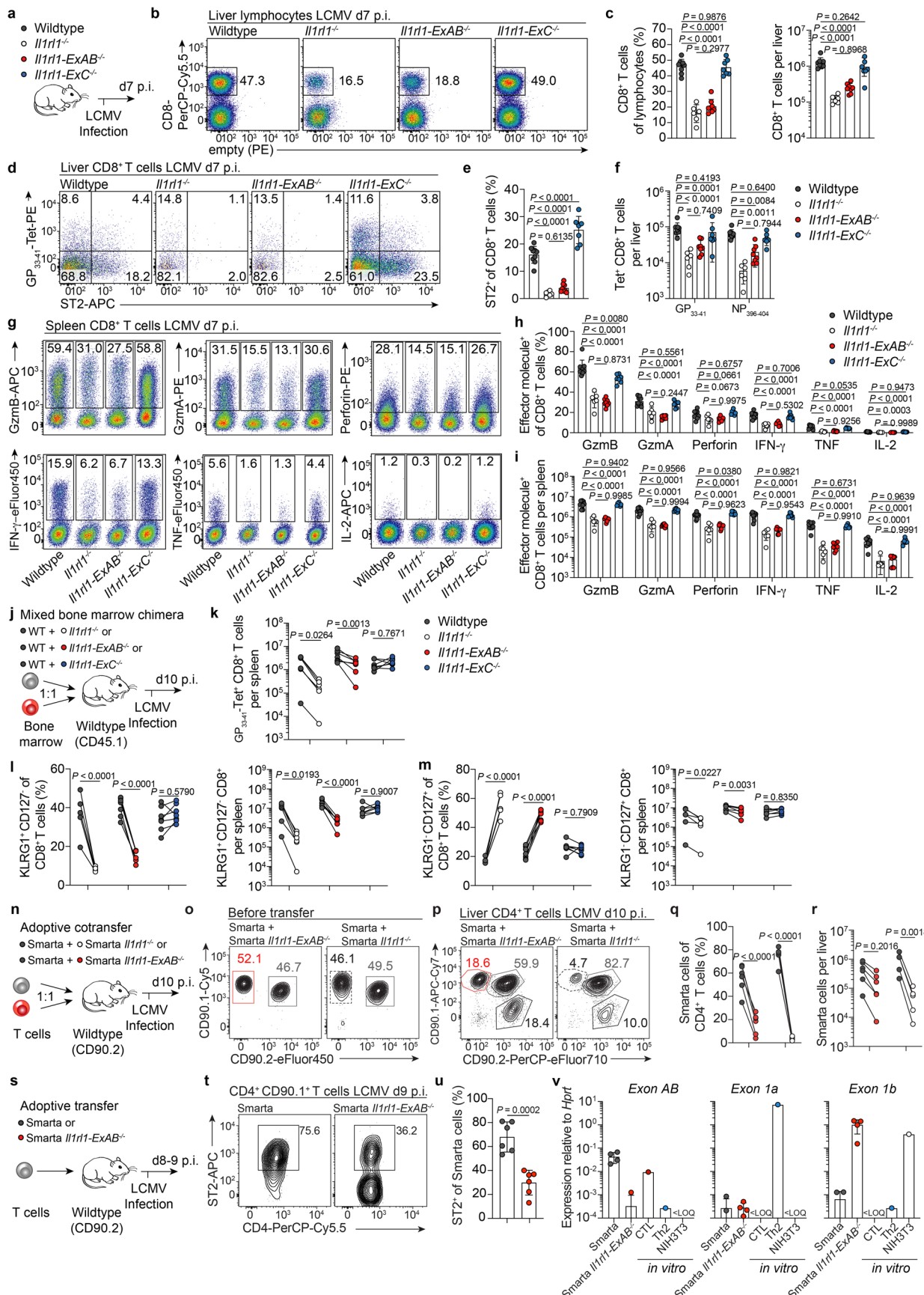

**Extended Data Fig. 6 | See next page for caption.**

**Extended Data Fig. 6 | The type 1 *Il1rl1* promoter drives expansion and activation of antiviral T cells. a-i**, WT, *Il1rl1^-/-^*, *Il1rl1-ExAB^-/-^* and *Il1rl1-ExC^-/-^* mice were infected with LCMV-WE and analyzed on d7 p.i. (WT: *n* = 9, *Il1rl1^-/-^*: *n* = 6, *Il1rl1-ExAB^-/-^*: *n* = 8, *Il1rl1-ExC^-/-^*: *n* = 7). **a**, Experimental outline. **b,c**, Representative FACS plots (**b**) and quantification (**c**) of liver CTLs. **d-f**, Representative FACS plots (**d**) and quantification of ST2 expression (**e**) and LCMV-Tetramer^+^ CTLs in livers (**f**). **g-i**, Representative FACS plots (**g**) and quantification (**h,i**) of effector molecule^+^ CTLs restimulated with LCMV GP$_{33-41}$. **j-m**, Irradiated WT recipients (CD45.1^+^) were reconstituted with WT (CD45.1^+^ CD45.2^+^) and *Il1rl1^-/-^*, *Il1rl1-ExAB^-/-^* or *Il1rl1-ExC^-/-^* (all CD45.2^+^) bone marrow, infected with LCMV-WE and analyzed on d10 p.i. (WT+*Il1rl1^-/-^*: *n* = 6, WT+*Il1rl1-ExAB^-/-^* and WT+*Il1rl1-ExC^-/-^*: *n* = 7). **j**, Experimental outline. **k**, Counts of LCMV GP$_{33-41}$-specific CTLs. **l,m** Frequencies and counts of SLECs (**l**) and MPECs (**m**). **n-r**, Smarta cells (CD90.1^+^) were cotransferred with *Il1rl1^-/-^* or *Il1rl1-ExAB^-/-^* Smarta cells (CD90.1^+^ CD90.2^+^) into WT mice (CD90.2^+^). Recipients were infected with LCMV-Cl13 and analyzed at d10 p.i. (*Il1rl1-ExAB^-/-^* Smarta: *n* = 6,

*Il1rl1^-/-^* Smarta: *n* = 5). **n**, Experimental outline. **o,p**, Representative FACS plots showing CD4^+^ T-cell populations before transfer (**o**) and after infection (**p**). **q,r**, Frequencies (**q**) and counts (**r**) of Smarta cells in livers. **s-v**, Smarta, *Il1rl1^-/-^* Smarta or *Il1rl1-ExAB^-/-^* Smarta T cells (CD90.1^+^) were cotransferred into WT mice (CD90.2^+^). Recipients were infected with LCMV-Cl13 and analyzed at d8-9 p.i. **s**, Experimental outline. **t,u**, Representative FACS plots (**t**) and quantification of ST2 expression (**u**) (*n* = 6). **v**, Transferred cells were sorted and *Il1rl1* first exon usage was quantified (Smarta: *n* = 4 with two samples <LOQ in Exon 1a and 1b reactions; *Il1rl1-ExAB^-/-^*: *n* = 4 with three samples <LOQ in Exon AB reaction; CTL, Th2, and NIH3T3 controls: *n* = 1). Data represent one (**j-v**), two (**a-f**) or three (**g-i**) independent experiments and are presented as mean ± SD with each dot representing one mouse. *P* was determined using two-tailed *t*-tests (**u**), one-way ANOVA with Tukey's post hoc test (**c,e,h,i**), two-way ANOVA with Tukey's post hoc test (**f**) or two-way RM ANOVA with Šidák's post hoc test (**k,l,m,q,r**).

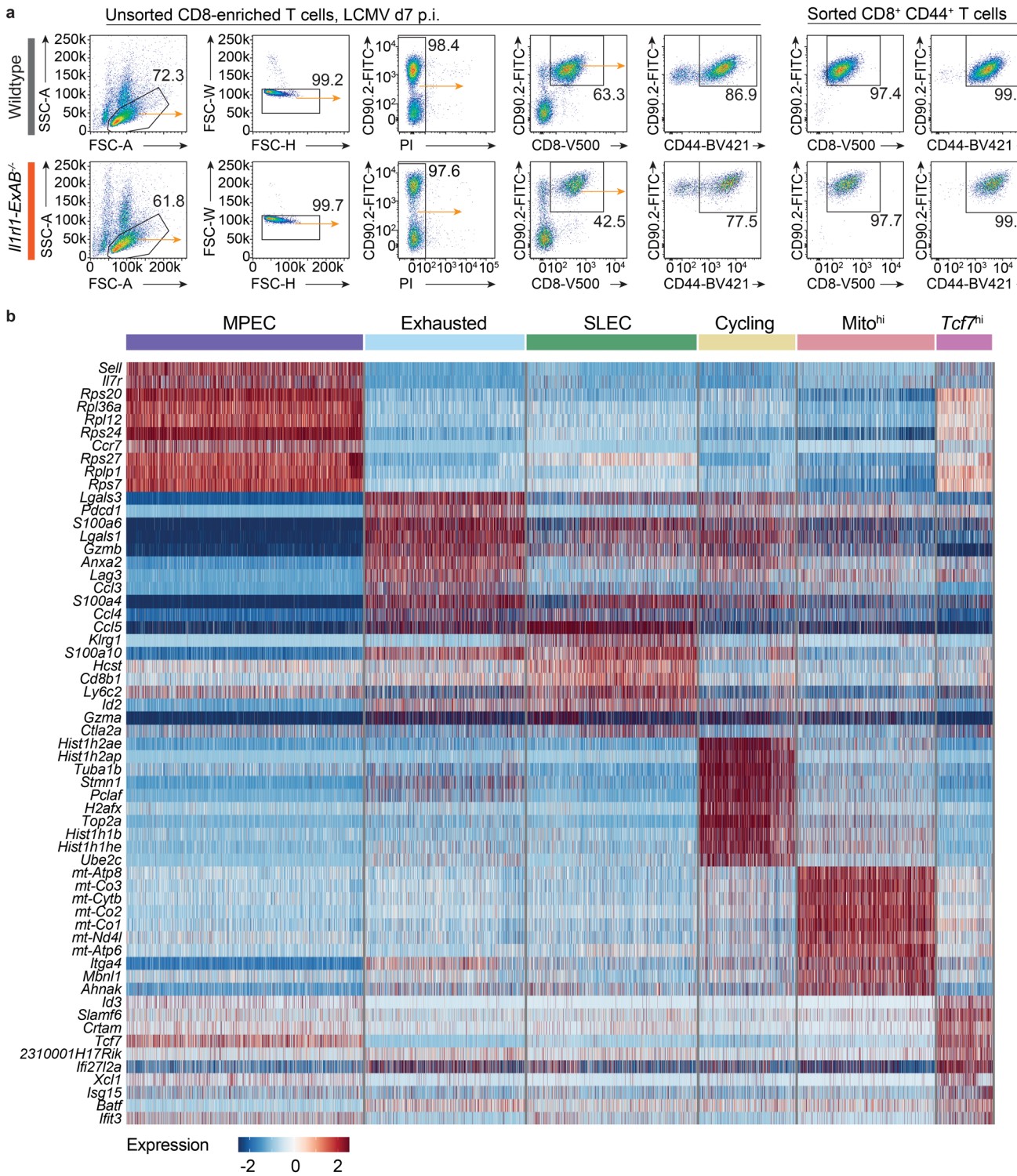

**Extended Data Fig. 7 | scRNA-seq profiling of antiviral T cells in WT and *Il1rl1-ExAB*[-/-] mice. a**, Gating strategy for the flow-cytometric sorting of CD44[+] CTLs from spleens of LCMV-infected WT and *Il1rl1-ExAB*[-/-] mice. **b**, Heatmap displaying all analyzed CTLs and the top ten marker genes per cluster.

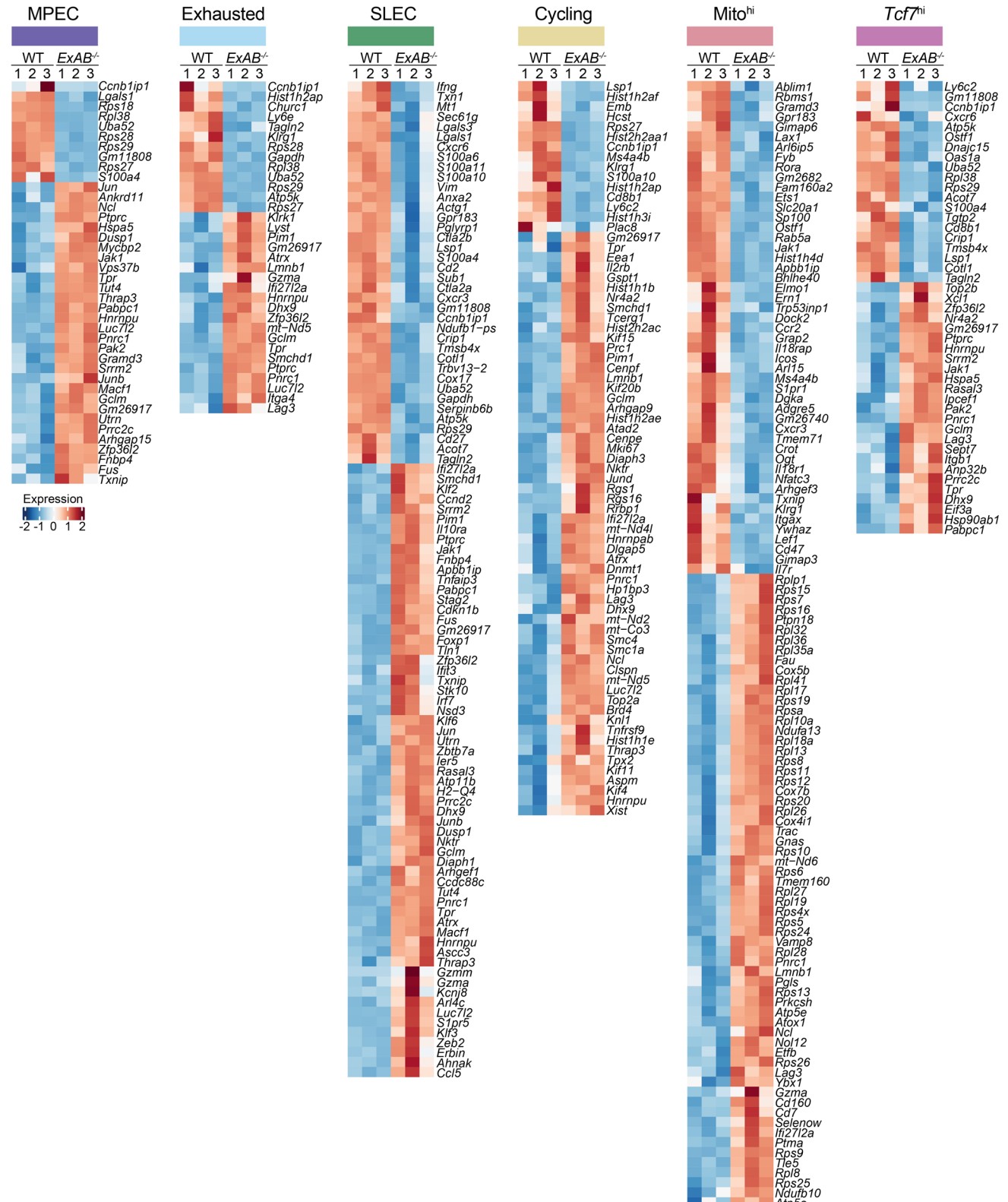

**Extended Data Fig. 8 | Gene expression comparison of WT and *Il1rl1-ExAB*⁻/⁻ CTLs in each scRNA-seq cluster.** Heatmaps displaying all genes differentially expressed between CTLs from LCMV-infected WT and *Il1rl1-ExAB*⁻/⁻ mice in each CTL cluster identified by scRNA-seq (*n* = 3) (log2 fold change > 0.5; *P* adjusted < 0.05). *P* was determined using two-sided Wilcoxon rank sum test with Bonferroni correction.

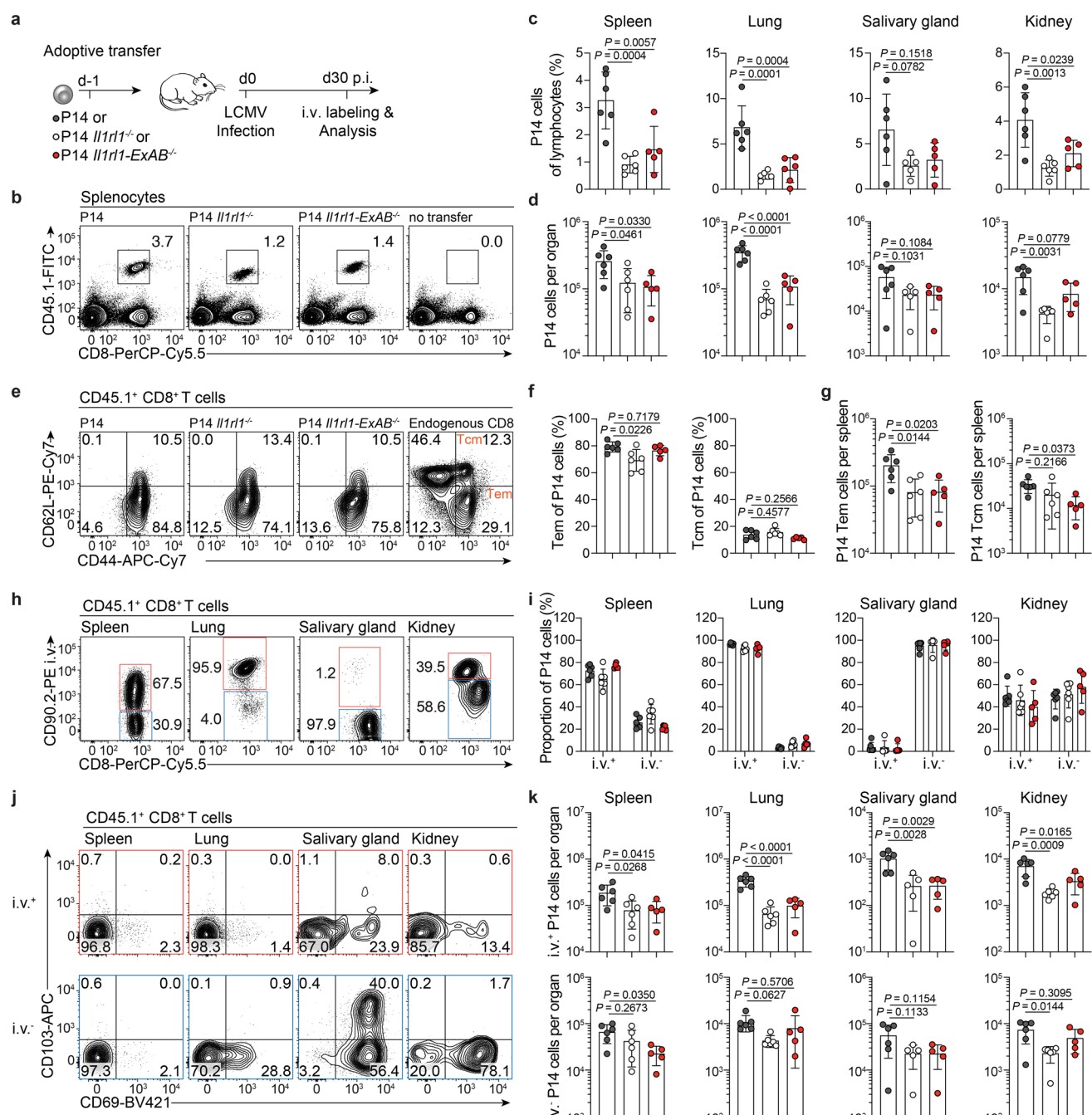

**Extended Data Fig. 9 | Efficient CD8 memory T-cell formation depends on T-cell-intrinsic ST2 signaling. a-k**, P14, *Il1rl1*[-/-] P14 or *Il1rl1-ExAB*[-/-] P14 T cells (all CD45.1[+]) were adoptively transferred into WT mice (CD45.2[+]). Recipients were infected with LCMV-Cl13 (200 PFU) and analyzed at d30 p.i. (P14: *n* = 6, P14 *Il1rl1*[-/-]: *n* = 6, P14 *Il1rl1-ExAB*[-/-]: *n* = 5). **a**, Experimental outline. **b-d**, Representative FACS plots (**b**) and quantification (**c,d**) of memory P14 cells in indicated organs. **e-g**, Representative FACS plots (**e**) and quantification (**f,g**) of splenic effector memory (Tem, CD44[+] CD62L[-]) and central memory (Tcm, CD44[+] CD62L[+]) T-cell subsets.

**h,i**, Representative FACS plots (**h**) and quantification (**i**) of P14 cells labeled in vivo by i.v. injection of CD90.2-PE antibody prior to sacrificing mice. **j**, Representative FACS plots showing CD69 and CD103 expression in i.v.[+] and i.v.[-] P14 cells. **k**, Counts of i.v.[+] and i.v.[-] P14 cells in spleens, lungs, salivary glands and kidneys of infected animals. Results are presented as mean ± SD with each dot representing one mouse. *P* was determined using one-way ANOVA with Tukey's post hoc test (**c,d,f,g,k**).

# Reporting Summary

## Statistics

For all statistical analyses, confirm that the following items are present in the figure legend, table legend, main text, or Methods section.

| n/a | Confirmed | |
|---|---|---|
| ☐ | ☒ | The exact sample size (*n*) for each experimental group/condition, given as a discrete number and unit of measurement |
| ☐ | ☒ | A statement on whether measurements were taken from distinct samples or whether the same sample was measured repeatedly |
| ☐ | ☒ | The statistical test(s) used AND whether they are one- or two-sided<br>*Only common tests should be described solely by name; describe more complex techniques in the Methods section.* |
| ☒ | ☐ | A description of all covariates tested |
| ☐ | ☒ | A description of any assumptions or corrections, such as tests of normality and adjustment for multiple comparisons |
| ☐ | ☒ | A full description of the statistical parameters including central tendency (e.g. means) or other basic estimates (e.g. regression coefficient) AND variation (e.g. standard deviation) or associated estimates of uncertainty (e.g. confidence intervals) |
| ☐ | ☒ | For null hypothesis testing, the test statistic (e.g. *F*, *t*, *r*) with confidence intervals, effect sizes, degrees of freedom and *P* value noted<br>*Give P values as exact values whenever suitable.* |
| ☒ | ☐ | For Bayesian analysis, information on the choice of priors and Markov chain Monte Carlo settings |
| ☒ | ☐ | For hierarchical and complex designs, identification of the appropriate level for tests and full reporting of outcomes |
| ☐ | ☒ | Estimates of effect sizes (e.g. Cohen's *d*, Pearson's *r*), indicating how they were calculated |

*Our web collection on statistics for biologists contains articles on many of the points above.*

## Software and code

Policy information about availability of computer code

| | |
|---|---|
| Data collection | FACS Aria, Aria II and LSRFortessa flowcytometers (BD);  MACSQuant (Miltenyi Biotec); Immunospot (CTL); QuantStudio 7 (Thermo Scientific); Diva software v8.1 (BD); Chromium Controller (10x Genomics); Qubit 2.0 Fluorometer (ThermoFisherScientific) ; Fragment Analyzer (Agilent); NextSeq500 (Illumina); NextSeq2000 (Illumina) |
| Data analysis | Flow cytometry data were analyzed using FlowJo (v10.7.1). Statistical analysis was performed with Graphpad Prism (v.10.0.3).<br>Analysis of sequencing data was performed using the following packages:<br><br>cellranger_3.1.0,<br>R_4.2.2,<br>Seurat_4.0.0,<br>immunarch_0.6.6,<br>TopHat_2.1.1,<br>Bowtie_2.2.8.0,<br>featureCounts_1.5.1,<br>DESeq2_1.36.0,<br>AnnotationDbi_1.58.0<br>EnhancedVolcano_1.14.0,<br>ComplexHeatmap_2.12.1,<br>pheatmap_1.0.12,<br>ggplot2_3.4.0,<br>clusterProfiler_4.4.4,<br>hisat2_2.2.1,<br>cufflinks_2.2.1,<br>proActiv_1.6.0, |

CrossMap_0.5.2
Integrated Genome Viewer (Snapshot 12/21/2019 and 2.16).

Code related to the data analysis and R session information of this study have been deposited to GitLab. (https://agloehninggitlab.gitlab.io/BrunnerServe-Type1-ST2/).

For manuscripts utilizing custom algorithms or software that are central to the research but not yet described in published literature, software must be made available to editors and reviewers. We strongly encourage code deposition in a community repository (e.g. GitHub). See the Nature Portfolio guidelines for submitting code & software for further information.

## Data

Policy information about availability of data

All manuscripts must include a data availability statement. This statement should provide the following information, where applicable:
- Accession codes, unique identifiers, or web links for publicly available datasets
- A description of any restrictions on data availability
- For clinical datasets or third party data, please ensure that the statement adheres to our policy

Generated single cell and bulk RNA-Seq data are provided via the Gene Expression Omnibus (GEO) under accession codes GSE204695 and GSE204693. Published data are accessible via the following GEO accession codes: Chip-Seq: GSM550303 (STAT4), GSM998272 (T-bet), GSM523226; (GATA-3) GSM776557 (human T-bet); ATAC-Seq: GSE120532 (CD4+ T cells), GSE111902 (CD8+ T cells). PhyloP conservation tracks are provided by the UCSC Sequence and Annotation database https://hgdownload.soe.ucsc.edu/goldenPath/mm10/phyloP60way/. Mast cell and ILC2 RNA-Seq data are available at the NCBI Sequence read archive via Run ID SRR7549295 (ILC2s) and SRR6155875 (mast cells). FANTOM5 CAGE-Seq data and CAGE-associated transcript data are available from the FANTOM5 collection (https://fantom.gsc.riken.jp/5/). Genome releases GRCm38.p6 and GRCh37.p13 are accessible at Ensemble (http://www.ensembl.org/index.html). Mouse data can be inspected in the UCSC genome browser: https://genome.ucsc.edu/s/agloehning/BrunnerServeetal2023. Source data are provided with this paper.

# Field-specific reporting

Please select the one below that is the best fit for your research. If you are not sure, read the appropriate sections before making your selection.

☒ Life sciences  ☐ Behavioural & social sciences  ☐ Ecological, evolutionary & environmental sciences

For a reference copy of the document with all sections, see nature.com/documents/nr-reporting-summary-flat.pdf

# Life sciences study design

All studies must disclose on these points even when the disclosure is negative.

| | |
|---|---|
| Sample size | No specific statistical tests were performed to predetermine the sample size. Sample size was chosen based on previous experiments (Bonilla et al. Science 2012; Baumann et al. PNAS 2015; Baumann et al. Frontiers in Immunology 2019) and availability of mice with the appropriate genotype. Sample size for each experiment is indicated in the legend. |
| Data exclusions | Low quality cells and biologically irrelevant cells types were removed during the sc-RNASeq analysis as follows: Features with correlation coefficients >0.85 to Gm42418, Malat1, AY036118 and Lars2 were removed from the count matrix. Hashtag demultiplexing (representing the 3 biological replicates per genotype) was performed based on Seurats HTODemux with the parameter positive-quantile at 0.99. Doublets and untagged cells were filtered out. Cells with expression values for Cd8a or Cd8b1 and Cd3g, Cd3d or Cd3e, with >200 and <4500 features and <10% UMI for mitochondrial genes were kept for further analysis.After ranking by residual variance, 3000 variable genes were determined. The genes encoding TCR variable regions (Trav, Trbv, Trdv, Trgv) were removed for further analyses. |
| Replication | Experiments were performed with multiple biologically independent samples. The number of independent samples (n) and the number of independent repetitions for each experiment are indicated in the legend. |
| Randomization | Allocation of mouse samples was not randomized as mice were age- and sex-matched and allocated to experimental groups based on genotyping results. Human buffy coats of heathly patients were obtained from the German Red Cross (DKR Berlin) without information about gender or age. Human CTLs, Th1, and Th2 cells were randomly sorted from individual donors. |
| Blinding | For experiments involving murine and human samples, investigators were not blinded as the investigators who performed the experiment also planned them. Additionally, for murine experiments, genotyping was required for group allocation. |

# Reporting for specific materials, systems and methods

We require information from authors about some types of materials, experimental systems and methods used in many studies. Here, indicate whether each material, system or method listed is relevant to your study. If you are not sure if a list item applies to your research, read the appropriate section before selecting a response.

## Materials & experimental systems

| n/a | Involved in the study |
|---|---|
| ☐ | ☒ Antibodies |
| ☐ | ☒ Eukaryotic cell lines |
| ☒ | ☐ Palaeontology and archaeology |
| ☐ | ☒ Animals and other organisms |
| ☐ | ☒ Human research participants |
| ☒ | ☐ Clinical data |
| ☒ | ☐ Dual use research of concern |

## Methods

| n/a | Involved in the study |
|---|---|
| ☒ | ☐ ChIP-seq |
| ☐ | ☒ Flow cytometry |
| ☒ | ☐ MRI-based neuroimaging |

## Antibodies

Antibodies used

The following antibodies have been used for flow cytometry analysis and sorting (Antibody, Conjugate, Clone, Provider and Cat#, Dilution):
Rat monoclonal anti-mouse CD25, Biotin, 7D4, eBioscience Cat#13-0252-85, 1:200
Hamster monoclonal anti-mouse CXCR3, Biotin, CXCR3-173, eBioscience Cat#13-1831-82, 1:500
Hamster monoclonal anti-mouse CXCR3, PerCP-Cyanine5.5, CXCR3-173, eBioscience Cat#45-1831-82, 1:100
Mouse monoclonal anti-mouse/human CD3, PerCP-eFluor710, 17A2, eBioscience Cat#46-0032-82, 1:300
Rat monoclonal anti-mouse CD90.2 (Thy1.2), PE, 30-H12, Biolegend Cat#105308, 3µg i.v.
Rat monoclonal anti-mouse CD4, Biotin, RM4-5, eBioscience Cat#13-0042-82, 1:300
Rat monoclonal anti-mouse CD4, PerCP-Cyanine5.5, RM4-5, eBioscience Cat#45-0042-82, 1:400
Rat monoclonal anti-mouse CD4, APC-Cy7, RM4-5, Biolegend Cat#100526, 1:200
Rat monoclonal anti-mouse CD4, FITC, RM4-5, eBioscience Cat#11-0042-85, 1:200
Rat monoclonal anti-mouse CD4, Pacific Blue, YTS19.1, DRFZ Inhouse Cat#-, 1:200
Rat monoclonal anti-mouse CD4, BV421, GK1.5, Biolegend Cat#100443, 1:200
Rat monoclonal anti-mouse CD4, APC-Cy7, GK1.5, BD Cat#552051, 1:200
Rat monoclonal anti-mouse CD8a, Biotin, 53-6.7, BD Cat#553029, 1:300
Rat monoclonal anti-mouse CD8a, PE, 53-6.7, BD Cat#553033, 1:200
Rat monoclonal anti-mouse CD8a, V500, 53-6.7, BD Cat#560778, 1:200
Rat monoclonal anti-mouse CD8a, BV605, 53-6.7, Biolegend Cat#100744, 1:100
Rat monoclonal anti-mouse CD8a, BV786, 53-6.7, Biolegend Cat#100750, 1:100
Rat monoclonal anti-mouse CD8a, PerCP-Cyanine5.5, 53-6.7, eBioscience Cat#45-0081-82, 1:200
Rat monoclonal anti-mouse CD8a, eFluor450, 53-6.7, eBioscience Cat#48-0081-82, 1:200
Hamster monoclonal anti-mouse TCR ß Chain, A488, H57-597, Biolegend Cat#109215, 1:200
Hamster monoclonal anti-mouse TCR ß Chain, BV510, H57-597, BD Cat#563221, 1:200
 Rat monoclonal anti-mouse IL18R, PE, P3TUNYA, eBioscience, Cat#12-5183-82, 1:200
Rat monoclonal anti-mouse TCR Va2, FITC, B20.1, BD Cat#553288, 1:100
Rat monoclonal anti-mouse CD19, Biotin, 1D3, eBioscience Cat#14-0193-85, 1:200
Rat monoclonal anti-mouse CD19, Cy5, 1D3, DRFZ Inhouse Cat#-, 1:400
Rat monoclonal anti-mouse CD19, APC-Fire750, 6D5, Biolegend Cat#115558, 1:200
Rat monoclonal anti-mouse CD19, APC, eBio1D3, eBioscience Cat#17-0193-82, 1:200
Rat monoclonal anti-mouse CD19, PerCP-Cyanine5.5, eBio1D3, eBioscience Cat#45-0193-82, 1:400
Rat monoclonal anti-mouse/human B220, Biotin, RA3-6B2, Biolegend Cat#103204, 1:100
Rat monoclonal anti-mouse/human B220, APC-Fire750, RA3-6B2, Biolegend Cat#103260, 1:200
Mouse monoclonal anti-mouse CD90.1 (Thy1.1), APC-Cy7, Ox-7 Biolegend 202520 1:200
Mouse monoclonal anti-mouse CD90.1 (Thy1.1), Cy5 Ox-7, DRFZ Inhouse - 1:400
Mouse monoclonal anti-mouse CD90.1 (Thy1.1), Pacific Blue, Ox-7, DRFZ Inhouse, Cat#-, 1:300
Rat monoclonal anti-mouse CD90.2 (Thy1.2), FITC, 53-2.1, Biolegend Cat#140304, 1:200
Rat monoclonal anti-mouse CD90.2 (Thy1.2), PE, 53-2.1, Biolegend Cat#140308, 1:200
Mouse monoclonal anti-mouse CD45.1 (Ly5.1), Pacific Blue, A20, Biolegend Cat#110722, 1:200
Mouse monoclonal anti-mouse CD45.1 (Ly5.1), FITC, A20, Biolegend Cat#110706, 1:100
Mouse monoclonal anti-mouse CD45.1 (Ly5.1), PE, A20, BD Cat#553776, 1:200
Mouse monoclonal anti-mouse CD45.2 (Ly5.2), APC, 104, BD Cat#558702, 1:100
Mouse monoclonal anti-mouse CD45.2 (Ly5.2), V500, 104, BD Cat#562129, 1:100
Mouse monoclonal anti-mouse CD45.2 (Ly5.2), APC-Fire750, 104, Biolegend Cat#109852, 1:200
Mouse monoclonal anti-mouse CD45.2 (Ly5.2), BV785, 104, Biolegend Cat#109839, 1:100
Rat monoclonal anti-mouse CD45.2 (Ly5.2), BV510, 30-F11, Biolegend Cat#103138, 1:200
Rat monoclonal anti-mouse CD44,  BV421, IM7, Biolegend, Cat#103040, 1:200
Rat monoclonal anti-mouse CD44, APC-Cy7, IM7, Biolegend Cat#103028, 1:200
Rat monoclonal anti-mouse CD62L, PE-Cy7, MEL-14, eBioscience Cat#25-0621-82, 1:300
Hamster monoclonal anti-mouse KLRG1, FITC, 2F1, eBioscience Cat#11-5893-82, 1:200
Hamster monoclonal anti-mouse KLRG1, PerCP-Cyanine5.5, 2F1, Biolegend Cat#100734, 1:100
Rat monoclonal anti-mouse CD127, PE-Cy7, A7R34, eBioscience Cat#25-1271-82, 1:100
Rat monoclonal anti-mouse CD127, BV421, A7R34, Biolegend Cat#135027, 1:100
Mouse monoclonal anti-mouse NK1.1, Biotin, PK136, eBioscience Cat#13-5941-85, 1:500
Mouse monoclonal anti-mouse NK1.1, PE, PK136, BD Cat#557391, 1:200
Rat monoclonal anti-mouse Gr-1, Biotin, RB6-8C5, eBioscience Cat#13-5931-85, 1:500
Rat monoclonal anti-mouse Gr-1, FITC, RB6-8C5, DRFZ Inhouse Cat#-, 1:800
Rat monoclonal anti-mouse Gr-1, APC-Cy7, RB6-8C5, Biolegend Cat#108424, 1:200
Rat monoclonal anti-mouse NKp46, FITC, 29A1.4, Biolegend Cat#137606, 1:100

Rat monoclonal anti-mouse Siglec-F, PE, E50-2440, BD Cat#552126, 1:100
Rat monoclonal anti-mouse c-kit, APC, 2B8, eBioscience Cat#12-1171-82, 1:100
Hamster monoclonal anti-mouse FceRIa, FITC, MAR-1, eBioscience Cat#11-5898-82, 1:100
Hamster monoclonal anti-mouse FceRIa, APC-Cy7, MAR-1, Biolegend Cat#134326, 1:100
Rat monoclonal anti-mouse MHC Class II, Cy5, M5/114, DRFZ Inhouse Cat#-, 1:100
Hamster monoclonal anti-mouse CD11c, Biotin, HL3, eBioscience Cat#13-0114-85, 1:500
Hamster monoclonal anti-mouse CD11c, APC-Fire750, N418, Biolegend Cat#117352, 1:200
Rat monoclonal anti-mouse/human CD11b, Biotin, M1/70, eBioscience Cat#13-0112-85, 1:800
Rat monoclonal anti-mouse/human CD11b, PE-Cy7, M1/70, BD Cat#552850, 1:200
Rat monoclonal anti-mouse/human CD11b, BV421, M1/70, Biolegend Cat#101236, 1:200
Rat monoclonal anti-mouse/human CD11b, APC-Fire750, M1/70, Biolegend Cat#101262, 1:200
Rat monoclonal anti-mouse Ly6G, PerCP-Cyanine5.5, 1A8, Biolegend Cat#127616, 1:100
Rat monoclonal anti-mouse F4/80, PE-Cy7, BM8, Biolegend Cat#123114, 1:100
Rat monoclonal anti-mouse F4/80, APC-Fire750, BM8, Biolegend Cat#123152, 1:100
Hamster monoclonal anti-mouse CD69, BV421, H1.2F3, Biolegend Cat#104545, 1:100
Hamster monoclonal anti-mouse CD103, APC, 2E7, Biolegend Cat#121414, 1:100
Rat monoclonal anti-mouse ST2, BV421, DIH9, Biolegend Cat#145309, 1:100
Rat monoclonal anti-mouse ST2, PE, DIH9, Biolegend Cat#145303, 1:100
Rat monoclonal anti-mouse ST2, PE, RMST2-2, eBioscience Cat#12-9335-82, 1:100
Rat monoclonal anti-mouse ST2, Digoxigenin, DJ8, mbBioproducts, Conjugated in DRFZ Cat#1001101, 1:900
Sheep polyclonal anti-Digoxigenin Fab fragments, APC, -, Roche, Conjugated in DRFZ Cat#11214667001, 1:800
Sheep polyclonal anti-Digoxigenin Fab fragments, PE, -, Roche, Conjugated in DRFZ Cat#11214667001, 1:800
Mouse monoclonal anti-mouse/human T-bet, BV421, 4B10, Biolegend Cat#644816, 1:100
Mouse monoclonal anti-mouse/human T-bet, PE, 4B10, eBioscience Cat#12-5825-82, 1:100
Rat monoclonal anti-mouse/human GATA3, eFluor660, TWAJ, eBioscience Cat#50-9966-42, 1:100
Rat monoclonal anti-mouse FoxP3, eFluor450, FJK-16s, eBioscience Cat#48-5773-82, 1:100
Rat monoclonal anti-mouse Ki67, FITC, SolA15, eBioscience Cat#11-5698-82, 1:100
Mouse IgG1k Isotype Control, BV421, MOPC-21, Biolegend Cat#400158, 1:150
Mouse IgG1k Isotype Control, PE, P3.6.2.8.1, eBioscience Cat#12-4714-82, 1:200
Rat IgG2bk Isotype Control, eFluor660, 10H5, eBioscience Cat#50-4031-82, 1:400
Rat monoclonal anti-mouse IFN-g, eFluor450, XMG1.2, eBioscience Cat#48-7311-82, 1:100
Rat monoclonal anti-mouse IFN-g, APC, XMG1.2, Biolegend Cat#505810, 1:100
Rat monoclonal anti-mouse IL-2, APC, JES6-5H4, BD Cat#554429, 1:100
Rat monoclonal anti-mouse TNF, eFluor450, MP6-XT22, eBioscience Cat#48-7321-82, 1:100
Mouse monoclonal anti-mouse Granzyme A, PE, 3G8.5, Biolegend Cat#149704, 1:100
Mouse monoclonal anti-mouse/human Granzyme B, APC, QA16A02, Biolegend Cat#372204, 1:100
Rat monoclonal anti-mouse Perforin, PE, S16009B, Biolegend Cat#154406, 1:100
Rat monoclonal anti-mouse IL-4, PE, 11B11, eBioscience Cat#12-7041-81, 1:100
Rat monoclonal anti-mouse IL-13, A488, eBio13A, eBioscience Cat#53-7133-82, 1:200
Rat monoclonal anti-mouse IL-10, APC, JES5-16E3, eBioscience Cat#17-7101-82, 1:100
TotalSeq-C0301 anti-mouse Hashtag 1, Barcoded, M1/42; 30-F11, Biolegend Cat#155861, 1:200
TotalSeq-C0302 anti-mouse Hashtag 2, Barcoded, M1/42; 30-F11, Biolegend Cat#155863, 1:200
TotalSeq-C0303 anti-mouse Hashtag 3, Barcoded, M1/42; 30-F11, Biolegend Cat#155865, 1:200
a-Galactosylceramide-loaded CD1d-Tetramers, PE, -, MBL Cat#TS-MCG-1, 1:50
LCMV GP33-41 peptide-loaded MHC class I (H2-Db) tetramers, PE, -, MBL Cat#TB-M512-1, 1:50
LCMV NP396-404 peptide-loaded MHC class I (H2-Db) tetramers, APC, -, NIH Tetramer Core facility Cat#-, 1:50
Mouse monoclonal anti-human IFN-g, V450, B27, BD Cat#560372, 1:100
Mouse monoclonal anti-human IL-13, FITC, PVM13-1, ebioscience Cat#11-7139-42, 1:40
Mouse monoclonal anti-human CXCR3, FITC, G025H7, Biolegend Cat#353704, 1:50
Mouse monoclonal anti-human CD45RA, PE, HI100, DRFZ Inhouse Cat#-, 1:100
Rat monoclonal anti-human CRTH2, Biotin, BM16, Miltenyi Biotec Cat#130-113-599, 1:10
Human monoclonal anti-human CD56, APC, REA196, Miltenyi Biotec Cat#130-113-310, 1:50
Mouse monoclonal anti-human CD4, Pacific Blue, TT1, DRFZ Inhouse Cat#-, 1:200
Mouse monoclonal anti-human CD62L, PE-Cy7, DREG-56, Biolegend Cat#304822, 1:300
Mouse monoclonal anti-human CD8, FITC, GN11/134.7, DRFZ Inhouse Cat#-, 1:100
Zombie UV fixable viability kit, -, -, BioLegend Cat#423108, 1:200
Live/Dead Fixable Near-IR Dead Cell stain kit, -, -, Invitrogen Cat#L34976, 1:500
Rat monoclonal anti-LCMV nucleoprotein, -, VL-4, DRFZ Inhouse Cat#-, 1:1000
Goat polyclonal anti-Rat IgG, HRP, -, Jackson ImmunoResearch Cat#112-035-003, 1:750
Faser Kit PE, PE, -, Miltenyi Biotec Cat#130-091-764, -
Faser Kit APC, APC, -, Miltenyi Biotec Cat#130-091-762, -
Strepdavidin, PE, -, BD Cat#554061, 1:400

Antibodies used for in vitro cultures (Antibody, Clone, Provider and #Cat, Concentration):
Hamster monoclonal anti-mouse CD3e, 145-2C11, eBioscience Cat#16-0031-86, 2.5 µg/ml
Hamster monoclonal anti-mouse CD28, 37.51, eBioscience Cat#12-0281-82, 2.5 µg/ml
Rat monoclonal anti-mouse IL-4, 11B11, DRFZ Inhouse Cat#-, 10 µg/ml
Rat monoclonal anti-mouse IFN-g, XMG1.2, DRFZ Inhouse Cat#-, 10 µg/ml
Rat monoclonal anti-mouse IL-12, C18.2, DRFZ Inhouse Cat#-, 10 µg/ml
Mouse monoclonal anti-human CD3e, OKT3, Miltenyi Biotec Cat#130-093-387, 2.5 µg/ml
Mouse monoclonal anti-human CD28, 15E8, Miltenyi Biotec Cat#130-093-386, 2.5 µg/ml
Mouse monoclonal anti-human IFN-g, 45-15, Miltenyi Biotec Cat#130-095-743, 10 µg/ml
Mouse monoclonal anti-human IL-4, 7A3-3, Miltenyi Biotec Cat#130-095-753, 10 µg/ml
Mouse monoclonal anti-human IL-12, C8.6, Miltenyi Biotec Cat#130-095-755, 10 µg/ml

| Validation | Commercially available antibodies have been validated by the manufacturers:

-Biolegend: https://www.biolegend.com/nl-be/quality/quality-control

Flow Cytometry Reagents
Specificity testing of 1-3 target cell types with either single- or multi-color analysis (including positive and negative cell types).
Once specificity is confirmed, each new lot must perform with similar intensity to the in-date reference lot. Brightness (MFI) is evaluated from both positive and negative populations.
Each lot product is validated by QC testing with a series of titration dilutions.

TotalSeq™ Antibodies
Bulk lots are tested by PCR and sequencing to confirm the oligonucleotide barcodes. They are also tested by flow cytometry to ensure the antibodies recognize the proper cell populations.
Bottled lots are tested by PCR and sequencing to confirm the oligonucleotide barcodes.

- eBioscience: https://www.thermofisher.com/de/de/home/life-science/antibodies/invitrogen-antibody-validation.html

Part 1—Target specificity verification
This helps ensure the antibody will bind to the correct target. Our antibodies are being tested using at least one of the following methods to ensure proper functionality in researcher's experiments:
Knockout—expression testing using CRISPR-Cas9 cell models
Knockdown—expression testing using RNAi to knockdown gene of interest
Independent antibody verification (IAV)—measurement of target expression is performed using two differentially raised antibodies recognizing the same protein target
Cell treatment—detecting downstream events following cell treatment
Relative expression—using naturally occurring variable expression to confirm specificity
Neutralization—functional blocking of protein activity by antibody binding
Peptide array—using arrays to test reactivity against known protein modifications
SNAP-ChIP™—using SNAP-ChIP to test reactivity against known protein modifications
Immunoprecipitation-Mass Spectrometry (IP-MS)—testing using immunoprecipitation followed by mass spectrometry to identify antibody targets

Part 2—Functional application validation
These tests help ensure the antibody works in a particular application(s) of interest, which may include
(but are not limited to):
Western blotting
Flow cytometry
ChIP
Immunofluorescence imaging
Immunohistochemistry

Most antibodies were developed with specific applications in mind. Testing that an antibody generates acceptable results in a specific application is the second part of confirming antibody performance.

- Miltenyi Biotec: https://www.miltenyibiotec.com/DE-en/products/macs-antibodies/antibody-validation.html
During development of an antibody, a suitable test to verify specificity of the clone is performed. Several approaches are possible (Counterstaining, Knockout of protein, Epitope competition assay, siRNA knockdown of protein, Stimulation of cells, Overexpression of target protein, Binding to purified antigen)

-BD Biosciences: https://www.biocompare.com/Antibody-Manufacturing/355107-Antibody-Manufacturing-Perspectives-BD-Bioscience/
We conduct quality control (QC) testing in primary model systems to ensure biological accuracy in an ISO 9001 certified facility. BD conducts rigorous QC testing of each antibody lot tested side-by-side with a previously produced lot as reference. Our product development process includes testing on a combination of primary cells, cell lines and/or transfectant cell models with relevant controls using multiple immunoassays to ensure biological accuracy. We also perform multiplexing with additional antibodies to interrogate antibody staining in multiple cell populations.

Further, several antibodies have been validated in-house by using knockout mice, unstimulated cells, or isotype antibodies as controls. Antibodies used for determination of viral titers in stock solutions were verified using virus-free samples as controls. |
|---|---|

# Eukaryotic cell lines

Policy information about cell lines

| Cell line source(s) | HEK293T, MC57G, and L929 cells were obtained at ATCC. NIH3T3 fibroblasts were kindly provided by the Max Plank Institute for Infection Biology (Berlin) and originally obtained at ATCC. BHK-21 cells were obtained at ECACC. |
|---|---|
| Authentication | None of the cell lines used were additionally authenticated. |
| Mycoplasma contamination | Cell lines were not tested for mycoplasma contatmination. |

| Commonly misidentified lines | HEK293T, MC57G, L929, NIH3T3, and BHK-21 cells are not commonly misidentified cell lines. |
|---|---|
| (See ICLAC register) | |

## Animals and other organisms

Policy information about studies involving animals; ARRIVE guidelines recommended for reporting animal research

| Laboratory animals | C57BL/6J mice (wildtype), LCMV-TCRtg P14 (Pircher et al. 1989) and Smarta mice (Oxenius et al. 1998) expressing the congenic markers CD45.1 (Ly5.1) or CD90.1 (Thy1.1), Il1rl1-/- (Townsend et al. 2000), Il1rl1-ExAB-/-, Il1rl1-ExC-/-, Stat4-/- (Kaplan et al. 1996), Tbx21-/- (Szabo et al. 2002), Smarta x Il1rl1-ExAB-/-, Smarta x Stat4-/-, Smarta x Tbx21-/-, P14 x Il1rl1-/-, and P14 x Il1rl1-ExAB-/- and TCRbd-/-(Mombaerts et al. 1992) mice were bred under specific-pathogen free conditions in approved animal-care facilities at the Research Institute for Experimental Medicine (FEM) of the Charité – Universitätsmedizin Berlin (Berlin, Germany) or at the Laboratory Animal Facility of the ETH Zürich (ETH Phenomics Center, Zürich, Switzerland). Mice were housed in individually ventilated cages with a 12 h light/dark cycle at an ambient temperature of 21°C and 45% to 65% relative humidity. Mice had ad libitum access to drinking water and chow. Both, male and female mice between 8 and 26 weeks of age were used for experiments. For LCMV infections, experimental groups were age- and sex-matched. Mice used for scRNA-Seq analyses were cohoused for 4 weeks prior to LCMV infection. Il1rl1-ExAB-/- and Il1rl1 ExC-/- mice were generated in the Transgenics Core Facility of the Max Delbrück Centrum Berlin. |
|---|---|
| Wild animals | The study did not use wild animals |
| Field-collected samples | The study did not include field-collected samples |
| Ethics oversight | Animal experiments were performed in accordance with institutional guidelines and the German and Swiss law for animal protection and were approved by the respective governmental authorities (Landesamt für Gesundheit und Soziales Berlin and the Cantonal Veterinary Office of the Canton of Basel; G0111/17, G0206/17, G0245/19). |

Note that full information on the approval of the study protocol must also be provided in the manuscript.

## Human research participants

Policy information about studies involving human research participants

| Population characteristics | Due to German Data Privacy laws, no data such as gender or age are legally available for healthy blood donors from the German Red Cross. No influence of gender or age on the readout was to be expected. |
|---|---|
| Recruitment | Human peripheral blood of random healthy donors was obtained from the German Red Cross (DRK Berlin, Germany). |
| Ethics oversight | The Charité ethics commitee approved the study (EA1/149/12). |

Note that full information on the approval of the study protocol must also be provided in the manuscript.

## Flow Cytometry

### Plots

Confirm that:

☒ The axis labels state the marker and fluorochrome used (e.g. CD4-FITC).

☒ The axis scales are clearly visible. Include numbers along axes only for bottom left plot of group (a 'group' is an analysis of identical markers).

☒ All plots are contour plots with outliers or pseudocolor plots.

☒ A numerical value for number of cells or percentage (with statistics) is provided.

### Methodology

| Sample preparation | To isolate lymphocytes, spleens were mechanically disrupted and filtered through 70 μm strainers. Erythrocytes were lysed by 3-5 min of incubation in erythrocyte lysis buffer (10 mM KHCO3, 155 mM NH4Cl, 0.1 mM EDTA, pH 7.5). Livers were collected in PBS/BSA, meshed and centrifugated at 30 g for 2 min to remove debris. Supernatants were subjected to Histopaque density centrifugation (1.083 g/ml, Sigma-Aldrich) and lymphocytes were collected at the gradient interphase. To stain ILC2s, lungs were cut into small pieces and digested with Collagenase D (0.1 U/ml) in RPMI1640 (supplemented with 10% FCS and 15mM HEPES) for 1 h at 37°C. Afterwards, lymphocytes were isolated by Histopaque density centrifugation (1.083 g/ml, Sigma-Aldrich). To isolate peritoneal cavity cells, 5 ml of cold PBS was injected into the peritoneal cavity of euthanized mice. After a brief massage of the peritoneum, cell-containing liquid was collected and subjected to Histopaque density centrifugation (1.083 g/ml, Sigma-Aldrich). For analysis of tissue-resident memory T cells, lungs, kidneys, and salivary glands were cut into pieces and digested in RPMI1640 + GlutaMax I (Thermo Scientific) medium containing FCS (5% v/v, Thermo Scientific), MgCl2 (2 μM, Carl Roth), CaCl2 (2 μM, Carl Roth), and collagenase type I (100 U/ml, Gibco) at 37°C for 45 min. Subsequently, tissue was further disrupted using a GentleMACS Dissociator (setting m_Spleen_01.01). Cells were filtered through 70 μm strainers, subjected to erythrocyte lysis, and analyzed. |
|---|---|
| Instrument | FACS Canto, LSR Fortessa (BD) and MACSQuant (Miltenyi Biotec) were used for data collection Aria I and Aria II (BD) devices were used for cell sorting. |

| Software | BD Diva software (v8.1), FlowJo (v10.7.1) |
|---|---|
| Cell population abundance | Frequencies of analyzed cell populations are indicated in representative plots. Purity of sorts are shown in representative plots in the Supplementary Information. |
| Gating strategy | Lymphocytes were defined using FSC-A and SSC-A and singlets were further selected using FSC-W vs. FSC-H and SSC-W vs. SSC-H or FSC-A vs. FSC-H. For cell sorting, dead cells were labelled with propidium iodide. For analysis of surface markers and intracellular stainings, dead cells were labelled with Zombie Aqua or Zombie NIR fixable live/dead staining reagents (Biolegend). Further gatings were defined based on bimodal expression of surface markers. Gating strategies are indicated in Figure Legends and representative plots showing gating strategies are provided in the Supplementary Information. |

☒ Tick this box to confirm that a figure exemplifying the gating strategy is provided in the Supplementary Information.

