## [Peer Review File · Nature Immunology]

Peer Review Information

Journal: Nature Immunology

Manuscript Title: A type-1 immunity-restricted promoter of the IL-33 receptor gene directs antiviral T cell responses

Corresponding author name(s): Professor Max Loehning, Dr Tobias Brunner

Reviewer Comments & Decisions:

Decision Letter, initial version:
--

23rd Feb 2023

Dear Professor Löhning,

Thank you for sharing your point-by-point reply to referees comments on your manuscript "A type-1 immunity-restricted promoter of the IL-33 receptor gene directs antiviral T cell responses". We would be interested in considering a revised version as outlined in your response.

We hope you will find the referees' comments useful as you decide how to proceed. If you wish to submit a substantially revised manuscript, please bear in mind that we will be reluctant to approach the referees again in the absence of major revisions. Please do not hesitate to get in touch if you would like to discuss these issues further

If you choose to revise your manuscript taking into account all reviewer and editor comments, please highlight all changes in the manuscript text file [OPTIONAL: in Microsoft Word format].

* If you have not done so already please begin to revise your manuscript so that it conforms to our Article format instructions at <http://www.nature.com/ni/authors/index.html>. Refer also to any

guidelines provided in this letter.

The Reporting Summary can be found here:

[REDACTED]

If you wish to submit a suitably revised manuscript we would hope to receive it within 6 months. If you cannot send it within this time, please let us know. We will be happy to consider your revision so long as nothing similar has been accepted for publication at Nature Immunology or published elsewhere.

Nature Immunology is committed to improving transparency in authorship. As part of our efforts in this direction, we are now requesting that all authors identified as 'corresponding author' on published papers create and link their Open Researcher and Contributor Identifier (ORCID) with their account on the Manuscript Tracking System (MTS), prior to acceptance. ORCID helps the scientific community achieve unambiguous attribution of all scholarly contributions. You can create and link your ORCID from the home page of the MTS by clicking on 'Modify my Springer Nature account'. For more information please visit www.springernature.com/orcid.

Thank you for the opportunity to review your work.

Sincerely,

Stephanie Houston
Editor
Nature Immunology

Reviewers' Comments:

Reviewer #1:

Remarks to the Author:

The authors identified a novel alternative TSS for *Il1rl1* gene that is specifically utilized by Th1 and CTL cells. Loss of this promoter region significantly impaired CTL differentiation in to SLEC resulting in deficient response to LCMV. The study is elegantly designed and conducted, and only a few minor issues need to be addressed.

1. In figure 2, the authors examined the conservation of type I *Il1rl1* promoter in humans and mice. Based on figure 2a, only the Tbet and STAT4 binding motifs in CNS5 are conserved and the conservation score for the actual promoter region is low. Therefore, only the usage of this promoter is conserved, but not the promoter itself. The section title "The type-1 *Il1rl1* promoter is conserved between mice and humans" is thus misleading and needs to be revised.
2. There seem to be an enhancer located very close to, but downstream of, the promoter region. This enhancer is also bound by T-bet and STAT4 in Th1 cells and could potentially be important for ST2 expression driven by T-bet and STAT4. Thus, the authors should demonstrate whether this enhancer is impacted by the ExAB deletion. In fact, the sequence alignment of this promoter region and surrounding area between mouse and human should be provided, and the area of deletion indicated.
3. In Figure 3k, loss of Type 1 promoter region seems to result only in partial loss of ST2 expression in Th1 cells. What is the explanation of this phenotype? Is there another yet undiscovered TSS for ST2 that is specific for Th1? Or are Th1 cells capable of utilizing GATA3-dependent type II promoters? The authors should comment on this.
4. In figure 4k and o. There seems to be a CD45.2+ CD45.1+ population. Yet, according to the method section, CD45.1+/- and CD45.2+/+ BM/T cells were used. Where did this double positive fraction is coming from? Or is there an error in the method section?
5. Figure 4f showed reduced Ki67+ cells in ExAB-/- mice, however, there seems to be a larger fraction of cycling cells in ExAB -/- mice by scRNA-seq as shown in figure 5b. What's causing this discrepancy?
6. Instead of a simple lack of SLEC expansion induced by the loss of type I promoter region, there seems to be a diversion from SLEC fate to MPEC fate in ExAB-/- mice, as evidenced by the significantly increased MPEC cluster shown in Figure 5b. The authors should conduct a LCMV re-challenge experiment to explore whether there is a memory-related phenotype in these mice.
7. A heatmap of differential gene expression between WT and ExAB-/- cells from each cluster should be shown to demonstrate the transcriptional impact resulting from loss of the type I promoter. The merged heatmap showing all only the top 10 marker for each cluster is not sufficient.
8. According to Figure 6b IL-33 induces *Zeb2* expression in all T cell subsets examined. *Zeb2* expression is required for SLEC differentiation. The phenotype of T cells specific loss of *Zeb2* during LCMV infection (Dominguez et al JEM 2015 and Omilusik et al JEM 2015) strongly resembles the phenotype observed in ExAB-/- mice. The authors should examine whether the effect on CD8 T cells during LCMV infection after the loss of Type I promoter is mediated by *Zeb2*.

Reviewer #2:

Remarks to the Author:

In this manuscript, Brunner et al. report the discovery and functional characterization of a type-1-immunity-restricted promoter of the gene *Il1rl1*, encoding the ST2 receptor for interleukin-33 (IL-33), a critical regulator of type-1, type-2 and regulatory immune responses. The authors demonstrate that this novel promoter is located 40 kb upstream of the annotated *Il1rl1* gene and conserved between mice and humans. In previous work, the authors reported that ST2 expression is upregulated on CD8 T cells (CTLs) and CD4 T cells (Th1s) during viral infection (LCMV model). Now, they demonstrate that this upregulation of ST2 on CTLs and Th1s does not occur during viral infection in mice with deletion of the type-1 *Il1rl1* promoter. They convincingly show that the expression of ST2 driven by the type-1 promoter is critical for clonal expansion of short-lived effector T cells during antiviral responses. They propose that IL-33/ST2 signalling provides a strong costimulatory signal for amplification of antiviral responses mediated by CTLs.

The authors performed a significant number of experiments and the results are novel, interesting and potentially important. Recent results in a Phase 2 clinical trial of COPD revealed promising efficacy of anti-IL-33 therapeutic antibodies, indicating that IL-33 plays important roles in type-1 inflammatory diseases in humans. A better understanding of the mechanisms implicated in the activation of type-1 immune responses by IL-33 is thus urgent.

However, at this point, the data are not sufficient to support the title and conclusions of the manuscript: "A type-1 immunity-restricted promoter of the IL-33 receptor gene...". The authors analysed Th1s and CTLs but did not analyse the potential role of the type-1 promoter in other immune cells that express ST2 during type-1 immune responses. These include NK and iNKT cells, macrophages, dendritic cells and neutrophils. To increase the significance of their findings, it is thus essential that the authors analyse the potential role of the type 1-promoter in the regulation of ST2 expression in other immune cell types, associated with type-1 immune responses.

Is there a general role of the type 1-promoter in all type-1 immune cells or a specific role in T cells? This is an important question because the identification of the type-1 immunity-restricted promoter of *Il1rl1* is the main message of the paper.

The authors have all the tools in hands, including mice knock-out for the type-1 *Il1rl1* promoter (or immune cells derived from these mice) and a validated flow cytometry assay for ST2 (with DJ8 mAb), to answer this critical question using ex vivo or in vivo assays (see Major Points).

MAJOR POINTS

1) Role of the type-1 promoter in NK and iNKT cells: both human and mouse iNKT and NK cells are direct targets of IL-33, and produce high levels of IFN γ after co-stimulation with IL-33 and IL-12 (1-3). NK cells are an important source of IFN γ in viral infection and cancer. The authors should determine whether the type-1 promoter controls ST2 expression and/or IL-33 responsiveness in NK cells and iNKT cells (ex vivo or in vivo assays).

2) Role of the type-1 promoter in macrophages: myeloid-derived antigen-presenting cells, including macrophages and DCs, can respond to IL-33. The authors should determine whether the type-1 promoter controls ST2 expression and/or IL-33 responsiveness in macrophages. For instance, they could use ex vivo assays with bone marrow-derived macrophages, as previously described (3, 4).

3) Role of the type-1 promoter in neutrophils: resting bone marrow neutrophils express low levels of

ST2 mRNA, and ST2 protein is upregulated at the cell surface after IL-33 treatment (5). The authors should determine whether the type-1 promoter controls ST2 expression and/or IL-33 responsiveness in neutrophils.

4) Detection of ST2 in immune cells and differences in expression levels: quantification of ST2 by flow cytometry is a critical method for this study. However, the authors used different mAb clones for the detection of ST2 in Th1 cells/CTLs (clone DJ8, with PE- or APC-FASER amplification; Fig 1b, 3e) and ILC2s/mast cells (clone DIH9, ST2-BV421; Ext Fig 4c, f). Commercial ST2 antibodies exhibit significant background staining and are best for highly expressing cells. Could it be that the background with clone DIH9 (ST2-BV421 direct fluorescent conjugate) was too high and the specific signal without amplification too low for detection of ST2 expression in Th1/CTLs? The authors should comment about the differences in ST2 expression levels in different immune cells and explain why they developed a more sensitive assay with DJ8 mAb for Th1/CTLs. In the extended data figures, they should compare the different ST2 mAbs on the same cell types (both type-1 and type-2 immune cells) using Irl1 KO cells as controls for gating. RMST2-2 is another widely used mAb clone. The authors have all the tools in hands (various Irl1 KO cells) to perform a careful comparison of ST2 detection with the three clones (DJ8, DIH9, RMST2-2) on different type-1 and type-2 immune cells. This will be very useful for the scientific community and will increase the interest and impact of the article, beyond the field of T cells.

OTHER POINTS

5) Fig. 2a and p6: it is not clear why the authors do not comment the T-bet/STAT4 ChIP-Seq and CD8+ SLEC (abbreviation not defined in the legend) ATAC-seq signals downstream of exons A + B? No T-bet or STAT4 binding sites? Sites not conserved in humans?

6) Fig 2c: the position of the CNS-5 in the human gene is not clear. ChIP-seq data/binding sites for STAT4?

7) p7 'However, due to hard-to-predict effects on the balance between IL-33-mediated inflammation and tissue repair, blockade of IL-33 or ST2 using therapeutic antibodies has shown conflicting results in preclinical disease models⁴⁷'. This sentence is outdated. Blockade of IL-33 using therapeutic antibodies has shown encouraging efficacy in Phase 2 clinical trials of asthma and COPD, as recently reviewed (6). A more appropriate sentence/paragraph could be 'Modulation of the IL-33–ST2 axis could represent a promising approach in treating inflammatory diseases. Blockade of IL-33 using therapeutic antibodies has shown encouraging efficacy in clinical trials of asthma and COPD (6). However, due to hard-to-predict effects on the balance between IL-33-mediated inflammation and tissue repair, fine-tuned treatment approaches might offer a critical advantage.'

References

1. Smithgall MD, et al. (2008) IL-33 amplifies both Th1- and Th2-type responses through its activity on human basophils, allergen-reactive Th2 cells, iNKT and NK cells. *Int Immunol* 20(8):1019-1030.
2. Bourgeois E, et al. (2009) The pro-Th2 cytokine IL-33 directly interacts with invariant NKT and NK cells to induce IFN-gamma production. *Eur J Immunol* 39:1046-1055.
3. Kearley J, et al. (2015) Cigarette smoke silences innate lymphoid cell function and facilitates an exacerbated type I interleukin-33-dependent response to infection. *Immunity* 42(3):566-579.

4. Dagher R, et al. (2020) IL-33-ST2 axis regulates myeloid cell differentiation and activation enabling effective club cell regeneration. *Nat Commun* 11(1):4786.
5. Alves-Filho JC, et al. (2010) Interleukin-33 attenuates sepsis by enhancing neutrophil influx to the site of infection. *Nat Med* 16(6):708-712.
6. Cayrol C & Girard JP (2022) Interleukin-33 (IL-33): A critical review of its biology and the mechanisms involved in its release as a potent extracellular cytokine. *Cytokine* 156:155891.

Reviewer #3:

Remarks to the Author:

The authors characterize a distinct transcriptional process for expression of the gene (*Il1rl1*) encoding ST2 (the IL-33 receptor) in type-1 (CTL, Th1) compared to type-2 (Th2) T cells. They go on to show that ablation of the exons used in the "type-1" transcripts of *Il1rl1* leads to selective loss of ST2 on those cells and impairs the generation of type-1 effector cells following viral infection. They also find that IL-33 promotes expression of genes which support TCR and costimulatory stimulation, and which are associated with effector cell differentiation.

The report carefully lays out the methodical way in which the authors identify different upstream non-coding exons that are selectively used for *Il1rl1* transcripts in CTL/Th1 vs Th2 cells, and present strong evidence that this accurately distinguishes the basis for ST2 expression by type-1 vs type-2 T cells (with Treg aligning with Th2, in this case). Their generation of mice lacking the "type-1" related exons confirmed this, with selective loss of ST2 on Th1 and CTL, and their evidence is compelling that the impact of this mutation almost phenocopies the impact of complete *Il1rl1* gene ablation, with respect to type-1 responses.

The data are convincing, and – as the authors point out – this appears to be a rare (if not unique) example in which genes that are expressed by both type-1 and -2 polarized T cells involve distinct transcriptional start sites (regulated by distinct transcription factors). At the same time, once the authors find that loss of exons A/B (*ExA/B*) leads to almost complete loss of ST2 on type-1 cells, much of the subsequent data corresponds with earlier work, showing a cell-intrinsic role for ST2 in the anti-viral response of these cells. This limitation is somewhat balanced by the authors going deeper than previous studies with respect to the gene expression changes driven by IL-33 in type-1 and -2 T cells (Fig. 6), and by their data indicating that IL-33 stimulation specifically enhances effector cell differentiation (Fig. 5). Nevertheless, there are remaining concerns.

1) The studies on impaired generation of short-lived effector cells (SLEC) by CD8+ T cells lacking *ExA/B* are convincing but the consequences for memory differentiation are underdeveloped. Numerous panels in Figs. 4 and 5 present the frequencies and numbers of SLEC phenotype cells, but do not show the corresponding data for the MPEC (KLRG1- CD127+) cells – based on panels such Fig. 5e, one would presume these cells are increased in frequency. It is unclear whether MPEC are unchanged (or, perhaps more likely, modestly reduced) in absolute numbers. These data are needed for a more complete picture of how defective ST2 regulation (*ExA/B* knockout) or expression (*Il1rl1* knockout) affects CD8+ T cells differentiating along the effector vs memory pathways.

2) Along the same lines, the authors' analysis of the anti-viral response of *ExA/B* and *Il1rl1* deficient T cells appears to be limited to the first ~10 days following infection, and exclusively (as far as this reviewer can ascertain) focuses on cells in the spleen. It would be important to know how mutation of

ExA/B impacts generation of the various subsets of circulating memory CD8+ and CD4+ T cells during these responses, and to extend the scope of the work to analysis of at least some non-lymphoid tissues to determine the effect on generation of resident memory T cells. As it stands, it is unclear whether the loss of normal ST2 regulation leads to a selective loss of type-1 effector differentiation – with, perhaps, minimal effect on establishment of long-lived memory – or a more generalized defect that is magnified in but not exclusive to the effector pool.

Author Rebuttal to Initial comments

See inserted PDF

Point-by-point Reply to Reviewers' Comments:

A type-1 immunity-restricted promoter of the IL-33 receptor gene directs antiviral T cell responses (NI-A35288-T)

We are grateful for the valuable input and feedback provided by the reviewers regarding our work. We have thoroughly addressed each of these comments and suggestions. Below, please will find our responses to the reviewer comments (Blue text), presenting the additional experiments performed and outlining the specific changes we have implemented in our manuscript.

Within the manuscript text, we have used Blue underlined text to indicate changes made to the original version as a result of the reviewer comments.

Reviewer #1

(Remarks to the Author)

The authors identified a novel alternative TSS for *Il1r1* gene that is specifically utilized by Th1 and CTL cells. Loss of this promoter region significantly impaired CTL differentiation in to SLEC resulting in deficient response to LCMV. The study is elegantly designed and conducted, and only a few minor issues need to be addressed.

We thank the reviewer for this very favorable feedback and the helpful comments. We have addressed all the minor points raised as outlined below.

1. In figure 2, the authors examined the conservation of type I *Il1r1* promoter in humans and mice. Based on figure 2a, only the T-bet and STAT4 binding motifs in CNS5 are conserved and the conservation score for the actual promoter region is low. Therefore, only the usage of this promoter is conserved, but not the promoter itself. The section title "The type-1 *Il1r1* promoter is conserved between mice and humans" is thus misleading and needs to be revised.

We fully agree with the reviewer and have amended the section title as follows: "Usage of the type-1 *Il1r1* promoter is conserved between mice and humans." (p. 6, l. 148)

2. There seem to be an enhancer located very close to, but downstream of, the promoter region. This enhancer is also bound by T-bet and STAT4 in Th1 cells and could potentially be important for ST2 expression driven by T-bet and STAT4. Thus, the authors should demonstrate whether this enhancer is impacted by the ExAB deletion. In fact, the sequence alignment of this promoter region and surrounding area between mouse and human should be provided, and the area of deletion indicated.

We concur that there is a region downstream of the identified exons A and B, which is bound by STAT4 and T-bet, and is marked by an ATAC-Seq peak in activated but not naive CD8⁺ T cells. Importantly, this putative enhancer specified by the reviewer is not deleted in the generated *Il1r1-ExAB*^{-/-} mice. We have provided a new subpanel (Extended Data Fig. 3b) to indicate the area of deletions in *Il1r1-ExAB*^{-/-} and *Il1r1-ExC*^{-/-} mice in relation to the ATAC- and ChIP-Seq peaks. An interpretable single-nucleotide sequence alignment of the promoter region, including all important sites, would exceed by far the size limit of a figure (multiple thousand base-pairs). To offer the reviewer and the readership a readily accessible way to investigate the data presented in the manuscript, we have provided access to the RNA-Seq, ATAC-Seq, and ChIP-Seq tracks in a UCSC Genome browser session. Using the following link: <https://genome.ucsc.edu/s/agloehning/BrunnerServeetal2023>, reviewers and readers can inspect the data in detail and compare the conservation of all positions at the desired resolution. We hope that these measures support the claims made in our study by enhancing the data transparency of our manuscript.

3. In Figure 3k, loss of Type 1 promoter region seems to result only in partial loss of ST2 expression in Th1 cells. What is the explanation of this phenotype? Is there another yet undiscovered TSS for ST2 that is specific for Th1? Or are Th1 cells capable of utilizing GATA3-dependent type II promoters? The authors should comment on this.

We thank the reviewer for addressing this highly interesting point. Indeed, while the loss of exons A and B largely abrogates ST2 expression of CTLs *in vitro* and *in vivo*, Th1 cells appear to be able to partially compensate for the loss of the type-1 promoter *in vivo*, as we observed a mere ~50% reduction in frequency of ST2⁺ Th1 cells in *Il1rl1-ExAB*^{-/-} mice. To address the question which promoter is utilized by *ExonAB*-deficient Th1 cells, we adoptively transferred naive Smarta or Smarta *Il1rl1-ExAB*^{-/-} cells into wildtype (WT) recipients, which were subsequently infected with LCMV (Extended Data Fig. 6s). Like the polyclonal Th1 population in *Il1rl1-ExAB*^{-/-} mice, also Smarta T cells displayed a ~50-60% reduction in frequencies of ST2-expressing cells at the peak of the acute T cell response (Extended Data Fig. 6t,u). Congenitally marked Th1 cells were flow-cytometrically sorted from spleens of infected animals, and RNA was isolated to assess their ST2 promoter usage. Interestingly, *ExonAB*-deficient, but not wildtype Smarta cells utilized the proximal promoter (reported to be used by e.g. fibroblasts) to express ST2, and thus incorporated exon 1b into their 5' untranslated regions (Extended Data Fig. 6v). Neither WT, nor *Il1rl1-ExAB*^{-/-} Smarta cells expressed significant amounts of exon 1a. Importantly, exon 1b was barely detectable in 2 out of 4 recipients of WT Smarta cells, indicating that it does not critically contribute to ST2 expression if the type-1 promoter is intact. We have incorporated these data into the manuscript (Extended Data Fig. 6s-v) (p. 10 l. 260-262). The molecular mechanisms underlying this finding will be addressed in future studies using the appropriate gene-targeted mice.

4. In figure 4k and o. There seems to be a CD45.2⁺ CD45.1⁺ population. Yet, according to the method section, CD45.1^{+/-} and CD45.2^{+/+} BM/T cells were used. Where did this double positive fraction is coming from? Or is there an error in the method section?

Indeed, in the experiment presented in Fig. 4j-m, irradiated CD45.1^{+/+} recipients were reconstituted with bone marrow from CD45.1^{+/-} (WT) mice and CD45.2^{+/+} (*Il1rl1*^{-/-}, *Il1rl1-ExAB*^{-/-}, or *Il1rl1-ExC*^{-/-}) mice. In Fig. 4n-r, CD45.1^{+/-} P14 cells were transferred together with CD45.1^{+/+} *Il1rl1-ExAB*^{-/-} or *Il1rl1*^{-/-} P14 cells into CD45.2^{+/+} WT recipients. Importantly, CD45.1 and CD45.2 are two variants of the CD45 gene differing in only few amino acids. Thus CD45.1^{+/-} cells express CD45.1 from one allele and CD45.2 from the second allele, which results in a double positive population if co-stained with CD45.1- and CD45.2-specific antibodies. Thus, the double positive fraction represents the reconstituted WT donor cells in Fig. 4k and the transferred P14 cells in Fig. 4o.

To make it easier to follow these experimental layouts, we amended the figure legends to indicate the expression of both variants (e.g. CD45.1⁺ CD45.2⁺ instead of CD45.1^{+/-}).

5. Figure 4f showed reduced Ki67⁺ cells in *ExAB*^{-/-} mice, however, there seems to be a larger fraction of cycling cells in *ExAB*^{-/-} mice by scRNA-seq as shown in figure 5b. What's causing this discrepancy?

In Fig. 4f, the frequency of Ki67-expressing cells among all CD8⁺ T cells is depicted. The reduced frequency of Ki67⁺ cells among CD8⁺ T cells in *Il1rl1-ExAB*^{-/-} mice is a result of the impaired expansion of activated CTLs, a large fraction of which is positive for Ki67. In contrast, in Fig. 5b, activated CD44⁺ CTLs were sorted and equal amounts of these CTLs per genotype were compared, thus reflecting the amount of cycling cells relative to an equal number of activated CD44⁺ CTLs. However, as Fig. 5n illustrates, there is still a drastic difference in the numbers of CD44⁺ CTLs per spleen in the scRNA-Seq experiment. When taking this into account, the absolute number of Ki67⁺ CTLs in WT mice far exceeds the number of Ki67⁺ CTLs in *Il1rl1-ExAB*^{-/-} mice.

6. Instead of a simple lack of SLEC expansion induced by the loss of type I promoter region, there seems to be a diversion from SLEC fate to MPEC fate in *ExAB*^{-/-} mice, as evidenced by the significantly increased MPEC cluster shown in Figure 5b. The authors should conduct a LCMV re-challenge experiment to explore whether there is a memory-related phenotype in these mice.

As pointed out in the answer to the previous question, in Fig. 5b equal numbers of activated CTLs per genotype were compared, irrespective of the differences in CTL expansion between the two

genotypes. Consequently, the large relative increase in MPECs is in part due to a >10-fold decrease in the amount of SLECs per organ. However, in absolute terms, the amount of MPECs per spleen is even slightly decreased in *Il1r1-ExAB^{-/-}* and *Il1r1^{-/-}* mice. We have amended the text and have added the absolute numbers of MPECs to address this point (Fig. 5h,i; Fig. 6j,k and Extended Data Fig. 6m)(p. 10-11, l. 290-306; p. 13, l. 368-370; p. 16, l. 450-454).

To address the reviewers' question whether *Il1r1-ExAB^{-/-}* or *Il1r1^{-/-}* mice display a memory-related phenotype, we have conducted a re-challenge experiment. We first infected WT, *Il1r1^{-/-}*, *Il1r1-ExAB^{-/-}*, or *Il1r1-ExC^{-/-}* mice with 200 PFU of LCMV-WE. Then, 30 days after primary infection, mice were challenged with 2×10^6 PFU of LCMV-C113 and analyzed 7 days after secondary infection (Reviewer Fig. 1a). We have found no significant differences in frequencies or absolute numbers of CD8⁺ T cells in spleens of re-challenged animals (Reviewer Fig. 1b,c). Further, no differences in frequencies of activated CD44⁺ CD62L⁻ CTLs or counts of LCMV-specific NP396-Tetramer⁺ CTLs were observed (Reviewer Fig. 1d,e). Lastly, in this recall setting the proportion of short-lived effector cells amongst CTLs of *Il1r1^{-/-}*, *Il1r1-ExAB^{-/-}* and *Il1r1-ExC^{-/-}* as well as their number was comparable to WT mice (Reviewer Fig. 1 f,g).

Reviewer Figure 1. IL-33-ST2 signaling is not required for CTL expansion and SLEC differentiation in LCMV re-challenged mice. Wildtype, *Il1r1^{-/-}*, *Il1r1-ExAB^{-/-}*, and *Il1r1-ExC^{-/-}* mice were primed with 200 plaque-forming units (PFU) of LCMV-WE. Mice were re-challenged with 2×10^6 PFU LCMV-C113 on day 30 p.i. and splenic T cells were analyzed 7 days after re-challenge (WT: $n = 6$, *Il1r1^{-/-}*: $n = 6$, *Il1r1-ExAB^{-/-}*: $n = 6$, *Il1r1-ExC^{-/-}*: $n = 5$). **a**, Experimental outline. **b**, Representative staining of CD8 on lymphocytes. **c**, Frequencies and absolute cell counts of CTLs. **d**, Frequencies of activated CD44⁺ CD62L⁻ CTLs. **e**, Absolute cell counts of LCMV NP₃₉₆₋₄₀₄-specific CTLs. **f**, Representative FACS plots showing KLRG1 and CD127 expression by CD8⁺ T cells. **g**, Frequencies and absolute cell counts of KLRG1⁺ CD127⁻ CTLs. Data are presented as mean \pm SD with each dot representing one mouse. P was determined using One-way ANOVA with Tukey's post-hoc tests.

In our opinion, this experiment nicely reflects the role of IL-33 as an alarmin.

Secondary LCMV infections are rapidly controlled by highly potent memory CD8⁺ T cells and additional antibody-dependent antiviral mechanisms¹⁻³. The data presented here show that ST2 signaling is not required for antiviral CD8⁺ T cell expansion and SLEC differentiation when previously infected mice are re-challenged. A similar phenotype was observed in MyD88-deficient mice, which is an adaptor protein essential for ST2 signaling^{4,5}.

However, we have addressed the role of IL-33–ST2 signaling in memory T cell formation, maintenance, and recall responses in Baumann et al.⁶. In the latter study, we found that a lack of IL-33 signaling in primary infections did not affect recall responses in an IL-33-competent environment. Yet, IL-33 signals were essential for efficient expansion and reactivation of CD8⁺ memory T cells when re-challenged in naive secondary WT recipients. These experiments have shown that also fully differentiated memory CTLs can (re-)express ST2 and can benefit from IL-33 signals, likely depending on the inflammatory context. Together, these findings raise interesting questions. For instance, it remains unknown whether IL-33 release differs between LCMV-primed and naive mice. Neither do we know whether the interval between priming and re-challenge affects the IL-33/ST2-dependence of secondary CTL responses.

The entirely novel finding of a type-1 immunity-restricted promoter that regulates ST2 expression in antiviral T cells remains the focus of the current manuscript. As we would like to address the above-mentioned questions in future studies, we decided to present the results of the re-infection experiment to the reviewer without adding it to the manuscript.

7. A heatmap of differential gene expression between WT and ExAB^{-/-} cells from each cluster should be shown to demonstrate the transcriptional impact resulting from loss of the type I promoter. The merged heatmap showing all only the top 10 marker for each cluster is not sufficient.

We have performed a differential gene expression analysis of each cluster to demonstrate the impact of the type-1 *Il1r1* promoter deletion on the transcriptome of individual subsets. The results are now displayed in the new Extended Data Fig. 8.

8. According to Figure 6b IL-33 induces Zeb2 expression in all T cell subsets examined. Zeb2 expression is required for SLEC differentiation. The phenotype of T cells specific loss of Zeb2 during LCMV infection (Dominguez et al JEM 2015 and Omilusik et al JEM 2015) strongly resembles the phenotype observed in ExAB^{-/-} mice. The authors should examine whether the effect on CD8 T cells during LCMV infection after the loss of Type I promoter is mediated by Zeb2.

We fully agree with the reviewer that the phenotype of *Zeb2*-deficient CD8⁺ T cells in LCMV infections is very similar to the one observed in *ExonAB*- or *ST2*-deficient mice and that this resemblance is of considerable interest.

Besides Zeb2, IL-33 stimulation of CTLs also induced expression of T-bet and Blimp1, both of which are critical for expansion and SLEC differentiation of LCMV-specific CTLs⁷. Thus, impaired SLEC differentiation observed in *ExAB*- and *Il1r1*-deficient mice is probably a combination of multiple factors and can possibly not be exclusively attributed to a reduction in Zeb2 expression alone. Interestingly however, analysis of scRNA-Seq data, as suggested by the reviewer in point 7, revealed a higher expression of *Zeb2* transcripts in *ExAB*-deficient SLECs as compared to WT SLECs (Extended Data Fig. 8). This might reflect a compensatory mechanism, which could allow very few cells to differentiate into SLECs without IL-33/ST2 signals, and further indicate an important link between ST2-signaling and Zeb2 expression *in vivo*.

A putative experiment to assess this question in more detail, would be to overexpress *Zeb2* in *ST2*-deficient T cells using transgenic mice or retro/lentiviral constructs and to study the T cell response after LCMV infections. We regret that we don't have the necessary tools at hand to perform these elaborate experiments and we hope that the reviewer agrees that establishment of these tools would exceed the timeframe available for this revision.

All things considered, we are very grateful for the reviewer's suggestions and believe that addressing the minor issues raised by the reviewer significantly improved the quality of our study.

Reviewer #2

(Remarks to the Author)

In this manuscript, Brunner et al. report the discovery and functional characterization of a type-1-immunity-restricted promoter of the gene *Il1r1*, encoding the ST2 receptor for interleukin-33 (IL-33), a critical regulator of type-1, type-2 and regulatory immune responses. The authors demonstrate that this novel promoter is located 40 kb upstream of the annotated *Il1r1* gene and conserved between mice and humans. In previous work, the authors reported that ST2 expression is upregulated on CD8 T cells (CTLs) and CD4 T cells (Th1s) during viral infection (LCMV model). Now, they demonstrate that this upregulation of ST2 on CTLs and Th1s does not occur during viral infection in mice with deletion of the type-1 *Il1r1* promoter. They convincingly show that the expression of ST2 driven by the type-1 promoter is critical for clonal expansion of short-lived effector T cells during antiviral responses. They propose that IL-33/ST2 signaling provides a strong costimulatory signal for amplification of antiviral responses mediated by CTLs.

The authors performed a significant number of experiments and the results are novel, interesting and potentially important. Recent results in a Phase 2 clinical trial of COPD revealed promising efficacy of anti-IL-33 therapeutic antibodies, indicating that IL-33 plays important roles in type-1 inflammatory diseases in humans. A better understanding of the mechanisms implicated in the activation of type-1 immune responses by IL-33 is thus urgent.

However, at this point, the data are not sufficient to support the title and conclusions of the manuscript: "A type-1 immunity-restricted promoter of the IL-33 receptor gene...". The authors analysed Th1s and CTLs but did not analyse the potential role of the type-1 promoter in other immune cells that express ST2 during type-1 immune responses. These include NK and iNKT cells, macrophages, dendritic cells and neutrophils. To increase the significance of their findings, it is thus essential that the authors analyse the potential role of the type 1-promoter in the regulation of ST2 expression in other immune cell types, associated with type-1 immune responses.

Is there a general role of the type 1-promoter in all type-1 immune cells or a specific role in T cells? This is an important question because the identification of the type-1 immunity-restricted promoter of *Il1r1* is the main message of the paper.

The authors have all the tools in hands, including mice knock-out for the type-1 *Il1r1* promoter (or immune cells derived from these mice) and a validated flow cytometry assay for ST2 (with DJ8 mAb), to answer this critical question using ex vivo or in vivo assays (see Major Points).

We thank the reviewer for this positive feedback and are delighted to hear that the reviewer appreciates the impact of our findings. We certainly agree that a better understanding of type-1 immunity-related aspects of IL-33 biology is of utmost importance.

We would like to emphasize that by using the wording "type-1 immunity-restricted", we do not want to imply that the promoter is used by all IL-33-responsive immune cells that contribute to type-1 immune responses. Rather, we use this description to highlight that we have not found any evidence of this promoter being active in type-2 immunity-associated T cells and innate cells, including ILC2s and mast cells nor in Treg cells. We have amended the text at two positions to communicate this more precisely (p. 2, l. 35; p. 14, l. 392).

In addition, we have performed a significant number of experiments and made use of our highly sensitive ST2 staining protocol, to assess if the type-1 *Il1r1* promoter is of importance for ST2 expression by NK cells, NKT cells, neutrophils and BMDMs (see below).

MAJOR POINTS

1) Role of the type-1 promoter in NK and iNKT cells: both human and mouse iNKT and NK cells are direct targets of IL-33, and produce high levels of IFN γ after co-stimulation with IL-33 and IL-12 (1-3). NK cells are an important source of IFN γ in viral infection and cancer. The authors should determine whether the type-1 promoter controls ST2 expression and/or IL-33 responsiveness in NK cells and iNKT cells (ex vivo or in vivo assays).

Indeed, NK cells are crucial players in the cellular immunity against infections and cancer. As cited, others have shown that NK cells can produce IFN- γ in response to combined stimulation with IL-12 and IL-33. To assess if the deletion of the type-1 *Il1r1* promoter controls ST2 expression in NK cells, we thus have flow-cytometrically sorted NKp46⁺ TCR β ⁻ NK cells from spleens of WT, *Il1r1-ExAB*^{-/-},

and *Il1r1*^{-/-} mice and cultured them in IL-33- and IL-12-containing medium. We found that after 2 days of stimulation, a fraction of WT NK cells expressed ST2, while no ST2 expression was found on *Il1r1-ExAB*^{-/-} or *Il1r1*^{-/-} NK cells (Extended Data Fig. 4d-f).

In addition, NKT cells were sort purified from thymocytes by utilizing α -Galactosylceramide-loaded CD1d tetramers. FACS-sorted NKT cells were then activated and expanded *ex vivo* in type-1 conditions. After 6 days of culture, ~60% of WT NKT cells expressed ST2 at considerable levels. In contrast, *Il1r1-ExAB*^{-/-} NKT cells were unable to express ST2 as it was the case in *Il1r1*^{-/-} NKT cells (Extended Data Fig. 4a-c).

Together, these results clearly demonstrate that usage of the type-1 *Il1r1* promoter is not restricted to T cells, but also directs ST2 expression in type-1 polarized innate immune cells. We thank the reviewer for the excellent recommendation to perform these exciting experiments. We have added the results to the manuscript in Extended Data Fig. 4 and modified the text accordingly (p. 4, l. 87; p. 8, l. 206-208; p. 14, l. 392).

2) Role of the type-1 promoter in macrophages: myeloid-derived antigen-presenting cells, including macrophages and DCs, can respond to IL-33. The authors should determine whether the type-1 promoter controls ST2 expression and/or IL-33 responsiveness in macrophages. For instance, they could use *ex vivo* assays with bone marrow-derived macrophages, as previously described (3, 4).

To address this question, we have cultured bone-marrow cells (BMDMs) from wildtype, *Il1r1-ExAB*^{-/-}, and *Il1r1*^{-/-} mice in the presence of recombinant M-CSF for 7 days as described in Kearley et al.⁸. As shown in Reviewer Fig. 2a, we achieved a high purity of BMDMs expressing F4/80 and CD11b. After 7 days, BMDMs were stimulated with IL-33, IL-4, IL-12 + IL-18, IFN α + IFN γ , Poly(I:C), or Poly(I:C) + IL-33. We stained ST2 using our amplified staining protocol, as this provides a more direct readout as compared to e.g. ELISAs of the cell culture supernatant.

Unfortunately, we were not able to detect ST2 on the surface of BMDMs irrespective of the genotype and the nature of stimulation (Reviewer Fig. 2b). Analysis of mRNA expression verified that ST2 transcript expression was very low in all culture conditions and substantially lower than in CTLs or Th2 cells (Reviewer Fig. 2c). In line with these results, IL-33 stimulation of BMDMs did not upregulate TNF expression or enhance Poly(I:C)-induced expression of TNF (Reviewer Fig. 2d).

Reviewer Figure 2. ST2 expression on bone-marrow-derived macrophages. Bone marrow cells were isolated from femurs and tibias of wildtype, *Il1r1*^{-/-}, or *Il1r1-ExAB*^{-/-} mice and differentiated in the

presence of recombinant M-CSF (20 ng/ml) for 7 days. **a**, Pre-gating strategy and representative FACS plot showing F4/80 and CD11b expression by differentiated BMDMs. **b**, Representative FACS plots showing no detectable expression of ST2 on BMDMs ($n = 3$ per genotype, experiment was performed twice). **c,d**, mRNA expression of ST2 (**c**) or TNF (**d**) by BMDMs after 24 h of stimulation with IL-33 (10 ng/ml), IL-4 (10 ng/ml), IL-12 + IL-18 (10 ng/ml each), IFN α (250 U/ml) + IFN- γ (10 ng/ml), Poly(I:C) (10 μ g/ml), or Poly(I:C) (10 μ g/ml) + IL-33 (10 ng/ml). **e**, Representative FACS plots showing ST2 expression when BMDMs were differentiated for 7 days in the presence of M-CSF (20 ng/ml) and IL-4 (10 ng/ml). **f**, Quantification of ST2 expression on BMDMs (WT: $n = 5$, *Il1r1*^{-/-}: $n = 4$, *Il1r1-ExAB*^{-/-}: $n = 4$, data of independent experiments were pooled). Data are presented as mean \pm SD with each dot representing one culture with bone marrow cells from individual mice. *P* was determined using two-tailed t-test (**f**, right panel) and One-way (f) or Two-way ANOVA (**c,d**) with Tukey's post-hoc tests.

The addition of IL-4 during BMDM culture was shown to promote differentiation into alternatively activated macrophages with higher expression of ST2⁹. Indeed, we detected ST2 on the surface of WT macrophages when IL-4 was added one day after plating bone marrow cells, albeit still at low levels (Reviewer Fig. 2e,f). As expected, *Il1r1-ExAB*^{-/-} macrophages did not exhibit a reduction in ST2 surface expression.

At this point, we cannot explain the discrepancies between our results and the data published by Kearley et al.⁸ Thus, we refrain from including the data in the manuscript and hope the reviewer concurs with this decision.

3) Role of the type-1 promoter in neutrophils: resting bone marrow neutrophils express low levels of ST2 mRNA, and ST2 protein is upregulated at the cell surface after IL-33 treatment (5). The authors should determine whether the type-1 promoter controls ST2 expression and/or IL-33 responsiveness in neutrophils.

To study whether the type-1 promoter controls ST2 expression in neutrophils, we have assessed ST2 expression of bone-marrow neutrophils in wildtype, *Il1r1-ExAB*^{-/-}, and *Il1r1*^{-/-} mice (Extended Data Fig. 5h,k,l). Supporting data published by Alves-Filho et al.¹⁰, we found that a small but obvious population of resting neutrophils expressed ST2 protein (Extended Data Fig. 5k). Importantly, this population was still evident in *Il1r1-ExAB*^{-/-} mice even though in slightly lower frequency, and ST2 expression intensity within this population was largely comparable to WT mice (Extended Data Fig. 5l). Of note, using the same staining, we have determined ST2 expression by bone-marrow eosinophils, which worked remarkably well. We thus have included these data in Extended Data Fig. 5i,j, to extend our analysis of ST2 expression by innate type-2 immune cells.

Next, we have purified bone-marrow neutrophils using a three-step Ficoll gradient¹¹ followed by 24 h of stimulation with IL-33, IL12, IL-4, or combinations of IL-33 and IL12 or IL-4. To our surprise, using *Il1r1*^{-/-} neutrophils as controls, no ST2 expression could be detected on stimulated CD11b⁺ Ly6G⁺ neutrophils isolated from wildtype or *Il1r1-ExAB*^{-/-} mice (Reviewer Fig. 3a-c).

Reviewer Figure 3. ST2 expression on neutrophils stimulated *in vitro*. **a**, Pre-gating strategy for the analysis of Ly6G⁺ CD11b⁺ neutrophils. **b,c**, Bone-marrow neutrophils of wildtype, *Il1r1*^{-/-}, and *Il1r1-ExAB*^{-/-} mice were isolated and stimulated with IL-33 (50 ng/ml), IL-12 (10 ng/ml), IL-4 (10 ng/ml), or combinations thereof, for 24 h. Representative FACS plots (**b**) and quantification (**c**) showing no detectable expression of ST2 on Ly6G⁺ CD11b⁺ neutrophils after *ex vivo* stimulation ($n = 3$ per genotype and condition, data represent one of two independent experiments). Data are presented as mean \pm SD with each dot representing one mouse.

We regret that the data shown in Reviewer Fig. 3b,c are not in line with results published by Alves-Filho et al.¹⁰. As it is unclear what accounts for these inconsistencies (e.g. mouse background or experimental procedures), we prefer not to include these results in the manuscript.

4) Detection of ST2 in immune cells and differences in expression levels: quantification of ST2 by flow cytometry is a critical method for this study. However, the authors used different mAb clones for the detection of ST2 in Th1 cells/CTLs (clone DJ8, with PE- or APC-FASER amplification; Fig 1b, 3e) and ILC2s/mast cells (clone DIH9, ST2-BV421; Ext Fig 4c, f). Commercial ST2 antibodies exhibit significant background staining and are best for highly expressing cells. Could it be that the background with clone DIH9 (ST2-BV421 direct fluorescent conjugate) was too high and the specific signal without amplification too low for detection of ST2 expression in Th1/CTLs? The authors should comment about the differences in ST2 expression levels in different immune cells and explain why they developed a more sensitive assay with DJ8 mAb for Th1/CTLs. In the extended data figures, they should compare the different ST2 mAbs on the same cell types (both type-1 and type-2 immune cells) using *Ilr1* KO cells as controls for gating. RMST2-2 is another widely used mAb clone. The authors have all the tools in hands (various *Il1r1* KO cells) to perform a careful comparison of ST2 detection with the three clones (DJ8, DIH9, RMST2-2) on different type-1 and type-2 immune cells. This will be very useful for the scientific community and will increase the interest and impact of the article, beyond the field of T cells.

Indeed, as the reviewer points out, a conventional single-step staining of ST2 is insufficient to achieve a good staining resolution on CTLs or Th1-polarized T cells, as the surface expression of ST2 on these cells is much lower than on Th2-polarized T cells, ILC2s, or mast cells¹². Thus, we have established a staining protocol utilizing a digoxigenin-conjugated primary ST2 antibody, an APC- or PE-conjugated secondary Fab fragment, and two-rounds of FASER-amplification to determine ST2 expression on CTLs or Th1 cells in a highly sensitive manner. However, for assessment of ST2 on mast cells and ILC2s *ex vivo*, we still utilize the commercially available, directly conjugated clone DIH9, as this staining takes less time and is much more cost effective. Further, using unamplified

stainings allows for higher flexibility in the flow-cytometry panel design, as the Faser amplification cannot be used when a staining with biotin-labelled antibodies or specific tandem-conjugates (e.g. APC-Cy7 or PE-Cy7) is required.

To report on this technical aspect in more detail, we have directly compared our highly sensitive staining with a single-step staining using the clones DIH9 and RMST2-2 on high and low ST2 expressing cells (Th2 and Th1 cells), as well as on ST2-deficient T cells as requested by the reviewer (Extended Data Fig. 1d,e) (p. 5, l. 101-105). Of note, prior to this experiment, the directly PE-conjugated antibodies (DIH9 and RMST2-2) were titrated to assure optimal staining quality. Our comparison shows nicely that stainings with directly conjugated DIH9 or RMST2-2 antibodies were sufficient to discriminate ST2⁺ and ST2⁻ cells when cells express high levels of ST2 (e.g. Th2 cells). However, only by using our established highly sensitive staining protocol, we can achieve a good ST2 staining on ST2 low-expressing cells (e.g. Th1 cells) (Extended Data Fig. 1d,e).

We thank the reviewer for the suggestion to specifically explain and elaborate the rationale behind using this enhanced staining procedure to visualize ST2 on antiviral T cells. We think that this comparison illustrates why ST2 expression and the impact of IL-33/ST2-signaling on type-1 polarized T cells, has been underestimated in so many studies.

OTHER POINTS

5) Fig. 2a and p6: it is not clear why the authors do not comment the T-bet/STAT4 ChIP-Seq and CD8⁺ SLEC (abbreviation not defined in the legend) ATAC-seq signals downstream of exons A + B ? No T-bet or STAT4 binding sites ? Sites not conserved in humans?

Indeed, we focused on the ATAC-Seq signal 5kb upstream of exons A and B, as the position of the ATAC-peak downstream of the promoter did not coincide with a conserved DNA element. We have now explained this rationale in the text (p. 6, l. 155-157).

6) Fig 2c: the position of the CNS-5 in the human gene is not clear. ChIP-seq data/binding sites for STAT4 ?

Due to additional DNA elements between this CNS and the first exon A, which are present in humans but not in mice, the distance of the CNS in relation to the first exon is different in humans and should thus not be termed CNS-5 in this context. Nevertheless, we have indicated the position of this particular CNS (CNS-5 in mice) in Fig. 2c with a red arrow and added a description to the figure legend (p. 28, l. 770-773). Further, as pointed out in question 2 of Reviewer 1, we now provide easy access to data used in our manuscript via the following link: <https://genome.ucsc.edu/s/agloehning/BrunnerServeetal2023>. By uploading the RNA-Seq, ChIP-Seq, and ATAC-Seq data to the web-based UCSC genome browser, we offer reviewers and readers a tool to investigate the data in detail and to assess the conservation of all regions. We regret that we have not found suitable STAT4 ChIP-Seq data for human type-1-polarized T cells to complement the mouse T-bet ChIP data.

7) p7 'However, due to hard-to-predict effects on the balance between IL-33-mediated inflammation and tissue repair, blockade of IL-33 or ST2 using therapeutic antibodies has shown conflicting results in preclinical disease models⁴⁷'. This sentence is outdated. Blockade of IL-33 using therapeutic antibodies has shown encouraging efficacy in Phase 2 clinical trials of asthma and COPD, as recently reviewed (6). A more appropriate sentence/paragraph could be 'Modulation of the IL-33–ST2 axis could represent a promising approach in treating inflammatory diseases. Blockade of IL-33 using therapeutic antibodies has shown encouraging efficacy in clinical trials of asthma and COPD (6). However, due to hard-to-predict effects on the balance between IL-33-mediated inflammation and tissue repair, fine-tuned treatment approaches might offer a critical advantage.'

We thank the reviewer for this suggestion and have amended the text accordingly (p. 7, l. 177-180).

References (of reviewer #2)

1. Smithgall MD, et al. (2008) IL-33 amplifies both Th1- and Th2-type responses through its activity on human basophils, allergen-reactive Th2 cells, iNKT and NK cells. *Int Immunol* 20(8):1019-1030.
2. Bourgeois E, et al. (2009) The pro-Th2 cytokine IL-33 directly interacts with invariant NKT and NK cells to induce IFN-gamma production. *Eur J Immunol* 39:1046-1055.
3. Kearley J, et al. (2015) Cigarette smoke silences innate lymphoid cell function and facilitates an exacerbated type I interleukin-33-dependent response to infection. *Immunity* 42(3):566-579.
4. Dagher R, et al. (2020) IL-33-ST2 axis regulates myeloid cell differentiation and activation enabling effective club cell regeneration. *Nat Commun* 11(1):4786.
5. Alves-Filho JC, et al. (2010) Interleukin-33 attenuates sepsis by enhancing neutrophil influx to the site of infection. *Nat Med* 16(6):708-712.
6. Cayrol C & Girard JP (2022) Interleukin-33 (IL-33): A critical review of its biology and the mechanisms involved in its release as a potent extracellular cytokine. *Cytokine* 156:155891.

Reviewer #3

(Remarks to the Author)

The authors characterize a distinct transcriptional process for expression of the gene (*Il1r1*) encoding ST2 (the IL-33 receptor) in type-1 (CTL, Th1) compared to type-2 (Th2) T cells. They go on to show that ablation of the exons used in the “type-1” transcripts of *Il1r1* leads to selective loss of ST2 on those cells and impairs the generation of type-1 effector cells following viral infection. They also find that IL-33 promotes expression of genes which support TCR and costimulatory stimulation, and which are associated with effector cell differentiation.

The report carefully lays out the methodical way in which the authors identify different upstream non-coding exons that are selectively used for *Il1r1* transcripts in CTL/Th1 vs Th2 cells, and present strong evidence that this accurately distinguishes the basis for ST2 expression by type-1 vs type-2 T cells (with Treg aligning with Th2, in this case). Their generation of mice lacking the “type-1” related exons confirmed this, with selective loss of ST2 on Th1 and CTL, and their evidence is compelling that the impact of this mutation almost phenocopies the impact of complete *Il1r1* gene ablation, with respect to type-1 responses.

The data are convincing, and – as the authors point out – this appears to be a rare (if not unique) example in which genes that are expressed by both type-1 and -2 polarized T cells involve distinct transcriptional start sites (regulated by distinct transcription factors). At the same time, once the authors find that loss of exons A/B (*ExA/B*) leads to almost complete loss of ST2 on type-1 cells, much of the subsequent data corresponds with earlier work, showing a cell-intrinsic role for ST2 in the anti-viral response of these cells. This limitation is somewhat balanced by the authors going deeper than previous studies with respect to the gene expression changes driven by IL-33 in type-1 and -2 T cells (Fig. 6), and by their data indicating that IL-33 stimulation specifically enhances effector cell differentiation (Fig. 5). Nevertheless, there are remaining concerns.

We thank the reviewer for the very positive feedback on our study. In particular, we are pleased to hear that our evidence for a dedicated type-1 promoter of the *Il1r1* gene is very convincing. Using the LCMV infection model *in vivo* in conjunction with our newly generated *ExAB*- or *ExC*-deficient mice allowed us to test precisely to which extent the type-1 promoter-deficient T cells recapitulate the reported phenotype of ST2-deficient T cells, while the data presented in Fig. 5 and Fig. 6 shed new light on IL-33/ST2 signaling and its functions in antiviral T cell responses. In addition, as shown in Fig. 3 our findings of alternative *Il1r1* promoters for the first time allow T cell lineage-selective targeting of ST2 expression even in WT T cells.

1) The studies on impaired generation of short-lived effector cells (SLEC) by CD8⁺ T cells lacking *ExA/B* are convincing but the consequences for memory differentiation are underdeveloped. Numerous panels in Figs. 4 and 5 present the frequencies and numbers of SLEC phenotype cells, but do not show the corresponding data for the MPEC (KLRG1⁺ CD127⁺) cells – based on panels such as Fig. 5e, one would presume these cells are increased in frequency. It is unclear whether MPEC are unchanged (or, perhaps more likely, modestly reduced) in absolute numbers. These data are needed for a more complete picture of how defective ST2 regulation (*ExA/B* knockout) or expression (*Il1r1* knockout) affects CD8⁺ T cells differentiating along the effector vs memory pathways.

We thank the reviewer for the recommendation to present the data for KLRG1⁺ CD127⁺ MPECs to further elaborate the impact of IL-33/ST2 signaling on these cells. The reviewer correctly assumes that, in contrast to SLECs, absolute counts of MPECs per organ are only modestly decreased in *Il1r1-ExAB*^{-/-} and *Il1r1*^{-/-} mice compared to wildtype mice. As suggested, we have added frequencies and absolute numbers of MPECs to all figures where SLEC frequencies were shown and amended the text accordingly (Fig. 5h,i, Fig. 6j,k, and Extended Data Fig. 6m) (p. 10-11, l. 290-299; p. 13, l. 368-370; p. 16, l. 450-454).

2) Along the same lines, the authors' analysis of the anti-viral response of *ExA/B* and *Il1r1* deficient T cells appears to be limited to the first ~10 days following infection, and exclusively (as far as this reviewer can ascertain) focuses on cells in the spleen. It would be important to know how mutation of *ExA/B* impacts generation of the various subsets of circulating memory CD8⁺ and CD4⁺ T cells during these responses, and to extend the scope of the work to analysis of at least some non-lymphoid tissues to determine the effect on generation of resident memory T cells. As it stands, it is unclear

whether the loss of normal ST2 regulation leads to a selective loss of type-1 effector differentiation – with, perhaps, minimal effect on establishment of long-lived memory – or a more generalized defect that is magnified in but not exclusive to the effector pool.

Indeed, we have focused our analyses to the first ~10 days after LCMV infection and primarily analyzed spleens, as well as in some cases livers, of infected animals (Extended Data Fig. 6a-f, n-r). ST2 expression on antiviral T cells and phenotypic differences between ST2-competent and ST2-deficient T cells are most pronounced in the acute phase of the infection^{6,12}. Thus, this time frame seemed optimal to accurately compare the response of *Il1rl1-ExAB^{-/-}* T cells with *Il1rl1^{-/-}* T cells and WT T cells. As mentioned in our reply to point 1) of this reviewer, the impact of IL-33–ST2 signaling does not selectively affect short-lived effector cells, yet the effect on MPEC numbers is less pronounced.

To assess the role of the type-1 promoter and the impact of ST2 signaling on memory T cell formation, we have adoptively transferred naive P14, P14 *Il1rl1^{-/-}*, or P14 *Il1rl1-ExAB^{-/-}* T cells into WT recipients, which were subsequently infected with LCMV. After 30 days, we have labeled T cells in circulation by intravenous injection of CD90.2 antibody and sacrificed the mice afterwards (Extended Data Fig. 9a). Analysis of spleens, lungs, salivary glands, and kidneys showed that a lack of ST2 or of the type-1 ST2 promoter reduced the amount of P14 memory cells found in these organs (Extended Data Fig. 9b-d). P14 cells of all genotypes predominantly formed effector memory cells (Tem) and no major impact on frequencies of central memory cells (Tcm) was observed (Extended Data Fig. 9e-g). Likewise, we have found that both circulating and tissue-resident memory T cells were formed, even in the absence of ST2 or the type-1 promoter (Extended Data Fig. 9h-j). Lastly, quantification of *in vivo* labeled and non-labeled P14 cells showed that numerical differences between P14 memory and P14 *Il1rl1^{-/-}* memory or P14 *Il1rl1-ExAB^{-/-}* memory cells were more pronounced in the intravascular compartment (Extended Data Fig. 9k).

We greatly appreciate the suggestion of the reviewer to investigate the impact of defective ST2 regulation on MPEC/SLEC fate decision and memory cell formation. We have complemented our study with these newly acquired *in vivo* data and amended our manuscript at several positions (p. 10-11, l. 299-306; p. 16, l. 450-454). We now report that deletion of the type-1 ST2 promoter or ST2 largely impairs SLEC expansion, but to a lesser extent, also affects MPEC numbers and consequently the number of antiviral memory CTLs generated. In our opinion, these changes made in response to the reviewer's comments have substantially improved the overall quality of our study.

References

- 1 Thomsen, A. R. & Marker, O. The complementary roles of cellular and humoral immunity in resistance to re-infection with LCM virus. *Immunology* **65**, 9-15 (1988).
- 2 Tebo, A. E. *et al.* Rapid recruitment of virus-specific CD8 T cells restructures immunodominance during protective secondary responses. *J Virol* **79**, 12703-12713, doi:10.1128/JVI.79.20.12703-12713.2005 (2005).
- 3 Cerny, A., Sutter, S., Bazin, H., Hengartner, H. & Zinkernagel, R. M. Clearance of lymphocytic choriomeningitis virus in antibody- and B-cell-deprived mice. *J Virol* **62**, 1803-1807, doi:10.1128/JVI.62.5.1803-1807.1988 (1988).
- 4 Rahman, A. H. *et al.* Antiviral memory CD8 T-cell differentiation, maintenance, and secondary expansion occur independently of MyD88. *Blood* **117**, 3123-3130, doi:10.1182/blood-2010-11-318485 (2011).
- 5 Schmitz, J. *et al.* IL-33, an interleukin-1-like cytokine that signals via the IL-1 receptor-related protein ST2 and induces T helper type 2-associated cytokines. *Immunity* **23**, 479-490, doi:10.1016/j.immuni.2005.09.015 (2005).
- 6 Baumann, C. *et al.* Memory CD8(+) T Cell Protection From Viral Reinfection Depends on Interleukin-33 Alarmin Signals. *Front Immunol* **10**, 1833, doi:10.3389/fimmu.2019.01833 (2019).
- 7 Joshi, N. S. *et al.* Inflammation directs memory precursor and short-lived effector CD8(+) T cell fates via the graded expression of T-bet transcription factor. *Immunity* **27**, 281-295, doi:10.1016/j.immuni.2007.07.010 (2007).
- 8 Kearley, J. *et al.* Cigarette smoke silences innate lymphoid cell function and facilitates an exacerbated type I interleukin-33-dependent response to infection. *Immunity* **42**, 566-579, doi:10.1016/j.immuni.2015.02.011 (2015).
- 9 Faas, M. *et al.* IL-33-induced metabolic reprogramming controls the differentiation of alternatively activated macrophages and the resolution of inflammation. *Immunity* **54**, 2531-2546 e2535, doi:10.1016/j.immuni.2021.09.010 (2021).
- 10 Alves-Filho, J. C. *et al.* Interleukin-33 attenuates sepsis by enhancing neutrophil influx to the site of infection. *Nat Med* **16**, 708-712, doi:10.1038/nm.2156 (2010).
- 11 Ubags, N. D. J. & Suratt, B. T. Isolation and Characterization of Mouse Neutrophils. *Methods Mol Biol* **1809**, 45-57, doi:10.1007/978-1-4939-8570-8_4 (2018).
- 12 Baumann, C. *et al.* T-bet- and STAT4-dependent IL-33 receptor expression directly promotes antiviral Th1 cell responses. *Proc Natl Acad Sci U S A* **112**, 4056-4061, doi:10.1073/pnas.1418549112 (2015).

Decision Letter, first revision:

28th Sep 2023

Dear Dr. Löhning,

Thank you for submitting your revised manuscript "A type-1 immunity-restricted promoter of the IL-33 receptor gene directs antiviral T cell responses" (NI-A35288A). It has now been seen by the original referees and their comments are below. The reviewers find that the paper has improved in revision, and therefore we'll be happy in principle to publish it in Nature Immunology, pending minor revisions to satisfy the referees' final requests and to comply with our editorial and formatting guidelines.

We will now perform detailed checks on your paper and will send you a checklist detailing our editorial and formatting requirements in about a week. Please do not upload the final materials and make any revisions until you receive this additional information from us.

If you had not uploaded a Word file for the current version of the manuscript, we will need one before beginning the editing process; please email that to immunology@us.nature.com at your earliest convenience.

Thank you again for your interest in Nature Immunology. Please do not hesitate to contact me if you have any questions.

Sincerely,

Stephanie Houston, PhD
Senior Editor
Nature Immunology

Reviewer #2 (Remarks to the Author):

The authors have satisfactorily addressed my previous concerns. The manuscript has been greatly improved thanks to the addition of new data, and modifications in the text of the manuscript.

The comparison of the different anti-ST2 mAbs in the flow cytometry assays (Extended data Fig 1d, e) will be very useful for the scientific community. The sensitivity of the enhanced staining procedure with clone DJ8 + FASER amplification is impressive.

The revised manuscript will be an important contribution to the field.

Reviewer #3 (Remarks to the Author):

The authors have responded well to the concerns raised previously, by both including requested data

(which involved several additional studies) and revising the text.

Regarding the authors' findings on the generation of memory ST2^{-/-} or ExA/B^{-/-} CD8 T cells in lymphoid and non-lymphoid tissues, it might be worth highlighting (e.g., in the discussion) that impaired ST2 expression had minimal (if any) statistically significant effect on likely resident memory T cells (i.e., those cells that are not stained by intravenous antibody labeling).

Since some effector-like CD8⁺ T cells persist in the circulation well past d30 in the response to acute LCMV infection, it is therefore possible that failure of CD8⁺ T cells to express ST2 is quite selective for effector/effector-like cell generation, while generation of bona fide memory CD8⁺ T cells is, perhaps, completely unaffected. While further work would be needed to thoroughly test this interpretation, it would be an interesting hypothesis to raise since it suggests a quite focused role for IL-33 sensitivity in generation of effector (but not memory) CD8⁺ T cells, during an acute LCMV infection.

Author Rebuttal, first revision:

Point-by-point Reply to Reviewers' Comments:

A type-1 immunity-restricted promoter of the IL-33 receptor gene directs antiviral T cell responses (NI-A35288A)

We are thankful for the reviewers' feedback concerning our revised work and are grateful that in principle the manuscript has been deemed suitable for publication in Nature Immunology. Below, please find our responses to the reviewer's final requests (Blue text), outlining the specific changes we have implemented in our manuscript. Within the manuscript text, we have used blue underlined text to indicate changes made to the original version.

Reviewer #2

The authors have satisfactorily addressed my previous concerns. The manuscript has been greatly improved thanks to the addition of new data, and modifications in the text of the manuscript.

The comparison of the different anti-ST2 mAbs in the flow cytometry assays (Extended data Fig 1d, e) will be very useful for the scientific community. The sensitivity of the enhanced staining procedure with clone DJ8 + FASER amplification is impressive.

The revised manuscript will be an important contribution to the field.

We thank the reviewer for this very supportive assessment.

Reviewer #3

The authors have responded well to the concerns raised previously, by both including requested data (which involved several additional studies) and revising the text.

Regarding the authors' findings on the generation of memory ST2^{-/-} or ExA/B^{-/-} CD8 T cells in lymphoid and non-lymphoid tissues, it might be worth highlighting (e.g., in the discussion) that impaired ST2 expression had minimal (if any) statistically significant effect on likely resident memory T cells (i.e., those cells that are not stained by intravenous antibody labeling).

Since some effector-like CD8⁺ T cells persist in the circulation well past d30 in the response to acute LCMV infection, it is therefore possible that failure of CD8⁺ T cells to express ST2 is quite selective for effector/effector-like cell generation, while generation of bona fide memory CD8⁺ T cells is, perhaps, completely unaffected. While further work would be needed to thoroughly test this interpretation, it would be an interesting hypothesis to raise since it suggests a quite focused role for IL-33 sensitivity in generation of effector (but not memory) CD8⁺ T cells, during an acute LCMV infection.

We thank the reviewer for this suggestion. Now we have included a statement addressing the apparently limited role for ST2 signals in the formation of antiviral tissue-resident CD8⁺ memory T cells in the Results section (p.10, l.255-257) and have added the resulting hypothesis to the Discussion section (p.14-15, l.387-391).

Final Decision Letter:

Dear Dr. Löhning,

I am delighted to accept your manuscript entitled "A type-1 immunity-restricted promoter of the IL-33 receptor gene directs antiviral T cell responses" for publication in an upcoming issue of Nature Immunology.

Over the next few weeks, your paper will be copyedited to ensure that it conforms to Nature Immunology style. Once your paper is typeset, you will receive an email with a link to choose the appropriate publishing options for your paper and our Author Services team will be in touch regarding any additional information that may be required.

Please note that *Nature Immunology* is a Transformative Journal (TJ). Authors may publish their research with us through the traditional subscription access route or make their paper immediately open access through payment of an article-processing charge (APC). Authors will not be required to make a final decision about access to their article until it has been accepted. [Find out more about Transformative Journals](https://www.springernature.com/gp/open-research/transformative-journals).

Authors may need to take specific actions to achieve [compliance with funder and institutional open access mandates](https://www.springernature.com/gp/open-research/funding/policy-compliance-faqs). If your research is supported by a funder that requires immediate open access (e.g. according to [Plan S principles](https://www.springernature.com/gp/open-research/plan-s-compliance)) then you should select the gold OA route, and we will direct you to the compliant route where possible. For authors selecting the subscription publication route, the journal's standard licensing terms will need to be accepted, including [self-archiving policies](https://www.springernature.com/gp/open-research/policies/journal-policies). Those licensing terms will supersede any other terms that the author or any third party may assert apply to any version of the manuscript.

Your paper will be published online soon after we receive your corrections and will appear in print in the next available issue. Content is published online weekly on Mondays and Thursdays, and the embargo is set at 16:00 London time (GMT)/11:00 am US Eastern time (EST) on the day of publication. Now is the time to inform your Public Relations or Press Office about your paper, as they might be interested in promoting its publication. This will allow them time to prepare an accurate and satisfactory press release. Include your manuscript tracking number (NI-A35288B) and the name of the journal, which they will need when they contact our office.

About one week before your paper is published online, we shall be distributing a press release to news organizations worldwide, which may very well include details of your work. We are happy for your institution or funding agency to prepare its own press release, but it must mention the embargo date and *Nature Immunology*. Our Press Office will contact you closer to the time of publication, but if you or your Press Office have any enquiries in the meantime, please contact press@nature.com.

Also, if you have any spectacular or outstanding figures or graphics associated with your manuscript - though not necessarily included with your submission - we'd be delighted to consider them as candidates for our cover. Simply send an electronic version (accompanied by a hard copy) to us with a possible cover caption enclosed.

If you have not already done so, we strongly recommend that you upload the step-by-step protocols used in this manuscript to the Protocol Exchange. Protocol Exchange is an open online resource that allows researchers to share their detailed experimental know-how. All uploaded protocols are made freely available, assigned DOIs for ease of citation and fully searchable through nature.com. Protocols can be linked to any publications in which they are used and will be linked to from your article. You can also establish a dedicated page to collect all your lab Protocols. By uploading your Protocols to Protocol Exchange, you are enabling researchers to more readily reproduce or adapt the methodology you use, as well as increasing the visibility of your protocols and papers. Upload your Protocols at www.nature.com/protocolexchange/. Further information can be found at www.nature.com/protocolexchange/about .

Please note that we encourage the authors to self-archive their manuscript (the accepted version before copy editing) in their institutional repository, and in their funders' archives, six months after publication. Nature Portfolio recognizes the efforts of funding bodies to increase access of the research they fund, and strongly encourages authors to participate in such efforts. For information about our editorial policy, including license agreement and author copyright, please visit www.nature.com/ni/about/ed_policies/index.html

Sincerely,

Stephanie Houston, PhD
Senior Editor
Nature Immunology